# The geologic history of marine dissolved organic carbon from iron oxides

Nir Galili[1,2 ✉], Stefano M. Bernasconi[1], Alon Nissan[3,4], Uria Alcolombri[5], Giorgia Aquila[1], Marcella Di Bella[6], Thomas M. Blattmann[1], Negar Haghipour[1,7], Francesco Italiano[8], Madalina Jaggi[1], Ifat Kaplan-Ashiri[9], Kang Soo Lee[10], Maxwell A. Lechte[11], Cara Magnabosco[1], Susannah M. Porter[12], Maxim Rudmin[13], Robert G. M. Spencer[14], Roman Stocker[3], Zhe Wang[1], Stephan Wohlwend[1] & Jordon D. Hemingway[1]

Dissolved organic carbon (DOC) is the largest reduced carbon reservoir in modern oceans[1,2]. Its dynamics regulate marine communities and atmospheric $CO_2$ levels[3,4], whereas [13]C compositions track ecosystem structure and autotrophic metabolism[5]. However, the geologic history of marine DOC remains largely unconstrained[6,7], limiting our ability to mechanistically reconstruct coupled ecological and biogeochemical evolution. Here we develop and validate a direct proxy for past DOC signatures using co-precipitated organic carbon in iron ooids. We apply this to 26 marine iron ooid-containing formations deposited over the past 1,650 million years to generate a data-based reconstruction of marine DOC signals since the Palaeoproterozoic. Our predicted DOC concentrations were near modern levels in the Palaeoproterozoic, then decreased by 90–99% in the Neoproterozoic before sharply rising in the Cambrian. We interpret these dynamics to reflect three distinct states. The occurrence of mostly small, single-celled organisms combined with severely hypoxic deep oceans, followed by larger, more complex organisms and little change in ocean oxygenation and finally continued organism growth and a transition to fully oxygenated oceans[8,9]. Furthermore, modern DOC is [13]C-enriched relative to the Proterozoic, possibly because of changing autotrophic carbon-isotope fractionation driven by biological innovation. Our findings reflect connections between the carbon cycle, ocean oxygenation and the evolution of complex life.

Marine dissolved organic carbon (DOC) contains $660 \times 10^{15}$ g of carbon (gC), roughly equal to the pre-industrial atmospheric $CO_2$ reservoir[1,2]. DOC is mainly produced by (1) planktonic communities in sunlit surface waters, in which dissolved compounds are either exuded directly by photosynthetic autotrophs or released during ecosystem interactions—for example, primary production, 'sloppy feeding', viral lysis and microbial loop interactions[10]—and (2) solubilization of sinking particulate organic carbon (POC) in the ocean interior. Subsequent transformation of solubilized carbon into more recalcitrant DOC compounds constitutes the microbial carbon pump[1,4,11–13]. By contrast, DOC is mainly consumed by POC scavenging and heterotrophic respiration in the deep ocean with rates that depend on particle sinking speed, temperature, dissolved $O_2$ content ($[O_2]$) and DOC concentration ($[DOC]$) by dilution effects and uptake kinetics[1,14]. Estimates suggest that 30% of deep-ocean DOC forms by the microbial carbon pump over decadal timescales (so-called labile, semi-labile and semi-refractory DOC),

whereas the remainder has been circulating subject to slow respiration with a residence time of about 30,000 years (that is, 30 ocean overturning cycles; so-called refractory to ultra-refractory DOC, which includes carboxyl-rich alicyclic molecules)[13,15,16]. Given the long residence time of this refractory component, small changes in consumption rates could drive large perturbations in DOC reservoir size—and thus atmospheric $CO_2$ levels ($pCO_2$)—over approximately $10^3$–$10^5$ year timescales[3,4].

Deeper in the history of Earth, DOC dynamics have been invoked to explain global glaciation (snowball Earth) events[17], deep-ocean oxygenation and the radiation of marine eukaryotes[6,18–22]. Specifically, in the Neoproterozoic Era (1,000–540 Ma), ref. 6 proposed that slow particle sinking before the evolution of multicellular and biomineralizing organisms[23], combined with an anoxic deep ocean[9,24], prompted a stronger microbial loop, slower respiration at depth and accumulation of a long-lived, recalcitrant DOC reservoir that was about $10^2$–$10^3$ times its modern size. Subsequent evolution of faster sinking eukaryotes

[1]Geological Institute, Department of Earth and Planetary Sciences, ETH Zurich, Zurich, Switzerland. [2]Department of Earth and Planetary Sciences, Weizmann Institute of Science, Rehovot, Israel. [3]Department of Civil, Environmental, and Geomatic Engineering, Institute of Environmental Engineering, ETH Zurich, Zurich, Switzerland. [4]Institute for Environmental Sciences, The Robert H. Smith Faculty of Agriculture, Food and Environment, The Hebrew University of Jerusalem, Rehovot, Israel. [5]Institute for Life Sciences, Department of Plant and Environmental Sciences, The Hebrew University of Jerusalem, Jerusalem, Israel. [6]National Institute of Oceanography and Applied Geophysics, Sgonico, Italy. [7]Laboratory of Ion Beam Physics, ETH Zurich, Zurich, Switzerland. [8]National Institute of Geophysical and Volcanology, Palermo, Italy. [9]Department of Chemical Research Support, Weizmann Institute of Science, Rehovot, Israel. [10]Department of Mechanical Engineering, Ulsan National Institute of Science and Technology, Ulsan, South Korea. [11]Department of Earth and Planetary Sciences, McGill University, Montreal, Quebec, Canada. [12]Department of Earth Science, University of California, Santa Barbara, Santa Barbara, CA, USA. [13]Division for Geology, Tomsk Polytechnic University, Tomsk, Russia. [14]Earth, Ocean, and Atmospheric Science, Florida State University, Tallahassee, FL, USA. ✉e-mail: ngalili@ethz.ch

would increase POC burial flux and thus atmospheric $O_2$ levels, whereas colder climates would increase $O_2$ solubility; both mechanisms would drive deep-ocean oxygenation and transient respiration of this large DOC reservoir[6]. Furthermore, DOC exhibits lower $\delta^{13}C$ values (‰ Vienna Pee Dee Belemnite (VPDB); Methods) relative to dissolved inorganic carbon (DIC = $CO_{2(aq)}$ + $H_2CO_3$ + $HCO_3^-$ + $CO_3^{2-}$) because of photosynthetic carbon-isotope fractionation ($\varepsilon_p \approx \delta^{13}C_{DIC} - \delta^{13}C_{org}$, where 'org' refers to bulk organic carbon). Transient DOC respiration to $CO_2$ would thus produce $^{13}C$-depleted DIC and could explain anomalously large negative isotope excursions observed in Neoproterozoic carbonate rocks (that is, because $\delta^{13}C_{carb} \approx \delta^{13}C_{DIC}$), particularly if the amount of DOC oxidized is large relative to the DIC reservoir size[6,18,20]. According to this model, a large DOC reservoir is thus a fundamental requirement to explain the Neoproterozoic geologic record.

Still, other studies reject the large DOC reservoir hypothesis. For example, ref. 25 suggested that respiration of this reservoir would deplete the surface of Earth of terminal electron acceptors ($O_2$, $SO_4^{2-}$) on timescales much shorter than those of observed isotope excursions. Similarly, ref. 26 demonstrated that $\delta^{13}C_{org}$ and $\delta^{13}C_{carb}$ are coupled in the latest Neoproterozoic, requiring respiration of an alternative carbon source such as methane ($CH_4$)—which exhibits lower $\delta^{13}C$ values than both DOC and DIC—as a driver of isotope excursions. Furthermore, some authors have proposed that negative isotope excursions—despite being globally distributed and broadly synchronous—do not reflect the general state of the global carbon cycle, but instead implicate mechanisms such as late-stage diagenesis[27], authigenic carbonate precipitation[28] or marine transgressions[29]. Consistent with these interpretations, a recent model proposed that decreased DOC respiration in the Proterozoic was balanced by decreased production via the microbial carbon pump in anoxic deep oceans, leading to largely invariant [DOC] through geologic time[7].

Such disagreement on the geologic history of marine DOC results from a dearth of direct records. To address this, we developed a proxy based on the content and $\delta^{13}C$ value of organic carbon co-precipitated with iron (oxyhydr)oxides (termed Fe-OC; here including amorphous ferrihydrite and crystalline goethite and haematite). We show that Fe-OC loadings and $^{13}C$ compositions quantitatively track [DOC] and $\delta^{13}C_{DOC}$. We then apply this proxy to geologically preserved marine iron ooids—sand-sized grains with concentric iron (oxyhydr)oxide laminae that form in well-constrained environments[30]—to generate one of the first data-based reconstructions of marine DOC signals since the late Palaeoproterozoic.

## Fe-OC signals reflect [DOC] and $\delta^{13}C_{DOC}$

Our approach leverages two key observations: (1) iron (oxyhydr)oxides entrap and preserve DOC within their crystal lattice during precipitation[31–34] and (2) the resulting Fe-OC content depends on carbon-to-iron mole ratio [DOC/Fe(III)] of the initial solution[35]. Therefore, if Fe(III) concentration is known, then Fe-OC loading and $^{13}C$ composition[32] will quantitatively reflect [DOC] and $\delta^{13}C_{DOC}$. However, several factors currently hinder the utility of Fe-OC as a proxy for past DOC signals, including unknown (1) $^{13}C$ fractionation factors ($\Delta^{13}C = \delta^{13}C_{DOC} - \delta^{13}C_{Fe-OC}$); (2) loading curves for geologically preserved iron oxides (that is, crystalline goethite and haematite; see ferrihydrite in ref. 35); and (3) impacts of marine DOC molecular composition (see soil humics in ref. 35), functional-group diversity[31,33] and environmental conditions (for example, temperature, pH, dissolved silica concentration).

To fill these knowledge gaps, we co-precipitated goethite and haematite with three molecularly unique DOC sources across several synthesis conditions and an approximately three orders of magnitude DOC/Fe(III) range, and we measured the resulting Fe-OC content and $^{13}C$ composition to generate Fe-OC loading and $\Delta^{13}C$ calibration curves (Methods). DOC sources include (1) modern-marine analogue DOC from microcosm cultures containing diatoms and marine heterotrophic

bacteria (termed M-DOC), representing eukaryote-dominated communities; (2) cyanobacterial biomass leachate (termed C-DOC), representing prokaryote-dominated communities; and (3) dissolved fulvic acid (termed FA), representing modern terrestrial humic substances[36] (Supplementary Discussion, Supplementary Fig. 1 and Supplementary Table 1). Because marine DOC naturally occurs at low concentrations, sources (1) and (2) were used to overcome the logistically impractical need to process several thousand litres of seawater to obtain adequate substrate. Synthesis conditions were chosen to test calibration curve sensitivity to (1) temperature (4–95 °C); (2) pH (1.5–11.5); (3) absolute DOC (0.04–35 mM) and Fe(III) (20–396 mM) concentrations; (4) dissolved silica concentrations (0.5–2.1 mM); (5) duration (5–120 days); (6) Fe(III) source [$FeCl_3$ compared with $Fe(NO_3)_3$]; and (7) DOC addition timing (pre- or post-precursor ferrihydrite formation). We also tested the effect of removing ferrihydrite-bound and adsorbed DOC, as this is unlikely to survive in the geologic record (Methods).

In all cases, iron oxides are mineralogically pure (Supplementary Figs. 2–7) and show dose-dependent co-precipitated Fe-OC loadings as well as $\Delta^{13}C$ values that approach zero with increasing DOC/Fe(III). Although signals diverge at unrealistically high temperature (95 °C) and pH (11.5), those within environmentally relevant conditions are remarkably consistent, with loading and $\Delta^{13}C$ typically ranging by ≤0.1 wt% and ≤2.5‰ at the same DOC/Fe(III) independent of temperature; pH; absolute DOC, Fe(III) and dissolved silica concentrations; duration; Fe(III) source; and DOC addition timing (Supplementary Discussion and Supplementary Figs. 8–15). However, removing ferrihydrite-bound and adsorbed DOC decreases observed loadings by up to 50% and shifts $\Delta^{13}C$ by up to 10‰ relative to raw fractions (Supplementary Discussion and Supplementary Figs. 16 and 17). These results indicate that Fe-OC signals are robust to potential changes in environmental conditions through geologic time—including in dissolved silica concentrations, which were probably significantly higher in Precambrian oceans relative to today[37,38]—and highlight the importance of removing non-geologically preserved material before building calibration curves.

Using results from environmentally relevant conditions, we generated Fe-OC loading and $\Delta^{13}C$ calibration curves to quantitatively predict past [DOC] and $\delta^{13}C_{DOC}$ from measured geologic iron ooid signals. We specifically fit loadings using a power-law function with exponent ≤1 (that is, Freundlich sorption isotherm)[39] and $\Delta^{13}C$ using a dual-exponential function to empirically capture observed trends, including that values converge to zero at high DOC/Fe(III) (Supplementary Discussion and Supplementary Figs. 18–22). For both minerals, all calibration curve fits exhibit root-mean square errors ≤0.09 wt% for Fe-OC loadings and ≤1.3‰ for $\Delta^{13}C$ with ordinary least squares $r^2 \geq 0.83$ (typically $r^2 \geq 0.95$), indicating the robustness of our approach (Supplementary Table 2). Finally, when comparing across sources, Fe-OC loadings depend strongly on DOC slope ratio ($S_R$), an absorbance-based measure of molecular weight and aromaticity[40] (Supplementary Fig. 23). This probably reflects the known importance of DOC methoxy and carboxyl groups in determining co-precipitation strength and loading[31,33]. We, therefore, consider several molecular composition evolutionary scenarios when reconstructing Earth-history DOC records.

## Iron ooids capture ambient Fe-OC signals

Iron ooids are spheroidal or ovoidal grains up to 2 mm in size that are composed of thin, concentric iron (oxyhydr)oxide layers[41]. Importantly, iron ooid precipitation requires an agitated aqueous environment—they are found at wave base or shallower (that is, up to about 80 m water depth) and confidently identified as marine in origin[30,42]. This is supported by the discovery of two modern sites containing goethite ooids that have been actively forming since about 10 ka (based on $^{14}C$ activities; Supplementary Discussion and Supplementary Table 3) due to the input of iron-bearing hydrothermal fluids[43,44]. Although the source of iron to ooids may have varied in the geologic past

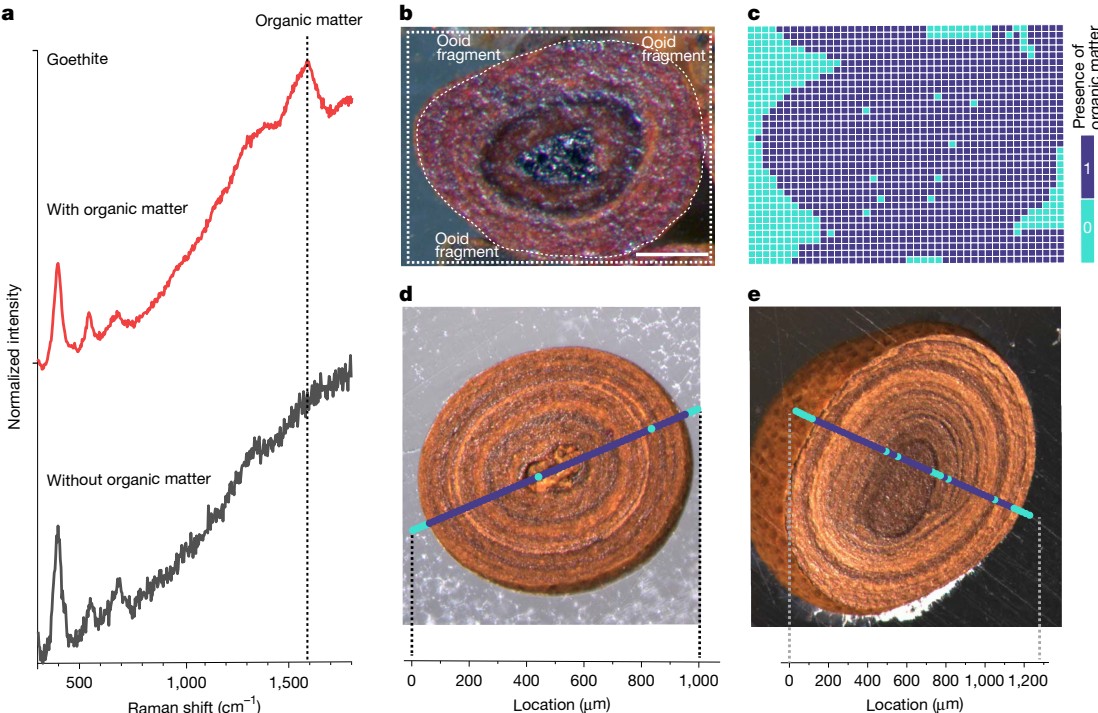

**Fig. 1 | Fe-OC spatial distribution within goethite ooids. a**, Raman spectra of synthetic goethite synthesized without organic matter (black line) and of a natural goethite ooid-containing organic matter (red line). The synthetic goethite was synthesized at 70 °C using $FeCl_3$ as the Fe(III) source at pH 7 and aged for 60 days. The red spectrum is from a goethite ooid sample from the Aseri Formation (Ordovician, Estonia), the same as in **d**. A diagnostic peak at approximately 1,590 cm$^{-1}$ (G-band) indicates the presence of organic carbon in goethite (Methods). **b**,**c**, An optical microscopy image and corresponding 2D Raman map of a modern ooid from Panarea Island, Italy, showing the presence (purple) and absence (turquoise) of Fe-OC at each pixel. **d**,**e**, Example line scans of goethite ooids from the Aseri Formation and the Sillaoru Formation (both Ordovician, Estonia), respectively, showing the spatial distribution of Fe-OC. The presence of OM was identified using the ratio $I_{1,450-1,700}/I_{1,200-1,450}$ (where $I$ represents the integrated intensity within the defined spectral windows), and a threshold value determined from the raw data was used to convert calculated values to a binary format indicating the presence or absence of organic matter (Methods). Scale bar, 100 μm (**b**).

(for example, upward diffusion after microbial iron reduction in deep sediments), their formation mechanism has remained similar: reduced iron migrates towards the sediment–water interface, in which it either forms iron-rich silicates (that is, chamosite, nontronite) or is directly oxidized to amorphous iron (oxyhydr)oxides that ripen to crystalline phases (that is, goethite, haematite; the subjects of this study) as concentric layers around a nucleus by wave action (see Supplementary Discussion for further description of formation pathways).

Because they form in specific environments at the sediment–water interface, we first validate that iron ooids robustly capture ambient marine DOC signals—which on continental shelves includes labile to refractory DOC[13]—before using our Fe-OC proxy to generate Earth-history records. By contrast, entrainment of detrital POC grains and/or iron-oxidizing bacterial biofilms would lead to Fe-OC signals that are decoupled from those of marine DOC. Importantly, both of these carbon sources would lead to patchy, rather than homogeneously distributed, Fe-OC within an ooid grain (Supplementary Discussion). To test the potential importance of local/detrital Fe-OC sources, we generated Raman microspectroscopy raster line scans and maps of modern and ancient iron oxide ooid cross sections (2.6 μm spatial resolution; Methods). The presence of Fe-OC in goethite and haematite leads to a diagnostic peak at 1,590 cm$^{-1}$ (G-band) that is not present in carbon-free materials (Fig. 1a). We, therefore, treat the presence or absence of this peak as a binary signal for the presence or absence of Fe-OC at that location (Supplementary Discussion).

In contrast to predictions for local, grain-scale carbon inputs, we find that Fe-OC is consistently detected in all iron oxide laminae for all measured ooids, indicating homogeneous carbon distributions throughout the grains (Fig. 1b–e). This result implies that ambient DOC

is incorporated into the crystal lattice during dissolution–reprecipitation of amorphous iron (oxyhydr)oxides as they ripen to crystalline phases[45]. To further validate that Fe-OC is not driven by iron-oxidizing biofilms, we extracted DNA from modern goethite ooids and performed 16S rRNA gene amplicon sequencing (Methods). Although sample replication and data quality are limited because of non-ideal sample storage conditions, well-known iron-oxidizing taxa were never detected (that is, *Mariprofundus*, *Gallionellaceae*, *Pseudoalteromonadaceae* and *Hyphomonadaceae*; Supplementary Discussion, Supplementary Fig. 24 and Supplementary Tables 4 and 5). Combined, these results strongly support the interpretation that iron ooid Fe-OC captures continental shelf marine DOC signals.

## Ooidal Fe-OC loading and δ$^{13}$C records

We applied our Fe-OC proxy to a collection of hand-picked geologic iron ooid grains. Each sample in our record—including from both modern sites—was assessed mineralogically, petrographically and stratigraphically to exclude samples affected by diagenetic alteration (Supplementary Discussion and Supplementary Figs. 25–123). Furthermore, many of our samples were analysed previously for oxygen- and iron-isotope compositions[42,46], which confirms formation in seawater. Resulting Fe-OC signals are uncorrelated with (palaeo)latitude, indicating no obvious spatial bias (Supplementary Fig. 124). Still, because ooids form in coastal to continental shelf regions—rather than the deep ocean—recorded signals may be subject to local variability. To test this, we analysed several within-period samples separated by about 10$^2$– 10$^3$ km, including from two modern, two Cretaceous, nine Jurassic, four Ordovician and two Tonian formations. In all cases, within-period

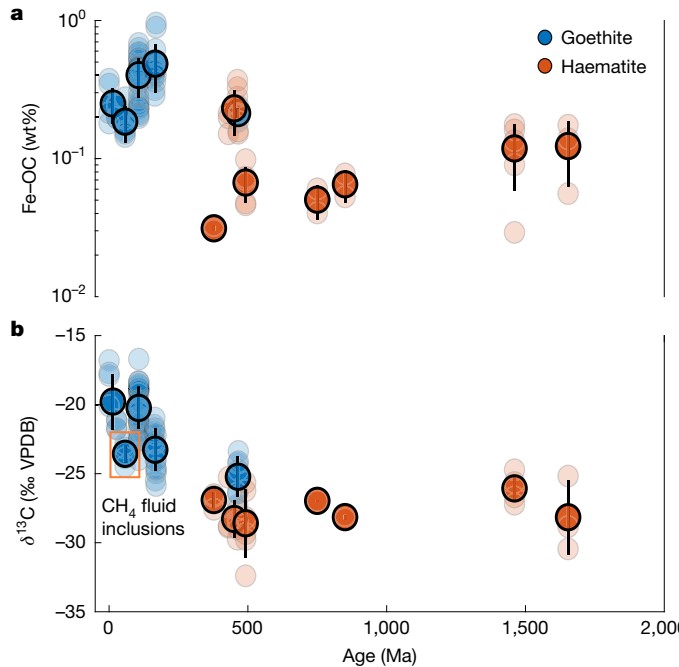

**Fig. 2 | The iron ooid organic carbon records. a,b**, Fe-OC loadings (wt%) (**a**) and $\delta^{13}C$ values (‰ VPDB) (**b**) since the late Palaeoproterozoic, separated by mineralogy (blue represents goethite; orange represents haematite). Opaque markers and vertical black lines are 100 Ma time-binned averages and $\pm 1\sigma$ uncertainties, whereas semi-transparent markers are individual sample results. The orange box indicates the PETM-aged Bakchar Horizon, which contains petrographic evidence for the presence of $^{13}C$-depleted $CH_4$ during ooid precipitation (Supplementary Discussion and Supplementary Figs. 33 and 34). The fact that this formation yields lower $\delta^{13}C$ values than those in immediately older and younger time bins indicates incorporation of $CH_4$-derived carbon and supports the notion that Fe-OC captures in situ signals. This notion is further strengthened by the fact that goethite and haematite loadings are statistically identical in the Ordovician, in which both mineralogies overlap (Supplementary Discussion). In general, we observe lower Fe-OC loadings and $\delta^{13}C$ values in the Proterozoic relative to the Phanerozoic.

Fe-OC loading and $\delta^{13}C$ variability never exceed $\pm 0.11$ wt% and $\pm 2.2$‰ ($\pm 1\sigma$; reduces to $\pm 1.7$‰ when separating Ordovician samples by mineralogy; Supplementary Fig. 125). Combined with the observation that modern [DOC] varies globally by only $\pm 58\%$ in fully saline coastal waters[47], such spatial homogeneity supports our interpretation that iron ooid Fe-OC captures globally relevant continental shelf signals. Our final record contains 100 individual iron ooid samples extracted from 26 geologic formations deposited since about 1,650 Ma, binned into 100 Myr periods.

We observe several notable secular trends (Fig. 2). First, time-binned average Fe-OC loadings exhibit more than 10-fold variability, from a minimum of $0.03 \pm 0.001$ wt% (Devonian haematites; $n = 3$) to a maximum of $0.49 \pm 0.19$ wt% (Jurassic goethites; $n = 16$). Moreover, goethite and haematite loadings are statistically identical in the Ordovician, in which both mineralogies overlap (goethite: $0.21 \pm 0.02$ wt%, $n = 7$; haematite: $0.26 \pm 0.10$ wt%, $n = 5$; two-tailed $t$-test $P < 0.05$), further supporting the notion that iron ooid Fe-OC captures globally relevant DOC signals. Temporally, Fe-OC loadings average about 0.1 wt% in the late Palaeoproterozoic to Mesoproterozoic, then decrease by a factor of around 2 in the Neoproterozoic before sharply rising in the Palaeozoic, in which they remain $\geq 0.2$ wt% until the modern day.

Second, time-binned average Fe-OC $\delta^{13}C$ values show >10‰ variability, from a minimum of $-28.5 \pm 2.7$‰ VPDB (Cambrian haematites; $n = 6$) to a maximum of $-18.1 \pm 1.4$‰ VPDB (actively forming modern goethites; $n = 4$). Unlike for loadings, goethite and haematite $\delta^{13}C$ values

are similar but not statistically identical in temporally overlapping Ordovician records (goethite: $-25.2 \pm 1.5$‰ VPDB, $n = 7$; haematite: $-28.7 \pm 0.8$‰ VPDB, $n = 5$; two-tailed $t$-test $P > 0.05$), suggesting a small offset between mineralogies. Still, we observe a marked $\delta^{13}C$ decrease in the Palaeocene–Eocene Thermal Maximum (PETM)-aged Bakchar Horizon relative to immediately older and younger formations. Because this Horizon also exhibits petrographical evidence for the presence of $^{13}C$-depleted $CH_4$ during ooid formation (Supplementary Discussion and Supplementary Figs. 33 and 34), we interpret this result as confirming that Fe-OC captures in situ signals. Temporally, $\delta^{13}C$ averages about $-27$‰ VPDB in the Proterozoic, then rises throughout the Phanerozoic (excluding the Bakchar Horizon) until reaching the modern-day values of $-18.1 \pm 1.4$‰ VPDB. However, limited temporal resolution prevents us from concluding whether this rise is gradual or represents a step change.

## [DOC] and $\delta^{13}C_{DOC}$ through geologic time

We developed a Monte Carlo model using our experimental calibration curves to reconstruct continental shelf [DOC] (termed [DOC]*, relative to modern) and $\delta^{13}C_{DOC}$ from measured Fe-OC signals, including propagated error (Supplementary Discussion). Importantly, because iron ooids capture both labile and recalcitrant DOC and may be contaminated by trace incorporation of POC, reconstructed [DOC]* should be treated as a maximum value and not equal to that of the deep ocean. Given the observed Fe-OC loading dependency on methoxy/carboxyl richness[31,33] and thus $S_R$ (Supplementary Fig. 23), we first constrain DOC molecular composition. Our model thus requires one free input parameter—the fractional contribution of each DOC end member through time ($f^i(t)$, $i = $ M-DOC, C-DOC or FA). We consider six scenarios (Supplementary Discussion and Supplementary Fig. 126): (1)–(3) all DOC derived from a single end member, to test the possible solution range; (4) unconstrained $f^i(t)$ (minimizing propagated error), to find a purely inverse solution; (5) rise of algae, following the canonical hypothesis that prokaryotes (represented by C-DOC) dominated primary production until about 780 Ma, after which eukaryotes (represented by M-DOC) came to dominate[23,48]; and (6) Proterozoic active eukaryotes, following a recently proposed scenario in which eukaryotic algae have comprised at least half of primary production through much of the Proterozoic[49,50].

Solving our model also requires five simplifying assumptions (Supplementary Discussion): (1) DOC is always a mixture of M-DOC, C-DOC and FA (that is, ignoring additional, potentially compositionally unique sources); (2) iron ooids represent primary or early secondary precipitates that form abiotically in communication with seawater (as evidenced by modern ooids); (3) Fe-OC signals reflect a weighted average of those derived from each end member; (4) Fe(II) flux to the site of ooid formation is constant through time despite large changes in open-ocean dissolved iron concentration[24,51]; and (5) all end members exhibit the same $\delta^{13}C$ value at a given point in time (as evidenced by spatial homogeneity across modern marine ecosystems)[52,53]. We tested each assumption for each $f^i(t)$ scenario; resulting uncertainty could lead to about 2× bias in [DOC]* and about 10‰ bias in $\delta^{13}C_{DOC}$ (Supplementary Discussion and Supplementary Figs. 127–130). Still, all temporal trends are qualitatively robust to assumption-induced bias.

Figure 3 shows our reconstructed DOC records. For both the rise of algae and Proterozoic active eukaryotes scenarios, [DOC]* was near modern levels in the late Palaeoproterozoic to Mesoproterozoic, decreased by about 90–99% in the Neoproterozoic, and returned to near-modern levels in the Palaeozoic, in which it has remained until today (Fig. 3b). Similarly, $\delta^{13}C_{DOC}$ for both scenarios was about 20‰ lower than today for much of the Proterozoic before rising to modern values starting in the Palaeozoic (Fig. 3c). Like for measured Fe-OC $\delta^{13}C$, Ordovician $\delta^{13}C_{DOC}$ reconstructed using goethite compared with haematite ooids are similar but not statistically identical (two-tailed

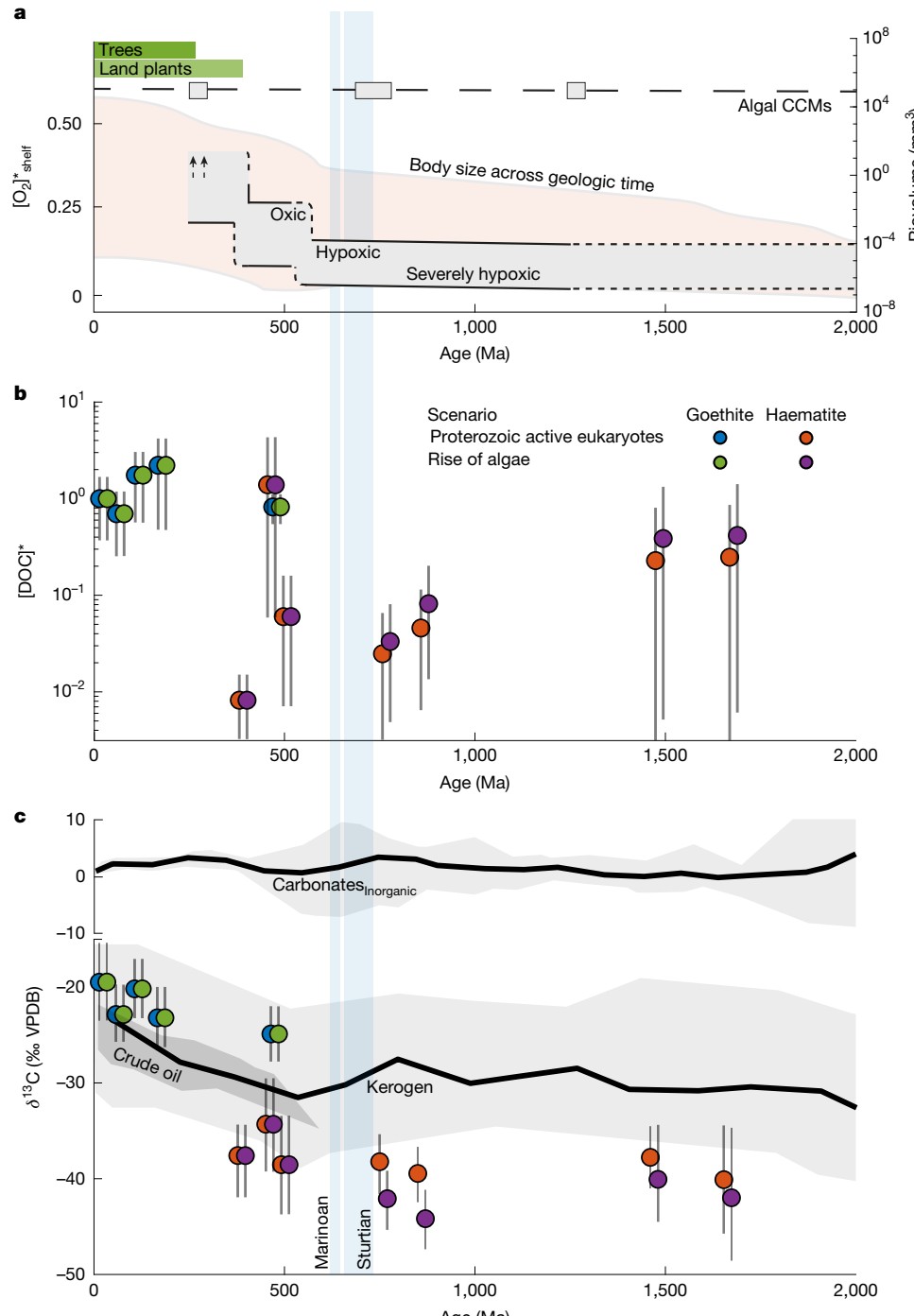

**Fig. 3 | Reconstructed DOC concentration and δ¹³C records. a**, Relevant Earth-system evolution and innovations, including predicted [O₂] in continental shelves relative to modern ($[O_2]^*_{shelf}$, fractional; grey-shaded region)[9]; range of body sizes for archaea, bacteria, algae and protozoa from the fossil record (biovolume, mm³; orange-shaded region)[8]; timing of Neoproterozoic snowball Earth events (vertical blue bars)[17]; proposed timing of algal carbon-concentrating mechanism (CCM) evolution (grey boxes with dashed line)[59]; and timing of land-plant and tree evolution (horizontal green bars)[58]. **b,c**, Predicted [DOC] relative to modern ([DOC]*; fractional) (**b**) and $\delta^{13}C_{DOC}$ (‰ VPDB) (**c**) since the late Palaeoproterozoic. Results are separated by mineralogy and microorganism evolution scenario (rise of algae: green represents goethite and purple represents haematite; Proterozoic active eukaryotes: blue represents goethite and orange represents haematite) and staggered for visual clarity. Opaque markers and vertical black lines are 100 Ma time-binned averages and 95% confidence intervals as predicted by our Monte Carlo model. The grey-shaded regions and thick black lines in **c** are compiled δ¹³C ranges and 100 Ma running averages for carbonates[56], kerogen and crude oil (note broken y-axis). See Supplementary Discussion for model derivation, implementation and crude oil and kerogen compilation.

t-test $P > 0.05$). Nevertheless, both reconstructions are near the range of compiled crude oil and kerogen values at this time (Fig. 3c, Supplementary Discussion and Supplementary Fig. 131), strengthening their interpretation as global signals and suggesting a common value across all well-constrained, time-equivalent organic reservoirs. Moreover, the

Devonian-aged Presles fraction modern (Fm) predicts lower [DOC]* and $\delta^{13}C_{DOC}$ relative to immediately older and younger formations, despite a lack of obvious petrographic evidence for exclusion (Supplementary Discussion and Supplementary Figs. S87 and S89). We, therefore, cannot determine whether this represents contamination by accessory

mineral phases, a local signal or a transient return to Neoproterozoic conditions.

## Implications for Earth-system evolution

Our data-based [DOC]* record differs from all previous models, which either predicted [DOC]* ≫ 1 in the Neoproterozoic[6,18–22] or little [DOC]* change through time[7]. We instead propose that [DOC]* reflects three distinct evolutionary periods in the history of Earth.

First, the late Palaeoproterozoic to Mesoproterozoic contained severely hypoxic continental shelves and deep oceans[9,24] and marine communities—whether mainly prokaryotic or eukaryotic[23,48–50]—dominated by small, single-celled organisms[8] (Fig. 3a). Slow particle sinking would strengthen the microbial loop, prompting high surface-water [DOC], whereas severely hypoxic deep oceans would restrict respiration in downwelled DOC-rich water, as predicted in ref. 6. However, observations indicate that this leads to a DOC reservoir size comparable to the modern ocean, not about $10^2$–$10^3$ times modern as originally proposed.

Second, the Neoproterozoic saw evolution of larger cells, colonial prokaryotes and complex multicellular eukaryotes[8,54] yet retained (severely) hypoxic continental shelves and deep oceans[9] (Fig. 3a). Faster particle sinking would weaken the microbial loop and drive higher export flux, but the microbial carbon pump would remain weak because of limited particle solubilization in hypoxic deep oceans[1,4,13]. This mechanism is consistent with an observed orders-of-magnitude increase in sedimentary POC accumulation rates at this time[55]. Increased export flux—possibly combined with increased scavenging onto POC[1]—would greatly lower [DOC], as observed. Furthermore, observed crude oil, kerogen and DOC $\delta^{13}C$ trends—when combined with constant $\delta^{13}C_{carb}$ (binned into 100 Ma periods to dampen transient excursions; Fig. 3c)[56]—either imply that photosynthetic organisms exhibited a larger $\varepsilon_p$ and/or that organic carbon reservoirs contained higher proportions of $^{13}C$-depleted compound classes (for example, lipids) or higher proportions of material derived from $^{13}C$-depleted sources (for example, methane) in the Proterozoic.

Third, the Phanerozoic is described by fully oxygenated deep oceans[9] and continued growth of marine ecosystem complexity and organism size—including grazing, biomineralization and the rise of sponges[8,57] (Fig. 3a). Although the microbial loop would remain weak relative to the Palaeoproterozoic to Mesoproterozoic, deep-ocean oxygenation and accompanying oxic respiration would increase particle solubilization and strengthen the microbial carbon pump, as proposed in ref. 7. Furthermore, although continental DOC runoff due to land-plant evolution could elevate [DOC]*, our record rises sharply in the Ordovician (488–444 Ma), about 55 Myr before the first land plants[58] (Fig. 3a). Reconstructed $\delta^{13}C_{DOC}$ values additionally argue against this explanation. We instead conclude that [DOC]* increased—rather than decreased—in response to Palaeozoic deep-ocean oxygenation and that a large DOC reservoir cannot be invoked to explain Neoproterozoic snowball Earth events[21] or $\delta^{13}C$ excursions[6,18,20].

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

## Methods

### Preparation of DOC for co-precipitation experiments

**Modern-marine analogue DOC.** Performing our co-precipitation experiments with modern marine DOC would require the collection of unreasonable water volumes (several thousand litres at typical surface ocean DOC concentrations). We, therefore, instead used large-batch microcosm cultures containing a mixture of marine primary producers and heterotrophic bacteria to generate modern-marine analogue DOC (termed M-DOC). Two diatom strains, *Phaeodactylum tricornutum* CCMP2561 and *Thalassiosira pseudonana* CCMP3367 (Bigelow NCMA), were allowed to grow for 4 days in 2 × 300 ml of L1 medium (Bigelow NCMA) placed in an AlgaeTron growth chamber. These cultures were then mixed in a 1:1 ratio, diluted into 2 × 20 l carboys filled with L1 medium and further grown at 14 °C with a light–dark cycle of 12–12 h. The cultures were stirred with a magnet bar (300 rpm) and sparged with filter-sterilized laboratory air for the entire duration. Five days post inoculation, 1 ml of a mixture of 96 uncharacterized isolated marine bacterial strains (provided by the Cordero Laboratory, MIT) was added to each culture. The cultures were grown for an additional 17 days (for a total of 26 days), reaching late stationary phase. The biomass was harvested by centrifugation at 7,000 RCF (relative centrifugal force) for 10 min at 14 °C (Eppendorf, CR30NX) and stored at −20 °C until further use. The total biomass yield was about 2 mg ml$^{-1}$ (wet weight). The biomass was then thawed, diluted into 18 M$\Omega$ cm$^{-1}$ double-distilled water (Milli-Q; termed DDW) and sonicated using a sonicator probe (Hielscher Ultrasonics) to lyse cells and release dissolved compounds. The lysed biomass was then centrifuged at 7,000 RCF for 10 min at 14 °C and filtered through a 0.2-µm polyether sulfone filter (PES; MilliporeSigma). The filtered solution was evaporated using a rotary evaporator at 40 °C under vacuum to yield dry DOC. Finally, the dried DOC was stored in an anoxic chamber until further use.

**Cyanobacterial DOC.** Anoxically dried *Arthrospira platensis* and *Arthrospira maxima* powders (BioSamara GmbH, Nu³) were processed in an agate ball mill to obtain particle sizes smaller than 20 µm and were subsequently sieved and combined. The powders were then mixed with DDW at a concentration of 1 g l$^{-1}$ to generate a large-batch cyanobacteria DOC solution (termed C-DOC). This solution was shaken for 48 h on a bottle shaker at 100 rpm and was then centrifuged, filtered, dried and stored exactly as described for modern-marine analogue DOC, above. The product composition and purity were assured by the manufacturers; no supplementary validation was undertaken.

**Fulvic acid.** Similar to cyanobacterial DOC, commercially purchased (Mark Nature) fulvic acid powder was processed in an agate ball mill to obtain particle sizes smaller than 20 µm and was subsequently sieved. The powder purity was confirmed by Fourier transform infrared spectroscopy (FTIR). The powder was then mixed with DDW at a concentration of 1 g l$^{-1}$ to generate a large-batch fulvic acid DOC solution (termed FA). This solution was shaken for 48 h on a bottle shaker at 100 rpm and was then centrifuged, filtered, dried and stored exactly as described for modern-marine analogue DOC, above.

### Characterization of DOC for co-precipitation experiments

We used absorbance and fluorescence spectroscopy to understand the molecular compositions of DOC involved in our co-precipitation experiments. Fluorescence excitation-emission matrices (EEMs) were acquired using a Horiba Scientific Aqualog spectrometer (Horiba) with a 1-cm cuvette at room temperature. The EEMs spanned excitation wavelengths from 250 nm to 500 nm at intervals of 5 nm and emission wavelengths from 300 nm to 600 nm at intervals of 2 nm. The integration times ranged from 0.5 s to 5 s. To refine resulting EEMs, three corrections were implemented following standard practice:

(1) a lamp intensity correction[60]; (2) adjustments for inner filter effects[61]; and (3) normalization to Raman units (RU), defined as the measure of Raman scattering intensity in the sample[62].

Fluorescent dissolved organic matter (FDOM) content in RU was determined by summing the maximum intensities of fluorescent peaks, including peak A (Ex: 250 nm, Em: 450 nm), peak B (Ex: 270 nm, Em: 304 nm), peak C (Ex: 350 nm, Em: 450 nm), peak M (Ex: 320 nm, Em: 411 nm) and peak T (Ex: 290 nm, Em: 349 nm) (refs. 63,64). Relative contributions of each peak were defined as the ratio of the intensity of a given peak to the total FDOM content. Several absorbance and fluorescence metrics were also identified and used here. First, the fluorescence index (FI) was derived from the emission intensities at 470 nm and 520 nm on excitation at 370 nm (refs. 65,66). Second, the specific UV absorbance at 254 nm (SUVA$_{254}$) was calculated as the absorbance at 254 nm divided by the total DOC concentration in solution[67]. Third, spectral slopes were ascertained by fitting absorption data to the equation

$$a_\lambda = a_{\lambda_{ref}} e^{-S(\lambda - \lambda_{ref})}, \tag{1}$$

where $a_\lambda$ is the absorption at wavelength $\lambda$, $\lambda_{ref}$ is a reference wavelength and $S$ is the spectral slope; slopes were determined over two wavelength ranges: 275–295 nm ($S_{275-295}$) and 350–400 nm ($S_{350-400}$) (ref. 40). Finally, slope ratios were calculated as $S_R = S_{275-295}/S_{350-400}$ (ref. 40).

### Iron (oxyhydr)oxide-DOC co-precipitation experiments

**General steps relevant to all syntheses.** An iron (oxyhydr)oxide synthesis protocol was developed to understand the loading responses of co-precipitated Fe-OC to (1) crystalline iron (oxyhydr)oxide mineralogy (that is, goethite compared with haematite) and synthesis pH; (2) timing of DOC introduction (that is, before or after precursor synthesis); and (3) DOC molecular composition. Specific synthesis protocols are detailed below, and additional synthesis tests (for example, as a function of temperature, duration, dissolved silica concentration, Fe(III) source, absolute Fe(III) and DOC concentrations) are described in the Supplementary Discussion. Broadly, all syntheses involved the addition of Fe(III)-bearing salts [Fe(NO$_3$)$_3$ · 9H$_2$O or FeCl$_3$ · 6H$_2$O] to an (anaerobic when necessary) parent solution, followed by pH adjustment to form the poorly crystalline precursor ferrihydrite. To generate crystalline phases, ferrihydrite was then rinsed, pH was adjusted and solutions were allowed to ripen to goethite or haematite depending on the recipe.

In the protocol descriptions below, we designate the timing of DOC addition with '+DOC; $x$', where $x$ = 'pre' for pre-ferrihydrite DOC addition or $x$ = 'post' for post-ferrihydrite DOC addition. Furthermore, to understand whether co-precipitated Fe-OC responds to absolute DOC concentrations or to DOC/Fe(III) ratios, we performed two sets of experiments in which we independently adjusted initial parent-solution DOC or dissolved iron concentrations (while holding the other constant) for all synthesis protocols. Importantly, reagent masses appearing below pertain to the standard concentrations for each type of synthesis. These masses were adjusted accordingly to achieve the desired DOC/Fe(III) ratios. A list of all concentrations used for each experiment is reported in the Supplementary Data.

**Goethite syntheses. Acidic goethite synthesis.** Goethite was formed in an acidic solution using a modified version of the approach developed in ref. 68. First, 101.071 g of reagent-grade Fe(NO$_3$)$_3$ · 9H$_2$O (Sigma-Aldrich) was dissolved into 250 ml of trace-metal grade 1 M HCl (Sigma-Aldrich) in DDW to form a base solution, which was diluted to a total volume of 500 ml by adding 250 ml of DDW (+DOC; pre). Then, 250 ml of 2 M reagent-grade NaOH (Sigma-Aldrich) was added while maintaining vigorous stirring to increase the pH to about 1.5 and facilitate the formation of a stable brown sol of ferrihydrite. The bottles were then sealed, and ferrihydrite (+DOC; post) was allowed to ripen to

goethite, which would typically commence after about 50 days following the procedure described in ref. 68 (that is, at room temperature). To expedite this transformation, we transferred the reaction mixture to an oven held at 50 °C and allowed it to age for 5 days. After ageing, goethite products were centrifuged, rinsed at least three times with DDW to remove excess −OH, dried at 60 °C and stored until further analysis.

**Circumneutral goethite synthesis.** First, 2-line ferrihydrite was prepared by dissolving 8.0 g of $Fe(NO_3)_3 \cdot 9H_2O$ in 100 ml of DDW (+DOC; pre) followed by adjusting the solution pH to 7 by adding 1 M NaOH (ref. 42). The solution was vigorously stirred to ensure complete precipitation. Fresh ferrihydrite precipitates underwent a series of rinsing and washing cycles using deaerated DDW (that is, purged with 99.99% pure $N_2$; PanGas). Ferrihydrite was then purged with $N_2$ gas for 2 h to remove any $O_2$ present in solution. The deaerated suspension was transferred to an anaerobic glovebox ($O_2 < 1$ppm; Coy Laboratory Products), where ferrihydrite was aliquoted into glass bottles. Reagent-grade $Fe(NH_4)_2(SO_4)_2 \cdot 6H_2O$ (Sigma-Aldrich) was dissolved in a deaerated solution and titrated to pH 7 using deaerated 0.5 M NaOH. This solution (+DOC; post) was then introduced into glass bottles containing ferrihydrite to achieve a final ratio of $Fe(II)/Fe_{total} \approx 0.19$. Glass bottles were then vigorously stirred, sealed with butyl rubber stoppers, crimped, quickly brought to their designated ageing temperature of 20 °C, and allowed to age to goethite for 10–120 days. After ageing, goethite products were centrifuged, rinsed at least three times with DDW to remove excess OH, dried at 60 °C, and stored until further analysis. We were not able to stabilize synthesis pH using organic buffers (see ref. 42), as this would add additional organic carbon to the experimental system. Rather, we targeted lower goethite yields to maintain a pH range of 5–7 throughout the experiments.

**Alkaline goethite synthesis.** Goethite was formed in alkaline solution using a modified version of the approach developed in ref. 69. First, 13.515 g of reagent-grade $FeCl_3 \cdot 6H_2O$ (Sigma-Aldrich) was dissolved in 50 ml of DDW (+DOC; pre). Then, 90 ml of reagent-grade 5M KOH (Sigma-Aldrich) was added while maintaining vigorous stirring to increase pH to about 11 and facilitate the formation of red-brown two-line ferrihydrite. This solution was quickly diluted with DDW to reach a final volume of 1 l (+DOC; post). The bottles were then sealed, and ferrihydrite was allowed to ripen to goethite in an oven held at 70 °C for 7–10 days. After ageing, the goethite products were centrifuged, rinsed at least three times with DDW to remove excess −OH, dried at 60 °C and stored until further analysis. Highly alkaline conditions required the use of Teflon flasks to avoid Si dissolution that may occur in glass vessels.

**Haematite synthesis. Acidic haematite synthesis.** Haematite was formed in acidic conditions using a modified version of the approach described in ref. 45. First, 5.405 g of $FeCl_3 \cdot 6H_2O$ was dissolved in 1 l of 1 mM HCl in DDW (+DOC; pre). The bottles were then sealed, and the solution was allowed to react (+DOC; post) in an oven held at 90 °C for 14 days to form haematite. After ageing, haematite products were centrifuged, rinsed at least three times with DDW, dried at 60 °C and stored until further analysis.

**Circumneutral haematite synthesis.** Haematite was formed in circumneutral conditions using a modified version of the approach described in ref. 42, which is itself modified from that in ref. 45. First, solutions of DDW, NaOH and reagent-grade $NaHCO_3$ (Sigma-Aldrich) were prepared and deaerated (that is, purged with 99.99% pure $N_2$) for 2 h to remove any dissolved $O_2$. The solutions were then sealed and transferred to an anaerobic glovebox ($O_2 < 1$ ppm) and were individually pre-heated to the final ageing temperature to ensure consistency in initial reaction conditions. Within the glovebox, 13.40 g $FeCl_3 \cdot 6H_2O$ was added to 252 ml of DDW (+DOC; pre), then 150 ml of 1M NaOH followed by 34 ml of 1M $NaHCO_3$ were added to the solution while maintaining vigorous stirring to prevent aggregate formation. The bottles were then sealed (+DOC; post), rigorously shaken and allowed to react in an oven held at either 50 °C or 70 °C for 5–30 days

to form haematite (at lower temperatures, it is not possible to obtain mono-mineral haematite under laboratory timescales and circumneutral pH). After ageing, the haematite products were centrifuged, rinsed at least three times with DDW, dried at 60 °C and stored until further analysis. Several iron salt concentrations and compositions were tested before choosing the exact protocol described here (Supplementary Data).

### Characterization of precipitated and natural iron oxides
To assess the properties and compositional attributes of both precipitated and natural iron oxides, we used a multifaceted characterization approach as detailed below.

**Powder X-ray diffraction.** X-ray diffractograms were obtained using an Empyrean 3 diffractometer equipped with a Cu-Kα (1.541,84 Å) X-ray source and a PIXcel[3D] detector. Patterns for laboratory-precipitated haematite and goethite were acquired by step scanning from 10° to 50° $2\theta$ in 0.01° increments at a scan rate of 0.35° min$^{-1}$. For natural samples, the scanning range was from 10° to 80° $2\theta$ in 0.02° increments at a scan rate of 0.95° min$^{-1}$. The obtained X-ray diffractograms were automatically matched against the International Center for Diffraction Data database and the Crystallography Open Database.

**Scanning electron microscopy and energy-dispersive X-ray spectroscopy.** Scanning electron microscopy imaging and energy-dispersive X-ray spectroscopy elemental mapping were conducted using a Zeiss Sigma scanning electron microscope equipped with an QUANTAX energy-dispersive X-ray spectroscopy (Brucker). The measurements were performed at an operating voltage of 20 kV, using an aperture of 60 μm in analytical high current and vacuum mode.

**Section preparation.** Round, thick, polished sections were prepared by immersing samples in epoxy resin poured into a 1″ round plastic mould. To prevent sample displacement during immersion, the samples were adhered with a double-sided tape. The samples were allowed to dry for 24 h, after which the sections were polished to expose the maximum submerged sample portion and to achieve the desired smoothness (usually ≤1 μm). Thin sections were prepared by first adhering a sample to a glass slide using epoxy. The adhered samples were then ground to a thickness of about 30 μm while ensuring uniformity across the section. After grinding, each section was polished (usually to ≤1 μm) to remove any surface scratches and enhance clarity.

**Optical microscopy.** Analysis under plane-polarized light was carried out using a Zeiss Axio Scope A1 microscope. Thick and thin sections were observed at varying magnifications to identify the mineralogical phases and their spatial distributions, photographed and documented (see Supplementary Discussion section 'Geologic descriptions of sampled formations').

**Raman microspectroscopy.** We used Raman microspectroscopy (LabRAM HR Evolution, Horiba Scientific) to investigate the spatial distribution of organic matter in iron ooids. This system integrates a commercial light microscope (BxFM, Olympus) with an optical box for Raman functionality, key components of which include a diffraction grating (300 lines mm$^{-1}$), a detector (back-illuminated deep-depleted CCD) and a laser (532 nm, 10 mW). The microscope is equipped with objectives and a motorised XYZ-stage for microspectroscopic Raman interrogation of regions of interest within a given sample.

Ethanol pre-sonicated whole ooids (verified by light microscopy) were physically fixed within a cylindrical epoxy mould as described above, thus ensuring a flat surface and enabling measurement of Fe-OC spatial distribution both within and on ooid surfaces. Importantly, sections were polished using carbon-free aluminium oxide powder to prevent carbon contamination. To synchronize visual images of

regions of interest with their Raman spectra, light microscopy images were acquired before either one-dimensional raster line scanning or two-dimensional raster area mapping using Raman microspectroscopy (Fig. 1). Considering that sample surfaces contain some roughness (that is, ≤1 µm polishing roughness), a low magnification objective was used (either 10× or 50×; MPlanN or LMPlanFLN, respectively, Olympus). This provides a relatively large depth of focus and thus compensates for Raman signal variation due to surface roughness. Theoretical spatial resolution was calculated to be $1.22\lambda/NA = (1.22 \times 532)/0.25 = 2.6$ µm. Therefore, a step size of 8–15 µm was used for raster line scanning and area mapping. For each point measurement, the laser exposure time was 5 s, and the spectral window ranged from 50 cm$^{-1}$ to 3,100 cm$^{-1}$.

The presence or absence of Fe-OC in ooid samples was determined using a Raman peak at 1,590 cm$^{-1}$ (that is, the G-band)[70,71] (Fig. 1a). Because haematite exhibits a strong peak at 1,300 cm$^{-1}$ (that is, near the G-band), the placement of this Fe-OC peak makes the identification of Fe-OC in haematite feasible only if its concentration—and thus signal intensity—is strong[72]. In an attempt to circumvent this issue, we used other laser wavelengths for haematite ooids (that is, 660 nm and 785 nm), but these spectra did not provide better sensitivity. Owing to this high background from the 1,300 cm$^{-1}$ haematite peak, the inability to identify Fe-OC cannot be reliably used as evidence for an absence of Fe-OC in haematite ooids. Goethite spectra, in contrast, do not exhibit this behaviour. We, therefore, focus primarily on goethite mineralogy for all Raman line scans and maps.

Raman spectra were processed using software built in-house. A spectral window of 300–1,800 cm$^{-1}$, which covers Raman signals of goethite and organic matter, was chosen for data processing and analysis. Resulting spectra underwent smoothing (that is, de-noising) using a Savitzky–Golay filter (polynomial order 3 and window size 5) and baseline subtraction using a first- or third-order polynomial algorithm. The presence of Fe-OC was determined by calculating the ratio $I_{1,450-1,700}/I_{1,200-1,450}$, where $I$ represents integrated intensity within the defined spectral window. These spectral windows were chosen here to represent the range containing the diagnostic Fe-OC peak (that is, 1,450–1,700 cm$^{-1}$) and that of a background peak that is insensitive to the presence or absence of Fe-OC (that is, 1,200–1,450 cm$^{-1}$). Using this ratio, a threshold value was selected based on visual inspection of the raw data to convert the calculated values to a binary format, indicating the presence or absence of organic matter (1 or 0, respectively). The threshold values are sensitive to sample-specific characteristics (for example, surface roughness) and are thus recalculated for each line scan or map (for example, Panarea Island threshold = 0.600; Aseri Fm threshold = 0.715; Sillaoru Fm threshold = 0.975; Fig. 1). Although variable and based on visual data inspection, these thresholds are chosen to be conservative (that is, equal to or greater than the maximum measured intensity ratio of line scan or map pixels that are known to be free of Fe-OC, for example of epoxy resin or silicate/carbonate cements).

**X-ray fluorescence.** To ascertain the elemental composition of extracted goethite and haematite ooids, the samples were ground to powder form and subsequently analysed using a Niton XL5 Plus XRF (Thermo Scientific). Each powdered sample was analysed three times, and the powder was mixed in the measuring cup between runs to ensure signals were representative.

### Extraction and analysis of ooid-bound DNA
Microbial community compositions of DNA contained in modern ooids from Panarea Island, Italy[44], were determined to assess the potential impacts of iron-oxidizing biofilms. Specifically, ooids were hand-picked and split into two groups that were either subjected to (1) sonication in pure ethanol using a sonicator probe to remove surface DNA contamination (that is, analysis of internal DNA only) or

(2) no cleaning procedure (that is, analysis of surface-bound and internal DNA). Both groups were then ground within a laminar flow hood (that is, to prevent biological contamination) to expose internal DNA and maximize extraction efficiency. DNA extractions were then performed on both fractions (3 g per extraction, triplicate extractions per group) using a phenol-chloroform-based approach[73]. Quantification using a Qubit 3.0 fluorometer and the dsDNA HS Assay (Thermo Fisher Scientific) showed that sonicated ooids yielded DNA concentrations of 0.171 ng µl$^{-1}$, 0.238 ng µl$^{-1}$ and 0.211 ng µl$^{-1}$ ($n = 3$ extractions), whereas non-sonicated ooids yielded higher concentrations of 0.442 ng µl$^{-1}$, 0.457 ng µl$^{-1}$ and 0.441 ng µl$^{-1}$ ($n = 3$ extractions); this supports the hypothesis that sonication effectively removes surface-bound DNA.

All extracts were then purified with AMPure XP beads (Beckman Coulter). To perform amplicon sequencing of the microbial 16S rRNA gene, sequencing libraries were prepared following a two-step PCR approach. In the first step, bacterial 16S PCRs were performed using Illumina-adaptor primer pair 515F (GTGYCAGCMGCCGCGGTAA) and 926R (CCGYCAATTYMTTTRAGTTT) and KAPA HiFi HotStart ReadyMix PCR kit (Roche Molecular Systems), targeting the V4–V5 hypervariable regions of the 16S rRNA gene (Supplementary Tables 4 and 5). Products of the first-step PCR were then submitted to Microsynth AG (Balgach, Switzerland) for preparation of Illumina Nextera barcoded second-step PCR libraries and sequencing on Illumina NovaSeq (2 × 250 cycles) (Illumina).

Unfortunately, the quantity and quality of most extracts were insufficient for reliable sequencing results, leading to only one successful analysis of a single non-sonicated extract and no successful analyses of any sonicated extracts. The results are thus interpreted to reflect a combination of surface-bound and internal DNA. For this successful extract, 115,236 reads were obtained after sequencing. Adaptor sequences were removed using cutadapt v.3.4 (ref. 74). Sequence analysis was then performed in R v.4.3.1 using DADA2 (v.1.30) (ref. 75) for quality filtering and amplicon sequence variant (ASV) assignment according to published protocols. As low microbial biomass samples are prone to being contaminated by trace amounts of DNA contaminants from extraction reagents, many taxa (for example, *Comamonadaceae*, *Sporichthyaceae*, *Sphingomonadaceae* and *Enterobacteriaceae*) were removed based on what was detected in the negative control sample (containing extraction reagents only). Taxonomy was assigned using the SILVA reference database v.138.1 using IDTAXA (threshold = 60) (ref. 76) and DECIPHER v.2.3 (ref. 77).

### Removal of ferrihydrite-bound and adsorbed OC
After each synthesis, the products were divided into three fractions and subjected to a washing procedure to step-wise remove OC bound to poorly crystalline phases and OC adsorbed to crystalline mineral surfaces. These fractions were treated as follows: First, one fraction was rinsed three times with DDW, dried at 60 °C and stored until further analysis (termed raw). Second, one fraction was treated with a 1 M HCl solution at room temperature for about 1 h to remove residual untransformed ferrihydrite or other poorly crystalline phases (termed −Fh/PC) (ref. 45). This fraction was then washed, dried and stored as described for the raw fraction. Finally, one fraction was treated with 1 M HCl solution as described for −Fh/PC and was subsequently treated with a 1 M MgCl$_2$ solution at room temperature for about 1 h to chelate and remove any loosely bound (adsorbed) Fe-OC complexes on the crystalline mineral surfaces (termed −Fh/PC− ads. OC) (ref. 78). Each fraction was assessed by XRD to ensure that each step does not influence crystalline iron oxide mineralogy.

### Preparation and extraction of geologic iron ooids
Bulk ooidal ironstone samples were initially subjected to gentle crushing using a hydraulic press. On fragmentation, individual ooids were selected from the debris on an aluminium foil using tweezers. For more efficient separation, a strong neodymium hand magnet was used to

enhance the ooid yield. Following this, a light microscope inspection ensured the inclusion of only intact ooids (that is, broken ooids were not considered).

Whole ooids subsequently underwent a washing cycle involving three DDW rinses. They were then exposed to a 1 M trace-grade HCl (Sigma-Aldrich) solution for 1 h at ambient conditions to eliminate any poorly crystalline phases and potential carbonates, both inherent to the rock matrix and within the ooids themselves. After acid treatment, ooids were immersed in a trace-metal grade 1 M MgCl$_2$ solution (Sigma-Aldrich) for 1 h to chelate and remove any loosely bound Fe-OC complexes. The ooids were then rinsed three times with DDW and air-dried in an oven at 60 °C and pulverized using an agate mortar and pestle until a fine powder was obtained. The preliminary tests verified that the resulting powder granularity was finer than 20 μm. Finally, the powdered samples were resubjected to the HCl and MgCl$_2$ treatments described above to ensure that the final products are free of carbonates, poorly crystalline phases and adsorbed OC (that is, −Fh/PC− ads. OC).

### Analytical methods for measuring Fe-OC content and isotope compositions

Organic carbon contents, reported as wt% OC, and isotope compositions, reported as δ[13]C in ‰ relative to Vienna Pee Dee Belemnite (VPDB), of iron (oxyhydr)oxides were quantified using a Thermo Scientific Flash Elemental Analyser coupled to a Thermo Scientific Delta V Isotope Ratio Mass Spectrometer (EA-IRMS). Samples were combusted in the presence of O$_2$ in an oxidation column at 1,020 °C. Combustion gases passed through a reduction column at 650 °C, in which the N$_2$ and CO$_2$ gases produced were separated chromatographically and transferred to the IRMS by an open split for online isotope measurements.

For calibration, approximately 30–60 mg of each protocol-treated iron (oxyhydr)oxide sample was aliquoted into triplicate tin capsules. These were crimped and loaded into a helium-flushed autosampler. Both wt% OC and δ[13]C values were calibrated using in-house standards, including Atropina (δ[13]C = −21.4‰, 70.56% carbon; Thermo Fisher), Nicotinamid (δ[13]C = −42.2‰, 59% carbon; Thermo Fisher), Pepton (δ[13]C = −15.6‰, 44% carbon; Sigma-Aldrich) and Bodenstand 5 (0.141% carbon; HEKAtech). These standards were analysed between about every 10 samples. Furthermore, δ[13]C was calibrated using the international standards NBS22 (δ[13]C = −30.03‰) and IAEA CH-6 (δ[13]C = −10.46‰). Uncertainty was determined as the standard deviation (±1σ) of triplicate measurements; system reproducibility is typically better than 0.2‰ for δ[13]C.

To estimate ooid formation timescales, actively forming modern iron ooids were additionally analysed for [14]C activities—reported as Fm relative to 95% of the [14]C activity of the NBS Oxalic Acid I standard in 1950 and corrected to a δ[13]C value of −25‰ VPDB[79]. Specifically, [14]C activities were measured using an Elementar PyroCube EA connected to an IonPlus mini radiocarbon dating system accelerator mass spectrometer (EA-AMS) operated using a gas ion source[80,81]. Similar to EA-IRMS analysis, about 30–60 mg of each iron (oxyhydr)oxide sample was aliquoted into tin capsules, which were crimped and loaded into a helium-flushed autosampler. Oxalic acid II NIST SRM 4990C standard was measured for fractionation correction and standard normalization. The background of the EA was assessed with phthalic anhydride (PHA, Sigma, PN-320064-500g, LN-MKBH1376V), which was weighed into the same capsules used for the samples and therefore included the respective possible contamination. Uncertainty was determined as the standard deviation (±1σ) of propagated blank correction and analytical error from [14]C counting statistics. Fm values were additionally converted to uncalibrated [14]C ages as

$$^{14}C_{age} = -8033 \times \ln(Fm), \tag{2}$$

reported in units of [14]C yr BP (ref. 82).

### Data compilation

To compare our Fe-OC δ[13]C record to other OC records, we compiled literature δ[13]C values for (1) kerogen, (2) crude oil and (3) carbonate-associated OC. All compiled data, including literature references for each data point, are reported in the Supplementary Data. Compilation details are described below:

**Kerogen.** Kerogen OC δ[13]C values over the period 0–2,500 Ma were compiled from refs. 83–111. This time range was chosen as it broadly encompasses our Fe-OC δ[13]C record as well as the Great Oxygenation Event (GOE), which would be expected to affect carbon cycle and thus OC δ[13]C values. All literature data that are explicitly stated to be kerogen and contain a reported depositional age were included in our compilation. Some datasets exhibit a positive correlation between measured OC content and δ[13]C value, indicating incomplete carbonate removal during analysis. For these datasets, we only include the δ[13]C value from the sample with the lowest OC content. Moreover, we specifically focused on samples characterized by conditions below the greenschist facies. This criterion was applied to exclude any potential isotopic alterations that might occur under higher-grade metamorphic influences.

**Crude oil.** Crude oil δ[13]C values over the period about 0–600 Ma (that is, the oldest reliable age reported in the literature) were taken from the compilation of ref. 112 and supplemented with data from refs. 90,98,113–124. Unlike for kerogen, more care must be taken when reporting crude oil ages because depositional age can differ from host-rock age due to oil migration. We, therefore, considered only literature data that explicitly reported oil depositional age, and we omitted those data points that reported only host-rock age (that is, with unknown oil depositional age). No further filtering was implemented.

**Carbonate-associated OC.** Finally, carbonate-associated OC δ[13]C values over the period 0–2,500 Ma were taken from the filtered compilation of ref. 56. The authors of ref. 56 excluded sediments from non-marine, authigenic and heavily metamorphosed settings in their filtering procedure. We adopted this filtered dataset without modification or addition.

### Computational methods

All model calculations (see Supplementary Discussion section 'Estimating DOC concentrations and δ[13]C values from iron ooid Fe-OC') were written in MATLAB and carried out on the ETH Euler scientific computing cluster (https://ethz.ch/de/news-und-veranstaltungen/eth-news/news/2014/05/euler-mehr-power-fuer-die-forschung.html).

### Data availability

All data supporting the findings of this study are publicly available at Zenodo[125] (https://doi.org/10.5281/zenodo.15849728). This includes all data necessary to reproduce the results. Sequence data are deposited in the European Nucleotide Archive (ENA) under accession number PRJEB90806.

### Code availability

The codes used to process data, generate results and make MC figures (written in MATLAB) are available at Zenodo[125] (https://doi.org/10.5281/zenodo.15849728).

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

**Acknowledgements** We thank E. Georgiadis, S. Bishop, L. Falk, G. Fellin, R. Wijker and L. Zehnder for laboratory assistance and support. This result is part of a project that has received funding from the European Research Council (ERC) under the Horizon 2020 research and innovation programme of the European Union (grant agreement no. 946150, to J.D.H.). Additional funding was provided by the Swiss National Science Foundation (grant no. 207786 to J.D.H. and grant no. CRSK-2_220718, to N.G.). K.S.L. acknowledges support from the Korea Basic Science Institute (National Research Facilities and Equipment Center) grant funded by the Korea Government (MSIT) (no. RS2025-00554860) and the 2025 Research Fund (1.2500XX.01) of UNIST.

**Author contributions** N.G. and J.D.H. conceived the study; N.G. and U.A. performed synthesis experiments; M.D.B., F.I., M.A.L., S.M.P., M.R. and S.W. provided geologic samples; R.G.M.S. generated DOC molecular composition data; N.G., C.M. and Z.W. generated and interpreted 16S rRNA data; N.G., I.K.-A. K.S.L. and R.S. generated petrographic data (light microscopy, XRD, electron microscopy, energy-dispersive X-ray spectroscopy and Raman spectroscopy);

N.G., S.M.B., G.A., T.M.B., N.H. and M.J. generated Fe-OC content and isotope ($^{13}$C, $^{14}$C) data; N.G., A.N. and J.D.H. developed the model; N.G. and J.D.H. interpreted the results; N.G. and J.D.H. wrote the paper with input from all authors.

**Funding** Open access funding provided by Swiss Federal Institute of Technology Zurich.

**Competing interests** The authors declare no competing interests.

**Additional information**

**Correspondence and requests for materials** should be addressed to Nir Galili.

