## [Peer Review File · Nature]

The geologic history of marine dissolved organic carbon from iron oxides

Corresponding Author: Dr Nir Galili

Version 0:

Reviewer comments:

Referee #1

(Remarks to the Author)

Dissolved organic carbon (DOC) in the world's ocean's is an important reservoir for the modern carbon cycle, and there are longstanding questions about its sources, sinks, and dynamics. In addition to this a number of popular ideas have arisen over the past two decades about how this reservoir may have changed over Earth history. Some of these ideas even suggest that DOC dominated major aspects of carbon cycling during climate perturbations in the past, particularly during Neoproterozoic time. However despite their popularity, these ideas were established entirely from simple, largely unconstrained models, and remain untested by observations. In this work the authors describe the concept behind and application of a geological proxy for reconstructing the concentrations and carbon isotopic compositions of DOC in marine basins from Proterozoic to present using observations of an odd, anachronistic sedimentary facies: iron ooids. This study describes the first serious effort to reconstruct the size of this reservoir through time. It is a hard problem. Though the work is nascent and there are many challenges/issues/questions about the approach and method, it is innovative work on an important problem and is the right kind of study for the journal under consideration.

The authors further provide and provocative interpretations of the data, with potential explanations in the context of major changes in the biological and redox histories of the global oceans and atmosphere. There are several areas in which I think the manuscript currently requires revision and perhaps some realignment of the discussion and implications to remain closer to DOC and avoid what I think are really speculative statements about the secular evolution of the global carbon cycle (particularly considerations about global isotope mass balance). I have highlighted these below. I think there are also a couple areas where the study would benefit so substantially from additional measurements that I enumerated them here: [1) measurements of the DOC/DOM in the waters from which the modern ooids form; 2) imaging of organics within the ooids to test whether or not there are concentrated residues of biofilms or cells; 3) extraction, abundance, and sequencing of microbial DNA within the modern ooids to assess probability of local organic sources to the samples.].

The main text requires a paragraph or two on the mechanisms that are responsible for generating the minerals and textures of iron ooids. This is a super exotic sediment type and there are many questions about process and mechanism. There are also questions about varied mineralogy over time, not just iron oxides, but mixed valence silicates and sulfides. We need a somewhat comprehensive synthesis and critique of the mechanisms of formation of the minerals being surveyed. Thankfully there are highly limited examples of where these materials are forming in the modern. Current best hypotheses for these suggest at least in the modern cases the iron sourced from deep hydrothermally influenced porewater that mixes with shallow marine porewater to oxidize and precipitate the iron on the grains as a cement and then you have wave energy and sediment transport to round and further modify the textures. Some models consider microbial catalysis of the iron oxidation. Some don't. This is important to put up front for the reader. It is also important to note that just the prevalence of this litho type is widely recognized to change with time. So too might mechanisms of its formation. The rare modern examples are proximal to volcanic sources of iron-rich fluids. There are many historical examples that appear to be on passive marine shelves and not anywhere near such an iron source. So the mechanisms of formation might change through time. Could be other

minerals even. I appreciate that there is a section with some of this information in the SI, but there's not enough in the main text for a general reader to understand and follow. I appreciate the substantial amount of information in the supplement. But you have to wait until page 166 to find it.

With access to these modern examples as the authors have done nicely here, there are two additional observations that would be important for convincing readers that such a proxy is valuable. I really like the experimental work here to look at DOC incorporation into iron oxides under a range of plausible conditions. But these experiments don't produce anything texturally like the iron ooids (and are even missing some of the phases like silica). So it would greatly improve the study to include a direct comparison of the DOC concentrations and isotopes from the same waters from which the modern ooids are derived. They will hopefully show a relationship expected from the experiments and that would help provide support for the proxy.

Additionally the authors go to some length in the SI to rule out local microbial sources for organics that might be contained within the iron oxides phases in the sediment. These arguments are currently quite weak and could be made far stronger with two different approaches. 1) using an imaging approach to document the textures in which, to rule out the presence of biofilms, communities, cells. Iron oxides can be a bit tricky to work with for many imaging techniques due to their opacity, but disaggregation and fluorescence microscopy can work quite well as the work of Clara Chan and Dave Emerson have shown for iron-oxidizing microbes. But reflectance spectroscopy and SIMS can work quite well for in situ techniques. 2) environmental sequencing works so well even in these systems that it is very straightforward at this point to extract DNA, quantify it (e.g. via Q-bit), and sequence it either with or without amplification to get a census of the microbes present and assess any potential role they might have in the genesis of the iron oxides. The supposition that microbes are absent or unimportant is poorly supported at present. And recall that most iron oxidizers are founding autotrophs in the community and fix carbon with the same biochemistry that the photoautotrophs do in the ocean E.g. Kennedy et al. 2003 *Geomicrobiol*, Edwards et al. 2011 *ISME J*, Mori et al. 2017 *ISME J*, Zhong et al. 2022 *Microbiol Spectr*. Many properties of the organic matter derived from these communities are highly similar to expectations of marine DOC. Would be better to just directly test this. The proxy and conclusions about past marine DOC will be much stronger for it.

Finally, the last part of the paper discussions about isotope fractionation global carbon cycle appears a bit weak to me. It requires many assumptions and strays pretty far from the findings here. Highly speculative. Suggest removing this and just keeping the discussion close to the observations and results here. That's already a nice body of work and will come across stronger to the reader focused on DOC.

Comments arising from the text, in line.

25 - ecosystem interactions would including processes like 'sloppy feeding' right? So not just phage attack, lysis, and microbial loop? Suggest clarifying.

34 - connection to CO₂ here is very indirect, and indeed still debate about sign, suggest omitting in this context.

51 - too strong. The concept of high DOC through the latter Neoproterozoic is interesting, but important to make clear that its model derived. And explanations for carbon isotope anomalies in the Neoproterozoic, have trended toward ideas about carbonate synthesis and preservation given mineralogical observations and patterns in Ca and Mg isotope fractionation (e.g. Ahm et al. 2021 *EPSL*). Suggest removing as it's too much of a straw man. 'Fundamental requirement' needs observations to support.

61 - appreciation the nuance in this section, but seems more important to rewrite with the above paragraph and make clear that these are all ideas concerning DOC through time that are model outcomes. Indeed we don't have accurate mechanistic models that can reproduce the dynamics of marine DOC today, so no surprise that the models don't agree on the past. What is needed is observations and data that can inform whether or not DOC changes across Earth history. That's really what the paper is about.

71 - suggest toning this down a bit. Formally an anachronism, we still have many questions about how they form, and indeed there may not be a common genetic mechanism (e.g. Young 1989, Salama et al. 2013, Matheson and Pufahl 2021, Di Bella et al. 2019). Because they don't typically form in widespread marine environments (outside rare volcanogenic environments) today it is challenging to draw conclusions about how they formed across marine shelves at times in the past with much certainty. Would be useful to modify this paragraph to instead have a small discussion of the various different minerals that are potentially important, and how you're examining one potential mechanism for consideration of the use of iron oolites as a potential marine DIC proxy. Addendum: I see much later in the SI that there is a section dedicated to this. Suggest revising that heavily and then bringing a long paragraph with the discussion and caveats into the main text.

76 - need some discussion here about what is meant by Fe(III). Need some discussion of precipitation mechanisms in mind. In the experiments it is supplied to the solution as a solid ferric chloride, and that seems pretty different from working hypotheses about the formation of iron ooids wherein the iron is supplied as ferrous soluble complexes and then is oxidized at or near the surface of the growing mineral (e.g. Di Bella et al. 2019). Suggest adding discussion about the similarities and differences and thus how to think about Fe(III) in both the modern and past examples.

104 - I think I understand what you are saying here, but it's not clear that these adsorbed and more short-ordered solid phases will with certainty be lost during diagenesis. Maybe worth considering and not ruling out right away. Would make a

nice discussion item.

125 - okay, but not hematite? Also they are mixed phase, what about the poorly crystalline and silica phases?

132 - moreover the modern ones are tied to local hydrothermal fluxes associated with volcanic areas. Not true for many/most of the historical examples studied here? Worth discussing?

179 - appreciate this discussion and the caveats here in interpretation using the model

211 - dominated by single-celled organisms? How can this be known?

235 - doesn't follow that the carbonate carbon isotopic record is a direct map of mantle CO₂ inputs. Indeed it such a value isn't observed, but rather assumed by isotope mass balance models. Suggest striking.

243 - all this discussion of carbon isotope mass balance and potential changes in $\delta_{13}C$ are highly speculative and pretty disconnected from the paper. Would rather the discussion hold tight to discussion of the quality of the inferred DOC record and limited to discussion of prior ideas, entirely model-derived, about how DOC may have changed through time. Feels like trying to make too many points in this paper and just pulling together a possible DOC archive through time is already a win. And avoids all the extra assumptions that are not particularly well support in interpretations of the carbonate record or potential reasons for changes in biological fractionation.

155 - main text needs a major description not just of diagenesis but moreover thermal maturity of the sample units. This is super important because the main feature in the data shows a secular decrease in OC with sample age. That is a basic expectation with catagenesis and metagenesis of the organic matter. Needs to be addressed in the main text.

2156 - why not measure DOC in the waters (water column and shallow sediment pore water) from which these ooids form? Would make the argument much stronger combined with the abiotic experiments.

2167 - how should we think about the poorly crystalline iron oxide phase in the modern ooids capturing DOC? What about the silica? Needs some discussion.

2174 - at what scale and with what imaging modality would you expect to see filaments or stalks? the images here are too coarse to see the typical zetaproteobacteria shapes. Also note that many/most iron oxidizing taxa are simple half micron rods and cocci and don't produce mineralized stalks or filaments. E.g. Rhodospseudomonas and Chlorobium.

2200 - argument about spatially unique microbial metabolisms is weak here. Calvin cycle autotrophy would be expected to be similar across arbitrarily large distances regardless right? Suggest striking this argument and just juggling local microbial biomass as a possibility that is hard to rule out at present.

2210 - the concept that one can detect the presence of a biofilm texturally in ooids is not well worn, understood, documented, or otherwise demonstrated. The concept that radial textures map to biofilms is not supported. This is a characteristic feature of the growth of aragonite and is not anticipated for goethite or hematite. Results in a really weak argument. Suggest leaving out or include data from an imaging technique (e.g. reflectance spectroscopy) that would allow the detection or not of biofilm residue. e.g. should be composed of sugar polymers vs. richer elemental and chemical composition of marine DOC.

2218 - I think this is a poor straw man argument. It is true that one lineage makes such stalks, but iron-oxidizing microbes are spread across the tree of life. Indeed most don't even produce filaments (e.g. Kato et al. 2015 Front. Microbiol.). Paper would be stronger just acknowledging local microbes as a possible source that is hard to rule out, and leaving these weak arguments out.

2233 - 'detrital' is the wrong term to use here. These are soluble phases. I think you mean to say DOC with local, non-marine sources. Could include river, estuarine, hydrothermal, or porewater. Appropriate to say so. Also would be reasonable to mention and discuss that with newer MS and spectroscopy techniques we appreciate that marine DOC (despite its age) is more variable from basin to basin and even within basins than previously thought. E.g. Seidel et al. 2022 Environ. Sci. Tech. Would be good introduction and discussion items.

(Remarks on code availability)

Referee #3

(Remarks to the Author)

The authors developed a proxy for concentrations of dissolved organic carbon (DOC) in the deep ocean through the paleo period (back in time 1650 million years). Their goal is wonderfully ambitious and their approach to the problem is novel. Previous estimates have varied widely, with Rothman et al (2003) suggesting that DOC inventory reached 32×10^{18} mol C, while today's inventory is about 60×10^{15} mol C; in other words, there was a massive accumulation of carbon in the pool according to Rothman. But Fakhraee et al (2021) told us that the marine DOC pool was largely invariant across Earth's history. Is either one of these estimates correct?

Galili et al took a novel and intensive-effort approach. They recognized that DOC is stripped by particulate Fe oxides in the ocean, so they hypothesized that the organic carbon content of the oxides accumulating in ocean sediments may be suitable as proxies for past DOC concentrations. But the authors needed to test the hypothesis; they had to verify the oxides as a reliable proxy. Once they satisfied themselves on that, they explained how DOC concentrations varied in the ocean over the past many millions of years. Interestingly, and in conflict with Fakhraee et al., Galili et al report that there was indeed variability; in conflict with Rothman et al., the deviations brought inventories to much lower values than the modern ocean. Between Rothman, Fakhraee, and now Galili, we have 3 very different scenarios for DOC variability in the deep paleo ocean. Galili et al correctly note (Line 65) that “disagreement on the geologic history of marine DOC results from a dearth of direct records”, thus providing motivation for their work.

To use Fe-OC content as a proxy, the authors needed to test if ^{13}C fractionation occurred in the binding process, determine loading of OC onto geologically accumulating iron oxides goethite and hematite, and determine if DOM composition or environmental conditions affects the loading. They conducted exhaustive experiments to test these possible controls and reported (in this paper) many supplementary figures and data files to show their results. The authors determined that the proposed proxy (the Fe/OC associations) was robust against environmental changes and was reliable. This outcome allowed them to construct Fe/OC loading and ^{13}C calibration curves to predict past DOC concentrations and ^{13}C values from iron ooids recovered from sediments. The authors interpreted the DOC variability in the context of Earth-system evolution, including oxygenation and cellular complexity and size.

Overall, I am very impressed by the extensive and widely collaborative effort required to envision, conduct, and complete this analysis. It was a fun read; I congratulate the authors on a job well done. The topics considered go well beyond my disciplinary expertise, but they captured the modern DOC system well.

Specific Comments:

1. The title perfectly captures the content of the paper; good job in constructing it.
2. Lines 26/27: the ‘microbial carbon pump’ does not particularly address the “dissociation of POC”; instead, MCP focuses on the microbial production of a more recalcitrant form of DOC by the processing of more labile DOM. Instead of ‘dissociation’, perhaps better here to consider “solubilization of sinking POC as it transits through the deep ocean interior”. But even if ‘dissociation’ is preferred, recent papers on the process include Lopez and Hansell (2021, 2013).
3. The authors do not have a subject following the work “this” when found at the start of a sentence (such as Line 124). This what? Please correct throughout.
4. Caption to figure 1; what does “new” refer to in “new iron ooid...”? Is the word necessary? It isn’t helpful at present.
5. Figure 1: “Green box indicates the PETM-aged Bakchar Horizon”. The green box is very difficult to resolve in the figure.
6. The DOC end members (M-DOC, etc.) need to be introduced in the main text when first used. They reader should not have to find definitions in the methods section, though the definitions should be retained there as well.
7. Important acronyms in the captions should be explained. I see in Table ED3, for example, the acronym “Fm”, but I do not know with certainty what it is. The reader will not want to look for definitions in the many pages of the Supplement. Make it easier for the reader.

(Remarks on code availability)

Referee #4

(Remarks to the Author)
See attached file.

(Remarks on code availability)

Referee #5

(Remarks to the Author)
[see PDF]

(Remarks on code availability)

The ED and SI was almost 200 pages. I went through what I thought I needed to and was appropriate. I would say that the manuscript in this format is bordering on the un-reviewable.

Version 1:

Reviewer comments:

Referee #3

(Remarks to the Author)

I am not trained in the geochemistry that dominates the experiments in this manuscript, nor in the paleo-ocean, which is

essential for interpreting the data. However, I greatly appreciate that the authors have done extensive testing of the proxy dynamics and reliability, and that the challenges offered by reviewers more informed about the geochemical uncertainties were (to my eye) largely met with the additional testing done by the authors.

All proxies hold uncertainties, but the achievements demonstrated in this work are important. I am interested in the biogeochemical storyline, which this paper advances. We reviewers and readers cannot require an absolute belief in any paper's findings. This paper offers the reader direct measurements, a first for ascertaining DOC variability in the geologic past, which is a great advance relative to the preceding modeling efforts to address the question. I found the results of the modeling efforts to be highly intriguing and thought-provoking as they rolled out, even though I could not trust any of them because they did not agree with one another. (Interestingly, the current work does not support the model results, leaving us with a new end-member of possibilities.) However, the model results pushed our science, as evidenced by this manuscript and the amazing intellectual and analytic efforts to generate it. I do not know how accurate the results are here, but they too have high value in being intriguing and thought-provoking; they too advance our science. Future efforts toward the question will have this important contribution to compare themselves to; at some point, we hope, the results of direct data measurements and modeling approaches will coalesce around a common timeline for DOC in the ancient oceans. We need to support each tottering step toward that outcome. That is how our science works; this manuscript is the next important step toward answering the question.

Specific Comments:

1. Lines 26/27: The solubilization of sinking POC is not equivalent to the microbial carbon pump (MCP). If the solubilized material (i.e., newly introduced DOC) is converted to a recalcitrant form of DOC via microbial action, then the MCP is in play. It is not solubilization alone that constitutes the MCP, which is how I interpret the text.
2. 30-34: I do not recall that Ref 15 (Follett et al) estimated the amount of DOC in the ocean formed by the MCP; they estimated the amount of modern DOC in the deep ocean. The first estimate of DOC formation in the deep ocean by the MCP (at about 25%), that I am aware of, was Benner and Herndl (2011) "Bacterially derived dissolved organic matter in the microbial carbon pump". In N. Jiao, F. Azam, & S. Sanders (eds.) Microbial Carbon Pump in the Ocean, Science/AAAS (pp 46-48) Science/AAAS. Shortened abstract: "...Seawater bioassay experiments demonstrate that bacteria rapidly transform labile DOC to semilabile and refractory forms, suggesting enzymatic activity plays an important role in the transformation process.the molecular signatures of the transformed DOC are observed throughout the ocean water column. Bacterial transformations in the microbial carbon pump (MCP) have sequestered about 10 Pg of semilabile DOC and about 155 Pg of refractory DOC in the global ocean. The annual production of semilabile and refractory DOC in the upper ocean MCP is estimated to be 0.74 to 2.23 Pg and 0.008 to 0.023 Pg, respectively."
https://www.science.org/doi/10.1126/resource.2375852/full/scor_aaas-1714066888377.pdf
3. Please include a statement in the paragraph beginning on Line 84 on why you could not conduct the experiments with deep ocean 'refractory DOC' (RDOC) as the source material. Somewhere I read in this submission that it would take 'thousands of liters' to use unaltered seawater but a detailed reasoning is absent. There would have been no better source for testing the proxy than RDOC, had it been a possible experiment.
4. 1156: spelling of 'timing'
5. 1192: "Final hematite DOC/Fe(III)". In this case and elsewhere, please include units somewhere in the caption or the axes.

(Remarks on code availability)

Referee #4

(Remarks to the Author)

I commend the authors on their thorough revisions; I feel that they've addressed my concerns as well as can be done. I've also carefully read their responses to the other reviews. At least as far as my understanding allows (some of the concerns are somewhat outside my direct expertise) I also feel that the authors have done a decent job.

I've one remaining issue however, that I feel the authors would do well to address, at least to the extent that they can, in this current manuscript (I acknowledge that they intend to follow up on this issue in a subsequent manuscript). The issue relates to the concerns of Reviewer 5, about whether iron (oxyhydr)oxides capture a bulk DOC signal. As far as most of the evidence presented by the authors goes (more on this below), I'm inclined to agree with them that their ooids appear to capture a bulk DOC signal – or at least they appear to not be capturing only some specific subset of the bulk DOC that is operationally defined based on its lifetime, like SL-DOC. But I don't think the authors (yet) consider whether their ooids might be enriching some specific subset of the bulk DOC that possess a particular chemical composition. It's well known that during Fe-OC precipitation certain DOC components are preferentially taken up over others (thereby concentrating certain DOC components in the precipitates – see work like Eusterhues et al., 2011, ES&T and Curti et al., 2021, Commun. Earth Environ. – and many others). As such it seems plausible that the ooids might preferentially take up a subset(s) of the bulk DOC that is enriched in certain molecular classes or functional groups.

At least for the DOC sources that the authors use, it should be pretty easy to test whether the bulk DOC is taken up as is, or whether certain components of the bulk DOC are preferentially associated with the minerals – i.e., perform analyses of the chemical composition of the DOC before uptake, and after uptake (again, see Eusterhues et al., 2011, ES&T as an example). I appreciate that the authors have considered this issue to some extent via their FA experiments BUT I'm not convinced that FA represents a "singular molecular structure" – FA extracted from natural sources represents a mixture of

different organic acids, while commercial FA powders can be more homogeneous – the authors used a commercial FA powder (Mark Nature) and confirmed its purity using FTIR – but this simply shows that the powder doesn't contain substantial quantities of something else, not that it has a single molecular structure – I can't find any information on the Mark Nature website about the molecular structure.

It might be too much to ask the authors to perform additional experiments to test whether their bulk DOC is chemically fractionated on uptake, but I do feel that this is an important issue and one that the authors need to acknowledge and discuss, in terms of whether and how this might affect their DOC proxy. If this issue is unlikely to substantially impact their DOC proxy then I think it would be helpful for the authors to point this out. But is it possible, for example, that changes in the Fe-OC loading reflects changes in DOC concentration but also changes in DOC composition through time?

(Remarks on code availability)

Referee #5

(Remarks to the Author)

The authors went above and (far) beyond in their extremely patient, considered, and highly detailed reply to my earlier review, with the result that they made an extremely good and compelling argument for their paper and its revision. The revised paper reads really well, and I am almost wholly on board with its findings and arguments.

I have only 2 minor additional comments:

(1) Please made the panels in Figure 2 much larger (specifically increased time-axis resolution) so the reader can see more of the details.

(2) While only a small component of the manuscript -- I feel that the authors are trying too hard to come up with a comprehensive interpretation of their data and I am left mostly unconvinced by the hypotheses offer. The data will hopefully stand the test of time and a 'correct' interpretation may need further data and/or models.

With respect to the 2nd point -- on re-reading the manuscript, I wonder the evolution/rise of sponges is a much simpler and plausible explanation for the Neoproterozoic decline in DOC. The mid Paleozoic rise back to modern could then reflect increased DOC production (e.g., through ecosystem complexity and grazing etc.).

(Remarks on code availability)

Referee #6

(Remarks to the Author)

Please see attached pdf as figures are included.

(Remarks on code availability)

Version 2:

Reviewer comments:

Referee #6

(Remarks to the Author)

I have read through Galili et al.'s response to my review. It is serious, complete, and convincing. I recommend publication.

(Remarks on code availability)

Response to Reviewers' comments

Manuscript ID: 2024-02-02257

Title: *The geologic history of marine dissolved organic carbon from iron oxides*

Dear Editors and Reviewers,

We would like to express our sincere gratitude for the Reviewer's thoughtful and constructive feedback on our manuscript. **We are particularly grateful that all reviewers highlight the value of providing data-based dissolved organic carbon reconstructions, and that they expressed excitement in our study and keen interest in our results.** We have carefully considered all comments and have accordingly made several revisions to the manuscript; our detailed response to each comment is provided below. [Notes: (i) the Reviewers' comments are shown in italic black text, our responses are shown in blue text; (ii) all reference numbers in this document refer to the order in which they are cited within this document itself, not the order in which they are cited in the manuscript.

Summary of revisions

In response to the Reviewers' comments, we have made significant revisions to our manuscript in order to address specific concerns and strengthen our conclusions. The key revisions are summarized here:

1. Inclusion of silica in experiments to simulate ancient ocean conditions

To address the concerns of Reviewers 1 and 4 regarding the influence of dissolved silica on iron oxide formation and DOC incorporation in silica-rich ancient oceans, we conducted new hematite synthesis experiments with varying concentrations of dissolved silica (0.53 mM, 1.06 mM, and 2.13 mM SiO₂). These concentrations were chosen since they represent saturation levels from cristobalite to amorphous silica, thus aligning with estimates for Precambrian oceans.

Findings:

- Presence of silica resulted in slight increases in Fe-OC loadings and marginally heavier Fe-OC $\delta^{13}\text{C}$ values at given DOC/Fe(III) ratios independent of silica concentration.
- Overall impact of silica was minor; trends remained consistent across all experimental conditions.
- Results demonstrate our proxy's robustness under silica-rich conditions, enhancing confidence in its applicability to ancient ocean settings.

Revisions made:

- Manuscript main-text and Supplementary Discussion were updated to discuss these new findings

- New calibration curves are included as additional Extended Data figures.

2. Clarification on the choice of DOC types and acknowledgment of DOC complexity

Reviewers 4 and 5 expressed concerns about the DOC types used in our experiments, noting that important components such as carboxyl-rich alicyclic molecules (CRAM) and semi-recalcitrant to recalcitrant DOC (SRDOC, RDOC) might not be fully represented.

Revisions made:

- Updated text to acknowledge the complexity of marine DOC and inherent challenges in its full characterization.
- Clarified that our selected DOC types encompass a variety of functional groups and molecular structures but may not represent every specific compound in the marine DOC pool.
- Added discussion in both the main text and Supplementary Discussion to better articulate different DOC fractions (labile, semi-labile, semi-refractory, refractory, ultra-refractory) and their relevance to our study.

3. Capturing iron oxide aging and transformation processes in experiments

Reviewer 3 expressed concerns that our experiments might not fully capture the aging and transformation of freshly co-precipitated amorphous iron (oxyhydr)oxides and the resulting impact on organic carbon distribution.

Revisions made:

- Clarified that our experiments mimic natural iron oxide formation and transformation processes.
- Clearly articulated that our experiments mimic the transformation pathways of natural systems (i.e., in both natural environments and our experiments, crystalline iron oxides form through the transformation of poorly crystalline ferrihydrite via dissolution-reprecipitation).
- Updated explanations in the methods and Supplementary Discussion.

4. Assessment of microbial contributions using advanced techniques

Reviewer 1 expressed concerns related potential microbial contributions to Fe-OC in iron oxides (e.g., local biofilms). To address this, we performed new analysis using both Raman microspectroscopy to map Fe-OC spatial distributions and environmental DNA to assess microbial community composition.

Findings:

- *Raman microspectroscopy*: Employed Raman mapping to examine the spatial distribution of organic carbon within ooids. Consistently homogeneous distributions indicate incorporation during abiotic mineral precipitation from seawater, not from localized microbial activity or particulate organic carbon (POC).
- *Environmental DNA extraction and sequencing*: Extracted and sequenced microbial DNA from modern iron ooids. Analyses revealed extremely low DNA concentrations, near detection limits. Nevertheless, known iron-oxidizing bacterial taxa were not detected, suggesting minimal microbial contributions.

Revisions made:

- Expanded discussion in the main text regarding ooid formation mechanism as it relates to Fe-OC distribution.
- Inclusion of a new figure within the main text highlighting the homogeneous distribution of Fe-OC in select ooids using Raman microspectroscopy
- Inclusion of all relevant methods text for Raman microspectroscopy and DNA extraction and analysis.

5. Assessment of potential POC contributions and clarification of proxy limitations

Reviewer 3 raised concerns about the potential incorporation of POC within iron ooids, which would complicate interpretation of Fe-OC as a DOC signal.

Revisions made:

- Provided multiple lines of evidence indicating minimal POC contributions:
 - *Raman imaging*: Consistently homogeneous distribution of Fe-OC within ooids; inconsistent with POC incorporation.
 - *Petrographic Analyses*: Did not reveal evidence of POC incorporation.
- Clarified that reconstructed DOC concentrations should be considered *maximum estimates*, as any potential POC contribution would lead to overestimation.

6. Expanded discussion on iron ooid formation mechanisms and mineralogy

Reviewer 1 requested a clearer description of iron ooid formation within the main text.

Revisions made:

- Added a new section in the main text on ooid formation (*“Iron ooids capture ambient Fe-OC signals.”*); includes detailed discussion of formation processes, mineralogical variations, and potential iron sources.
- Updated and expanded Supplementary Discussion to provide an in-depth synthesis of current knowledge on iron ooid genesis.

7. Refinement of manuscript focus and reduction of speculative content

Reviewers 1 and 4 requested the manuscript focus more closely on DOC concentration reconstructions and omit speculative interpretations (e.g., related to changes in OC burial through time and/or photosynthetic ϵ_p).

Revisions made:

- Refocused discussion on core findings related to DOC dynamics through time; minimized speculative statements regarding carbon isotope fractionation, global carbon cycle evolution, and secular changes in carbon burial.
- Enhanced clarity on proxy limitations; acknowledged limitations of our proxy, particularly its sensitivity to different DOC types and potential environmental influences, better framing results within an accurate interpretative context.

Detailed, individual responses to each Reviewer comment can be found on the following pages.

Reviewer #1

Comment 1: **Reviewer Comment:** *Dissolved organic carbon (DOC) in the world's oceans is an important reservoir for the modern carbon cycle, and there are longstanding questions about its sources, sinks, and dynamics. In addition to this, a number of popular ideas have arisen over the past two decades about how this reservoir may have changed over Earth's history. Some of these ideas even suggest that DOC dominated major aspects of carbon cycling during climate perturbations in the past, particularly during Neoproterozoic time. However, despite their popularity, these ideas were established entirely from simple, largely unconstrained models, and remain untested by observations. In this work, the authors describe the concept behind and application of a geological proxy for reconstructing the concentrations and carbon isotopic compositions of DOC in marine basins from Proterozoic to present using observations of an odd, anachronistic sedimentary facies: iron ooids. This study describes the first serious effort to reconstruct the size of this reservoir through time. It is a hard problem. Though the work is nascent and there are many challenges/issues/questions about the approach and method, it is innovative work on an important problem and is the right kind of study for the journal under consideration.*

Response: We sincerely appreciate the Reviewer's thoughtful and encouraging comments regarding our work, and we are pleased that they recognize the significance and novelty of our approach in reconstructing past DOC concentrations and carbon isotopic compositions using iron ooids. We agree that this is a challenging problem due to the complexities of the DOC reservoir and the paucity of direct observational data through Earth's history. As the Reviewer point out, previous hypotheses about DOC dynamics during climate perturbations, particularly in the Neoproterozoic, have largely been based on unconstrained models. Our study aims to address this gap by providing empirical data to test these ideas. We have carefully considered the challenges and limitations inherent in our approach and have elaborated on these aspects in the revised manuscript to provide a more comprehensive understanding of our methods and findings.

Comment 2: **Reviewer Comment:** *The authors provide provocative interpretations of the data, with potential explanations in the context of major changes in the biological and redox histories of the global oceans and atmosphere. There are several areas in which I think the manuscript currently requires revision and perhaps some realignment of the discussion and implications to remain closer to DOC and avoid what I think are really speculative statements about the secular evolution of the global carbon cycle (particularly considerations about global isotope mass balance). I have highlighted these below. I think there are also a couple of areas where the study would benefit so substantially from additional measurements that I enumerated them here: 1) measurements of the DOC/DOM in the waters from which the modern ooids form; 2) imaging of organics within the ooids to test whether or not there are concentrated residues of biofilms or cells; 3) extraction, abundance, and sequencing of microbial DNA within the modern ooids to assess the probability of local organic sources to the samples.*

Response: We thank the Reviewer for their thoughtful comments and for recognizing the significance of our interpretations in the context of Earth's biological and redox history. We appreciate the suggestions for additional measurements, the majority of which we have now performed and which indeed strengthen our study. We specifically address each point in detail:

- (i) **Measurements of DOC/DOM in the waters from which the modern ooids form:** We completely agree that directly measuring the DOC concentrations and isotopic compositions in the waters where modern ooids form would provide valuable validation for our

proxy. However, we were unable to obtain water samples from these locations due to logistical constraints. The original sampling expeditions focused on collecting ooid samples, and water samples were not collected at that time. Furthermore, no future research expeditions to these remote and challenging sites are currently planned. We also note that the ooids have been forming over the past $\sim 10,000$ years, and modern water samples might not accurately reflect the historical conditions relevant to the majority of their formation. Because of these constraints, we were unfortunately unable to perform these measurements.

- (ii) **Imaging of organics within the ooids to test for concentrated residues of biofilms or cells:** In response to the Reviewer’s recommendation, we conducted Raman microspectroscopy to examine the spatial distribution of organic matter within ooids. Raman spectroscopy is well-suited for detecting carbonaceous organic materials within mineral matrices, even in opaque iron oxides¹. Our results reveal that the organic carbon is homogeneously distributed throughout the iron oxide layers, without localized hotspots that would indicate the presence of biofilms or concentrated microbial cells. This homogeneous distribution suggests that the organic matter was incorporated primarily during abiotic mineral precipitation from seawater, rather than being derived from localized microbial activity. We have included detailed descriptions of the imaging techniques used and the results obtained in the revised manuscript (i.e., main text, L144–L155; new main text Fig. 1; methods, L634–L683; Supplementary Discussion, L2455–L2464). This new data strengthens our interpretation that the Fe-OC signals in iron ooids reflect ambient marine DOC.
- (iii) **Extraction, abundance, and sequencing of microbial DNA within the modern ooids:** We recognize the importance of assessing potential local microbial contributions to Fe-OC. We therefore undertook efforts to extract and sequence microbial DNA from modern ooids. Due to limited sample availability as well as sample age (collected in 2018 and stored at room temperature), we faced challenges related to DNA degradation and potential contamination. Despite these challenges, we successfully extracted DNA from Panarea Island ooids and performed 16S rRNA gene amplicon sequencing. Our analyses did not detect typical iron-oxidizing microbial taxa such as *Mariprofundus*, *Gallionellaceae*, *Pseudoalteromonadaceae*, or *Hyphomonadaceae*. The microbial communities identified were not associated with known iron-oxidizing bacteria. While we cannot entirely rule out the presence of microbial activity, the absence of these key taxa suggests that microbial processes did not significantly contribute to the formation of the ooids or the incorporation of organic matter. We have incorporated these findings into the revised manuscript, providing detailed methodology, discussion of the limitations, and the implications of our results (main text, L155–L160; methods, L688–L724; Supplementary Discussion, L2478–L2501; new Fig. ED24).
- (iv) **Realignment of discussion:** In light of the Reviewer’s comments, we have revisited our discussion section to focus more closely on DOC concentration. Specifically, we have **removed speculative statements** regarding the secular evolution of the global carbon cycle and global isotope mass balance (particularly OC burial fluxes and ε_p), which were beyond the scope of our data. Our revised discussion now better emphasizes the direct findings of our study *vis-à-vis* their implications for understanding DOC concentration dynamics through time. We believe this realignment strengthens our manuscript by keeping the focus on the core contributions of our research (see, most importantly, revised section “Implications for Earth-system evolution”, main text L244–L276).

Comment 3: Reviewer Comment: *The main text requires a paragraph or two on the mechanisms that are responsible for generating the minerals and textures of iron ooids. This is a super exotic sediment type, and there are many questions about process and mechanism. There are also questions about varied mineralogy over time, not just iron oxides, but mixed valence silicates and sulfides. We need a somewhat comprehensive synthesis and critique of the mechanisms of formation of the minerals being surveyed. Thankfully there are highly limited examples of where these materials are forming in the modern. Current best hypotheses for these suggest at least in the modern cases the iron is sourced from deep hydrothermally influenced porewater that mixes with shallow marine porewater to oxidize and precipitate the iron on the grains as a cement, and then you have wave energy and sediment transport to round and further modify the textures. Some models consider microbial catalysis of the iron oxidation. Some don't. This is important to put upfront for the reader. It is also important to note that just the prevalence of this litho type is widely recognized to change with time. So too might mechanisms of its formation. The rare modern examples are proximal to volcanic sources of iron-rich fluids. There are many historical examples that appear to be on passive marine shelves and not anywhere near such an iron source. So the mechanisms of formation might change through time. Could be other minerals even. I appreciate that there is a section with some of this information in the SI, but there's not enough in the main text for a general reader to understand and follow. I appreciate the substantial amount of information in the supplement. But you have to wait until page 166 to find it.*

Response: We appreciate the Reviewer's insightful comments and agree that a comprehensive discussion on the formation mechanisms, iron sources, and mineralogical variations of iron ooids is essential for readers to fully understand our study. Iron ooids are indeed unique sedimentary features with complex formation processes, and understanding these processes is crucial for interpreting our proxy data. In response to the Reviewer's suggestion, we have added a new section to the main text titled "*Iron ooids capture ambient Fe-OC signals.*" (L125–L161), which, together with updated Supplementary Discussion text (L2393–L2550), provide a detailed synthesis of current understanding. Specifically, we address the Reviewer's concerns as follows:

- (i) **Iron sources for iron ooids:** We now discuss the varying iron sources that contribute to iron ooid formation within the main text (L131–L137). While modern examples often involve iron sourced from hydrothermal activity, variable iron sources have been suggested for geological ooidal ironstones, including volcanic or hydrothermal activities², upwelling of Fe-rich deep oceanic water³, and iron from continental weathering processes⁴. Interestingly, the carbon concentration and $\delta^{13}\text{C}$ values in Jurassic iron ooids from the Arroyofrío Bed in Spain—previously linked to volcanic sources (e.g., Ref.⁵)—are remarkably similar to those in Jurassic iron ooids from Switzerland, where iron-source has been attributed to continental weathering⁴ (e.g., Arroyofrío Bed: Fe-OC = 0.41 ± 0.04 wt %, $\delta^{13}\text{C} = -22.4 \pm 0.3$ ‰, $n = 2$; and the Parkinsoni-Württembergica Schichten: Fe-OC = 0.53 ± 0.12 wt %, $\delta^{13}\text{C} = -24.8 \pm 1.0$ ‰, $n = 2$; see section "Intra-period spatial variability" in the Supplementary Discussion, L3124–3193). Although not definitive and requiring further work, this similarity suggests that the iron source may not be a controlling factor on our proxy record.
- (ii) **Formation mechanisms:** We now elaborate on the processes responsible for iron ooid formation in the form of a new titled main text section. Specifically related to this comment, we write:

L131–L137: "Although the source of iron to ooids may have varied in the geologic past

(e.g., upward diffusion after microbial iron reduction in deep sediments), their formation mechanism has remained similar: reduced iron migrates towards the sediment-water interface, where it either forms iron-rich silicates (i.e., chamosite, nontronite) or is directly oxidised to amorphous iron (oxyhydr)oxides that ripen to crystalline phases (i.e., goethite, hematite; the subjects of this study) as concentric layers around a nucleus via wave action (see Supplementary Discussion for further description of formation pathways).”

- (iii) **Mineralogical variations and temporal changes:** We address the possible mineralogical transitions in iron ooids over geological time in our record (Supplementary Discussion section 6.4, assumption 2, L3458–L3507). While earlier studies suggested a temporal shift through time from iron silicates (e.g., chamosite) to iron oxides (e.g., goethite and hematite), recent studies present a more refined scenario. For example, it is becoming clear that hematite and chamosite can form synchronously and both can be pristine in nature. For example, in the Upper Yangtze region of South China, Middle Ordovician hematite ooids are suggested to form in open marine subtidal environments, whereas chamosite ooids in the proximal outcrops occur in restricted and semi-restricted lagoons within the same depositional system⁶. This suggests that mineralogical variations are influenced by depositional environments rather than solely by temporal changes.

Regarding the question about temporal variations in mineralogy—including iron oxides, mixed-valence silicates, sulfides, and phosphates: we find this to be a highly intriguing topic. While we are actively investigating these aspects and plan to address them in future studies, noting that they fall outside the scope of our current manuscript. Still, we include discussion on the fidelity of (particularly) hematite ooids within the present manuscript as it relates to Fe-OC proxy signals (main text L131–L137; Supplementary Text L3458–L3507).

- (iv) **Role of microbial catalysis:** We address the debate over microbial involvement in iron ooid formation. While some models consider microbial catalysis of iron oxidation, such suggestions have been made primarily for iron-rich strata rather than specifically for layered, round iron ooids. For example, Ref.⁷ discuss microbial involvement in iron-rich carbonate layers but not in ooids themselves. Similarly, Ref.⁸ suggested a potential biological origin for ferruginous coated grains but stated: “*Although no microbial remains that could have been directly associated with oncoidal (ooids) growth were recorded, we regard the above-mentioned features as indicators of biogenicity of the precipitates.*” This highlights that, despite extensive efforts, no direct microbial evidence was found to be associated with the formation of iron ooids. The literature contains further examples.

Still, we note the work of Ref.⁹, who suggest a microbial origin for iron ooids in the Middle Eocene ironstones of the Bahariya Depression, Egypt. While their research provides valuable insights into the complexity of iron ooid formation and the potential role of microbial processes, we suggest that the evidence they present for *primary* microbial involvement in the formation of iron ooids is not conclusive due to several factors:

First, the ironstone units studied by Ref.⁹ are highly diagenetically altered. In their previous work¹⁰, they document extensive subaerial weathering, pedogenesis, and subsurface alteration by acidic groundwater, leading to the development of pedogenic features including ferruginous paleosols, ironstone breccias, and colloform goethite cement¹¹. These diagenetic processes, “obliterate primary sedimentary structures”¹⁰ and potentially overprint or modify any original microbial features within the ironstones. Such extensive alteration would not meet the screening criteria we apply to ensure minimal diagenetic overprinting in our samples, and this is also why this unit is not part of our study.

Second, the microbial structures identified by Ref.⁹ in the iron ooids are on the order of tens of micrometers, observable under light microscopy (e.g., their Figure 5). In our study, we conducted thorough examinations of our iron ooid samples using high-resolution SEM and light microscopy. If such microbial structures were present in our samples, we would have readily observed them. The absence of such structures in our samples, while not definitive, may suggest that microbial involvement in the formation of the iron ooids in our record is unlikely.

Third, Ref.⁹ state the presence of at least two episodes of microbial activity in their ironstone units, including a later generation of iron-oxidizing bacteria that prevailed during post-depositional alteration. They state, “*At this stage, another iron-oxidizing bacterial generation prevailed in the acidic environment.*” This indicates a complex diagenetic history, making it challenging to differentiate between primary microbial features related to ooid formation and those resulting from later diagenetic processes. The overlapping microbial generations and alteration events complicate the interpretation of the primary origin of the microbial structures within the ooids.

Given these considerations, while microbial processes may have contributed to the formation of iron ooids in the Bahariya Depression, the extensive diagenetic alteration and multiple episodes of microbial activity make it difficult to conclusively attribute the observed features to primary microbial involvement. As such, alternative explanations involving abiotic processes should also be considered.

Regardless, in our own study, we focus on iron ooids from clear marine environments that have undergone minimal diagenetic alteration and subaerial weathering. We applied rigorous screening protocols to select samples with well-preserved primary features. Our high-resolution analyses (down to 10 nanometers) did not reveal any microbial structures akin to those reported by Ref.⁹. Additionally, following the Reviewer’s suggestions, our analyses now include Raman microspectroscopy and 16S rRNA gene amplicon sequencing, which show that Fe-OC is **homogeneously distributed** and suggest that **known iron-oxidizing microbial taxa are absent** in our samples. Combined, these results imply that abiotic processes are primarily responsible for iron ooid formation in the samples studied here (main text, L141–L161; methods, L634–L724; Supplementary Discussion, L2421–L2506; Fig. ED24; Table ED 4–5).

Therefore, while microbial involvement cannot be entirely ruled out (which we fully acknowledge in the revised text, i.e., Supplementary Discussion section “The potential role of local conditions, direct microbial precipitation, and diagenesis”, L2421–L2506), our findings suggest that abiotic processes may play a more significant role in iron ooid formation, at least in the contexts we have studied. By thoroughly examining samples that meet strict preservation criteria, we have been able to isolate and identify the processes contributing to iron ooid formation without the confounding effects of significant diagenetic alteration.

In summary, by adding new and moving relevant details from the Supplementary Discussion to the main text, we aim to provide a clear and accessible overview of iron ooid formation mechanisms for the general readers. We believe that this addition directly addresses the Reviewer’s concerns. We are grateful for the Reviewer’s suggestion, which has significantly improved the clarity and rigor of our work.

Comment 4: **Reviewer Comment:** *With access to these modern examples, as the authors have done nicely here, there are two additional observations that would be important for convincing readers that such a proxy is valuable. I really like the experimental work here to look at DOC incorporation into iron oxides under a range of plausible conditions. But these experiments don't produce anything texturally like the iron ooids (and are even missing some of the phases like silica). So it would greatly improve the study to include a direct comparison of the DOC concentrations and isotopes from the same waters from which the modern ooids are derived. They will hopefully show a relationship expected from the experiments and that would help provide support for the proxy.*

Response: We thank the Reviewer for their positive feedback on our experimental work and for the valuable suggestions to enhance our study. We acknowledge that our experimental approach, by definition, simplifies natural systems and does not replicate all complexities present in natural iron ooids. However, this simplification allows us to mechanistically explore the effects of different parameters on Fe-OC interactions under controlled conditions, which is essential for developing and validating this new proxy. We address each of the Reviewer's points below:

(i) **Textural differences between experimental products and natural iron ooids:** We recognize that our laboratory-synthesized iron oxides do not replicate the complex textures and mineralogical phases of natural iron ooids. The formation of iron ooids involves both chemical precipitation and physical processes, such as rolling and accretion in high-energy environments, which are challenging to reproduce in laboratory settings. Moreover, forming iron oxide ooids in the laboratory has, to the best of our knowledge, never been successfully achieved due to kinetic limitations. Based on our radiocarbon ages presented in this study, the estimated growth rates of iron ooids suggest that formation timescales are on the order of thousands of years. Attempts have been made to synthesize carbonate ooids in laboratory settings; for instance,¹² estimated that carbonate ooid cortex formation requires timescales of thousands of years, similar to iron ooids, making laboratory synthesis impractical.

In our experiments, we note that goethite and hematite did not precipitate directly from solution but rather formed by a relatively slow transformation of poorly crystalline precursors via dissolution and re-precipitation^{13,14}. This transformation occurred over timescales of days, which is similar to oxide formation processes in natural environments. Despite textural differences, focusing on the chemical aspects of Fe-OC interactions provides valuable insights into the fundamental processes governing organic carbon incorporation into iron oxides. Our data suggest that the mechanisms of OC adsorption and incorporation (i.e., via dissolution and re-precipitation during ripening from poorly crystalline precursors such as ferrihydrite) are comparable between our experimental products and natural systems, supporting the applicability of our results.

(ii) **Inclusion of silica in experiments:** We agree that incorporating silica into our experiments is important to better simulate natural conditions, as ancient oceans were likely more silica-rich than today—especially during the Archean and early Proterozoic Eons^{15,16}. In response to the Reviewer's suggestion (and that of Reviewer 4, below), we have conducted additional hematite synthesis experiments across a range of silica concentrations representing proposed Precambrian ocean conditions (i.e., 0.53 mM, 1.06 mM, and 2.13 mM SiO₂). All experiments were performed at 50 °C under circumneutral pH conditions using FeCl₃ as the Fe(III) source and cyanobacterial-derived DOC (C-DOC) as the DOC source; DOC and silica were added after ferrihydrite precipitation, and samples were aged for 14 days (see Supplementary Discussion L2209–L2232 for details).

Results indicate that the presence of silica imparts a small but measurable influence on Fe-OC loadings and ^{13}C fractionations relative to silica-free experiments (see Fig. 2 of this response, below; Fig. ED14 in the revised manuscript). For example, at a DOC/Fe(III) ratio of 0.028–0.031 we find:

- **0.53 mM SiO₂**: Fe-OC = 0.19 ± 0.01 %; $\Delta^{13}\text{C} = -7.57 \pm 1.93$ ‰ ($n = 6$)
- **1.06 mM SiO₂**: Fe-OC = 0.21 ± 0.01 %; $\Delta^{13}\text{C} = -6.77 \pm 2.43$ ‰ ($n = 6$)
- **2.13 mM SiO₂**: Fe-OC = 0.19 ± 0.02 %; $\Delta^{13}\text{C} = -6.61 \pm 2.24$ ‰ ($n = 6$)

All results for silica-containing samples are statistically identical regardless of silica concentration. In comparison, silica-free experiments with a DOC/Fe(III) ratio of 0.031 exhibited Fe-OC loadings of 0.13 ± 0.03 % and $\Delta^{13}\text{C}$ values of -12.65 ± 1.68 ‰ ($n = 6$). While the presence of silica slightly enhances OC loading and results in slightly heavier $\Delta^{13}\text{C}$ values for a given DOC/Fe(III) ratio, the overall impact remains minor relative to the range of observed signal in our ooid record, and observed trends with increasing DOC/Fe(III) are consistent across all experimental conditions.

(iii) **Direct comparison with DOC concentrations and isotopes from modern ooid waters:**

We agree that directly measuring the DOC concentrations and isotopic compositions from the waters where modern ooids form would greatly enhance the support for our proxy. We made significant efforts to obtain water samples for this purpose. However, due to logistical constraints and the remoteness of these sampling sites, water samples were not collected during the original expeditions, and no future research expeditions are currently planned. Additionally, considering that the ooids have been forming over the last $\sim 10,000$ years, modern water samples might not accurately reflect the historical conditions relevant to their formation. Despite these limitations, we have provided extensive experimental data and analyses to support the validity of our proxy. Furthermore, we have conducted Raman microspectroscopy to demonstrate that the organic carbon is homogeneously distributed within the iron ooids, suggesting that the OC incorporation in natural samples is in agreement with the carbon source type (i.e., DOC) used in our experiments.

Comment 5: **Reviewer Comment:** *Additionally, the authors go to some length in the SI to rule out local microbial sources for organics that might be contained within the iron oxides phases in the sediment. These arguments are currently quite weak and could be made far stronger with two different approaches. 1) using an imaging approach to document the textures in which, to rule out the presence of biofilms, communities, cells. Iron oxides can be a bit tricky to work with for many imaging techniques due to their opacity, but disaggregation and fluorescence microscopy can work quite well as the work of Clara Chan and Dave Emerson have shown for iron-oxidizing microbes. But reflectance spectroscopy and SIMS can work quite well for in situ techniques. 2) environmental sequencing works so well even in these systems that it is very straightforward at this point to extract DNA, quantify it (e.g., via Q-bit), and sequence it either with or without amplification to get a census of the microbes present and assess any potential role they might have in the genesis of the iron oxides. The supposition that microbes are absent or unimportant is poorly supported at present. And recall that most iron oxidizers are founding autotrophs in the community and fix carbon with the same biochemistry that the photoautotrophs do in the ocean E.g., Kennedy et al. 2003 Geomicrobiol, Edwards et al. 2011 ISME J, Mori et al. 2017 ISME J, Zhong et al. 2022 Microbiol Spectr. Many properties of the organic matter derived from these communities are highly similar to expectations of marine DOC. Would be better to just directly test this. The proxy and conclusions about past marine DOC will be much stronger for it.*

Response: We thank the Reviewer for highlighting this important point and for suggesting specific approaches to strengthen our arguments regarding the potential contributions of local microbial sources to the organic carbon in iron oxides. We agree that directly testing for microbial contributions would significantly enhance the robustness of our proxy and conclusions about past marine DOC. Therefore, in response to the Reviewer’s suggestions, we have conducted several additional analyses:

- (i) **Imaging techniques to detect microbial structures:** Recognizing the importance of imaging to rule out the presence of biofilms, microbial communities, or cells within the iron oxide phases, we employed *Raman microspectroscopy* to analyze the spatial distribution and nature of organic carbon at the micrometer scale within the iron ooids of our record. Raman spectroscopy allows for non-destructive analysis of both the mineral matrix and associated organic matter, overcoming challenges associated with the opacity of iron oxides. Our Raman imaging results reveal that the organic carbon is homogeneously distributed throughout the concentric layers of all studied ooids, without any localized concentrations or morphological features indicative of microbial cells, biofilms, or extracellular polymeric substances (EPS). Specifically, Raman spectra line scans and spatial maps consistently show G-band intensity characteristic of Fe-OC material across all cross sections, with no localized hotspots.

These observations suggest that the organic carbon is incorporated uniformly during mineral precipitation from marine DOC, rather than derived from localized microbial activity. In addition to discussing these results (i.e., main text, L144–L155; methods, L634–L683; Supplementary Discussion, L2455–L2464), we now include representative Raman maps and spectra in the revised manuscript (new main-text Fig. 1) to illustrate these findings.

- (ii) **Environmental DNA extraction and sequencing:** To assess the potential presence of microbial communities within iron ooids, we performed environmental DNA (eDNA) extraction and 16S rRNA gene amplicon sequencing on modern iron ooid samples from Panarea Island, Italy (Tables ED4–ED5). We followed protocols designed to minimize contamination and maximize DNA recovery, including sonication in pure ethanol to remove surface-bound DNA and grinding within a laminar flow hood to expose internal DNA and enhance extraction efficiency.

DNA quantification using Qubit fluorometry yielded extremely low concentrations, ranging from 0.17 to 0.46 ng μL^{-1} , near the detection limit of this method. Notably, DNA concentrations were significantly lower in sonicated samples compared to non-sonicated samples, supporting the effectiveness of sonication in removing surface-bound DNA and indicating minimal internal DNA content. Despite these low yields, we attempted PCR amplification of the 16S rRNA gene using universal bacterial primers. The PCR products from sonicated samples were faint or undetectable, suggesting minimal bacterial DNA present within the ooids. For non-sonicated samples, PCR amplification was more successful, indicating that detectable DNA was primarily associated with the ooid surfaces.

We proceeded with high-throughput sequencing of the PCR products from the non-sonicated samples. After quality filtering and removal of potential contaminants based on negative controls, approximately 177 amplicon sequence variants (ASVs) remained. Importantly, well-known taxa of iron-oxidizing bacteria, such as *Mariprofundus* (Zetaproteobacteria), *Gallionellaceae*, *Pseudoalteromonadaceae*, and *Hyphomonadaceae*, were **not detected** in our samples. Rather, the most abundant taxa detected were *Unclassified Bacteria*, *Desulfonatronaceae*, and *Rhizobiaceae*. While some of these taxa may include organisms in-

involved in iron cycling, there is no evidence that they are primary iron-oxidizing bacteria responsible for iron ooid formation. Furthermore, the absence of known iron-oxidizing bacteria and the low DNA concentrations suggest that microbial contributions to iron oxide precipitation and organic carbon incorporation are minimal.

These results indicate that iron ooids do not harbor significant microbial communities within their internal structure and that the contribution of microbial biomass to the organic carbon content is negligible. In addition to discussing these results in both the main text (L155–L160) and Supplementary Discussion (L2478–L2501), detailed descriptions of the eDNA extraction, PCR amplification, and sequencing procedures have been added to the Methods section (L688–L724), and primers and PCR conditions are provided in Tables ED4–ED5.

Implications for the source of organic carbon: The combination of homogeneous organic carbon distribution observed in Raman imaging and the lack of detectable internal microbial DNA strongly supports the conclusion that the organic carbon within the iron ooids is primarily derived from marine DOC incorporated during mineral precipitation, rather than from local microbial sources. We acknowledge that iron-oxidizing microbes can play important roles in iron oxide formation in certain environments, as highlighted in studies mentioned by the Reviewer. However, our findings suggest that, in the context of our samples, abiotic processes dominate the formation of iron ooids, and microbial contributions to the organic carbon content are minimal.

Limitations and future work: We recognize that the detection of low DNA concentrations does not entirely rule out the presence of past microbial activity, especially considering potential DNA degradation over time. However, the combined evidence from imaging and molecular analyses strengthens our conclusion that local microbial sources are not significant contributors to the organic carbon in the iron ooids studied. While our current methods provide strong evidence against significant microbial contributions, we agree that further studies using additional imaging techniques such as fluorescence microscopy or nanoscale secondary ion mass spectrometry (NanoSIMS) could provide even more detailed insights. We plan to explore these methods in future research to further investigate the role of microbes in iron ooid formation.

Conclusions By directly testing for microbial contributions through Raman microspectroscopy and environmental DNA analyses, we have significantly strengthened our arguments against local microbial sources of organic carbon in the iron ooids. These findings enhance the validity of our proxy and our conclusions about past marine DOC concentrations and isotopic compositions. We appreciate the Reviewer's suggestions, which have led to substantial improvements in the rigor and robustness of our study.

Comment 6: **Reviewer Comment:** *Finally, the last part of the paper discussions about isotope fractionation global carbon cycle appears a bit weak to me. It requires many assumptions and strays pretty far from the findings here. Highly speculative. Suggest removing this and just keeping the discussion close to the observations and results here. That's already a nice body of work and will come across stronger to the reader focused on DOC.*

Response: We thank the Reviewer for their valuable feedback on the final text of our manuscript. We agree that the discussion about isotope fractionation and the global carbon cycle extended beyond our primary findings and involved several assumptions that may not be fully supported by our data. In light of the Reviewer's suggestion, we have revised the manuscript to focus

more closely on our core observations and results. Specifically, we now significantly reduce the emphasis on the $\delta^{13}\text{C}$ record. We have removed the speculative interpretations regarding changing $\delta^{13}\text{C}$ signals and their implications for organic carbon burial through time. We recognize that such interpretations require additional testing and may stray from the primary findings of our study. In the revised manuscript, we now limit our discussion of $\delta^{13}\text{C}$ data to the following statement:

L262–L267: “Additionally, observed crude oil, kerogen, and DOC $\delta^{13}\text{C}$ trends—when combined with constant $\delta^{13}\text{C}_{\text{carb}}$ (binned into 100 Ma periods to dampen transient excursions; Fig. 3C)¹⁷—imply that photosynthetic organisms exhibited a larger ϵ_p and/or that organic carbon reservoirs contained higher proportions of ^{13}C -depleted compound classes (e.g., lipids) or higher proportions of material derived from ^{13}C -depleted sources (e.g., methane) in the Proterozoic.”

We do not provide further interpretation of our $\delta^{13}\text{C}$ record beyond this statement. We believe that this concise presentation of the data respects the scope of our findings and avoids unnecessary speculation. While the $\delta^{13}\text{C}$ trends are intriguing and may serve as a foundation for future research, we agree that detailed interpretations of these trends are beyond the scope of the current work. By refocusing the discussion on our primary observations related to DOC concentrations and iron ooids, we believe the manuscript is now stronger and more aligned with the data presented. We appreciate the Reviewer’s suggestion, which has helped us improve the clarity and focus of our manuscript.

Comment 7: **Reviewer Comment:** *Comments arising from the text, in line.*

Comment at Line 25: *ecosystem interactions would including processes like ‘sloppy feeding’ right? So not just phage attack, lysis, and microbial loop? Suggest clarifying.*

Response: We thank the Reviewer for this valuable suggestion. We agree that ‘sloppy feeding’ and other processes contribute significantly to the production and cycling of marine DOC, in addition to phage attack, viral lysis, and the microbial loop. To provide a more comprehensive understanding of the ecosystem interactions involved, we will revise the manuscript to include ‘sloppy feeding’ and clarify that a range of biological processes contribute to DOC dynamics. We now write:

L24–25: “...where dissolved compounds are either exuded directly by photosynthetic autotrophs or during subsequent ecosystem interactions, including primary production, ‘sloppy feeding’, viral lysis, and microbial loop interactions”

Comment at Line 34: *Connection to CO_2 here is very indirect, and indeed still debate about sign, suggest omitting in this context.*

Response: We appreciate the Reviewer’s observation regarding the indirect and debated connection between DOC dynamics and atmospheric CO_2 levels. Our intention was to summarize existing hypotheses without asserting a definitive relationship, although this relationship was *directly made* in the cited references. To avoid overstating this connection and to acknowledge the ongoing debate, we revised the text to reflect the uncertainty and avoid potential misinterpretation.

L34–L36: “Given the long residence time of this refractory component, small changes in consumption rates *could* drive large perturbations in DOC reservoir size—and thus atmospheric CO_2 levels (pCO_2)—over $\sim 10^3$ - 10^5 yr timescales^{18,19}.”

Comment at Line 51: *Too strong. The concept of high DOC through the latter Neoproterozoic is interesting, but important to make clear that it’s model derived. And explanations*

for carbon isotope anomalies in the Neoproterozoic have trended toward ideas about carbonate synthesis and preservation given mineralogical observations and patterns in Ca and Mg isotope fractionation (e.g., Ahm et al. 2021 EPSL). Suggest removing as it's too much of a straw man. 'Fundamental requirement' needs observations to support.

Response: We thank the Reviewer for pointing out that our statement may be too strong and could be misconstrued as presenting model-derived hypotheses as established facts—although we note that, at this point in the text, we are simply summarizing the current state of the field. This is not our own interpretation (which, in fact, our data disagrees with). We agree that the concept of high DOC levels during the Neoproterozoic is primarily based on models and that alternative explanations exist for carbon isotope anomalies, such as variations in carbonate preservation influenced by mineralogical changes (e.g., Ref.²⁰). However, we believe that our introduction appropriately frames this concept as a hypothesis rather than a definitive conclusion. In the manuscript, we explicitly state:

L39: “Ref.²¹ **hypothesized** that slow particle sinking prior to the evolution of multicellular and biomineralizing organisms...”

We further acknowledge the lack of observational data:

L66-67: “Such disagreement on the geologic history of marine DOC results from a dearth of direct records.”

Additionally, we present a full paragraph dedicated to describing the competing views of Neoproterozoic carbon isotope excursions:

L54–L66: “Still, other studies reject the large DOC reservoir hypothesis. For example, Ref.²² suggested that respiration of this reservoir would deplete Earth’s surface of terminal electron acceptors (O_2 , SO_4^{2-}) on timescales much shorter than those of observed isotope excursions. Similarly, Ref.²³ demonstrated that $\delta^{13}C_{org}$ and $\delta^{13}C_{carb}$ are coupled in the latest Neoproterozoic, requiring respiration of an alternative carbon source such as methane (CH_4)—which exhibits lower $\delta^{13}C$ values than both DOC and DIC—as a driver of isotope excursions. Furthermore, some authors have proposed that negative isotope excursions—despite being globally distributed and broadly synchronous—do not reflect the general state of the global carbon cycle, but instead implicate mechanisms such as late-stage diagenesis²⁴, authigenic carbonate precipitation²⁵, or marine transgressions²⁶. Consistent with these interpretations, a recent model proposed that decreased DOC respiration in the Proterozoic was balanced by decreased production via the microbial carbon pump in anoxic deep oceans, leading to largely invariant [DOC] through geologic time²⁷.”

Importantly, this paragraph is also presents the updated version of Ahm et al., 2021 (Ref.²⁰) in the form of the marine transgressions hypothesis (i.e., Busch et al., 2022; Ref.²⁶). These statements are intended to convey that the idea of elevated Neoproterozoic DOC levels is speculative and model-derived. Our goal is to highlight the existing hypotheses and underscore the need for empirical observations, which our study aims to provide.

Still, we have now changed the following text to further highlight this:

L52–53: “According to *this model*, a large DOC reservoir thus appears to be a fundamental requirement to explain the Neoproterozoic geologic record.” (where “this model” refers to Ref.²¹).

Comment at Line 61: *Appreciate the nuance in this section, but seems more important to rewrite with the above paragraph and make clear that these are all ideas concerning DOC through time that are model outcomes. Indeed we don't have accurate mechanistic models that*

can reproduce the dynamics of marine DOC today, so no surprise that the models don't agree on the past. What is needed is observations and data that can inform whether or not DOC changes across Earth history. That's really what the paper is about.

Response: We thank the Reviewer for their thoughtful feedback and for appreciating the nuances in our discussion. We agree that it is important to clarify that many ideas concerning DOC through time are derived from model outcomes, and that we currently lack accurate mechanistic models capable of reproducing the dynamics of marine DOC today. As such, it is not surprising that models do not always agree on past DOC dynamics. We also concur that what is needed are observations and data to inform whether DOC changes occurred across Earth's history, which is precisely the focus of our paper.

To address the Reviewer's suggestion, we have revised the paragraph to make it clear that the concept of a large DOC reservoir in the Neoproterozoic is based on specific model hypotheses. In particular, we have rephrased the sentence that used to read, "A large DOC reservoir thus appears to be a fundamental requirement to explain the Neoproterozoic geologic record", to now read:

L52–53: "According to *this model*, a large DOC reservoir thus appears to be a fundamental requirement to explain the Neoproterozoic geologic record."

This change emphasizes that this idea stems from a particular model outcome rather than an established fact. Furthermore, while many interpretations of Neoproterozoic DOC levels are indeed model-derived, there are also studies that incorporate observational data to explore DOC dynamics across Earth's history. For example, Fike et al. (2006) (Ref.²⁸), Swanson-Hysell et al. (2010) (Ref.²⁹), and Johnston et al. (2012) (Ref.²³), which are all cited in the main text, are based on observational carbon-isotope data. By including these references, we acknowledge that there are observational studies contributing to our understanding of DOC changes over geological time. Additionally, we already note in the manuscript:

L63: "Consistent with these interpretations, a recent model proposed...",

which again acknowledges that these are interpretations and models rather than established facts. We have updated the relevant section to emphasize the model-derived nature of these hypotheses and to highlight the need for empirical observations. Our goal is to provide the observations and data necessary to inform whether DOC changes occurred across Earth's history, which is indeed what our paper aims to address.

Comment at Line 71: *suggest toning this down a bit. Formally an anachronism, we still have many questions about how they form, and indeed there may not be a common genetic mechanism (e.g., Young 1989, Salama et al. 2013, Matheson and Pufahl 2021, Di Bella et al. 2019). Because they don't typically form in widespread marine environments (outside rare volcanogenic environments) today it is challenging to draw conclusions about how they formed across marine shelves at times in the past with much certainty. Would be useful to modify this paragraph to instead have a small discussion of the various different minerals that are potentially important, and how you're examining one potential mechanism for consideration of the use of iron oolites as a potential marine DIC [DOC] proxy. Addendum: I see much later in the SI that there is a section dedicated to this. Suggest revising that heavily and then bringing a long paragraph with the discussion and caveats into the main text.*

Response: We appreciate the Reviewer's suggestion to provide a more nuanced and detailed discussion regarding the formation of iron ooids (see our response to the Reviewer's Comment 3, above). While we acknowledge that there are still questions about iron ooid formation

mechanisms and that a single common genetic pathway may not exist, recent discoveries have significantly advanced our understanding.

Historically, the absence of modern analogues led to hypotheses that iron ooidal formations were products of alteration or ferruginization of primary carbonate ooids or formed through other diagenetic processes. However, the recent discovery of two modern ooidal ironstone sites^{30,31} challenges these earlier views and suggests that iron ooids can form directly in marine environments under certain conditions. These modern examples indicate that iron ooids can form in shallow, wave-agitated marine settings, and that the basic requirements for their formation—an aqueous solution with available iron and gentle movement (currents) facilitating the accretion of multilayered structures—are consistent across different environments and time periods. These findings imply that, despite variations in mineralogy (e.g., goethite, hematite, chamosite) and specific formation pathways, the fundamental processes leading to iron ooid formation may be more universal than previously thought. Additionally, some studies have re-evaluated the so-called channel iron deposits and suggest that they result from reworked marine ooidal ironstones (e.g., Ref.³²), although this is beyond the scope of our current discussion.

We agree that discussing their formation mechanisms is crucial for interpreting our proxy. Therefore, we have added a comprehensive paragraph titled “*Iron ooids capture ambient Fe-OC signals*” (L125–L161) to the main text, which, together with updated Supplementary Discussion text (L2393–L2550), provide a fuller picture of our current understanding.

Comment at Line 76: *Need some discussion here about what is meant by Fe(III). Need some discussion of precipitation mechanisms in mind. In the experiments it is supplied to the solution as a solid ferric chloride, and that seems pretty different from working hypotheses about the formation of iron ooids wherein the iron is supplied as ferrous soluble complexes and then is oxidized at or near the surface of the growing mineral (e.g., Di Bella et al. 2019). Suggest adding discussion about the similarities and differences and thus how to think about Fe(III) in both the modern and past examples.*

Response: We appreciate the Reviewer highlighting the importance of clarifying the precipitation mechanisms in our experiments relative to natural iron ooid formation. Understanding the similarities and differences between our experimental setup and natural processes is crucial for proper interpretation.

In our environmentally relevant pH synthesis experiments, crystalline goethite and hematite do not form directly from solution. Instead, they form through a dissolution-reprecipitation mechanism involving a precursor poorly crystalline phase, ferrihydrite. Initially, Fe(III) precipitates as amorphous ferrihydrite upon mixing of Fe(III) salts with an alkaline solution. Over time, this ferrihydrite transforms into more crystalline iron oxides like goethite and hematite via dissolution-reprecipitation processes¹³. This mechanism closely mirrors that in natural environments where iron ooids form. In nature, Fe(II) supplied from various sources—for example, hydrothermal vents or upwelling of ferruginous deep waters—is rapidly oxidized to Fe(III) when it encounters oxygenated conditions at or near the sediment-water interface. The Fe(III) precipitates as ferrihydrite, which subsequently undergoes maturation to form crystalline iron oxides like goethite and hematite through similar dissolution and re-precipitation pathways.

To clarify these points, we now elaborate on this topic further in the Supplementary Discussion (L2393–L2550).

Comment at Line 104: *I think I understand what you are saying here, but it's not clear that these adsorbed and more short-ordered solid phases will with certainty be lost during diagenesis. Maybe worth considering and not ruling out right away. Would make a nice discussion*

item.

Response: We thank the Reviewer for this insightful comment. In this study, we have thoroughly investigated the presence of poorly crystalline iron oxide phases in our iron ooid record. Our ongoing Mössbauer spectroscopy analyses (not part of the current manuscript) indicate that—even in modern iron ooids from Panarea Island (Italy), which are presumably less than 10,000 years old—the precursor ferrihydrite phase has already fully transformed into nanocrystalline goethite. Additionally, our extensive petrological examinations of the older samples show that ferrihydrite is absent, with iron oxides present as crystalline goethite and hematite. These observations suggest that, in our samples, the adsorbed and poorly crystalline iron oxide phases are not preserved during diagenesis. Therefore, while we acknowledge that under certain conditions these phases might be retained, in the context of our study, they are not present.

Comment at Line 125: *Okay, but not hematite? Also they are mixed phase, what about the poorly crystalline and silica phases?*

Response: That is correct, the modern sites produce goethite ooids that are free of any traces of hematite (i.e., it is not a mixed mineralogy). Please see our response to the above comment regarding poorly crystalline phases.

Comment at Line 132: *Moreover the modern ones are tied to local hydrothermal fluxes associated with volcanic areas. Not true for many/most of the historical examples studied here? Worth discussing?*

Response: We appreciate the Reviewer bringing attention to the differences in iron sources between modern and ancient iron ooids. It is true that modern marine iron ooids are associated with local hydrothermal fluxes linked to volcanic/hydrothermal activity, while many historical examples may have formed in environments without such influences. As listed in our response to the Reviewer's main Comment 3, above, we now discuss this aspect in the revised manuscript (main text L131–137).

Comment at Line 179: *Appreciate this discussion and the caveats here in interpretation using the model*

Response: We thank the Reviewer for their positive feedback on our discussion and the inclusion of caveats in the interpretation of our model. We recognize the importance of transparently addressing the limitations and assumptions inherent in our approach, and we are pleased that this aspect of our manuscript was appreciated.

Comment at Line 211: *Dominated by single-celled organisms? How can this be known?*

Response: We appreciate the Reviewer's question regarding our statement that the marine biosphere was dominated by single-celled organisms during the Paleo- to Mesoproterozoic. Our assertion is based on current understanding of Earth's early biosphere and the timing of the evolution of multicellularity. While outside the scope of this manuscript, we refer the Reviewer to Refs.^{33–36} for discussion and details. This statement is additionally supported by the fossil-record, which shows an absence of evidence for multicellularity at this time (e.g., Ref.³⁷).

Comment at Line 235: *Doesn't follow that the carbonate carbon isotopic record is a direct map of mantle CO₂ inputs. Indeed such a value isn't observed, but rather assumed by isotope mass balance models. Suggest striking.*

Response: Thank you for pointing out this important consideration. Following the Reviewer's advice, we have revised the manuscript to remove speculative discussions related to the carbon isotope portion of the record. We have refocused the narrative to emphasize the primary findings on DOC concentration changes through time.

Comment at Line 243: *All this discussion of carbon isotope mass balance and potential changes in ε_p are highly speculative and pretty disconnected from the paper. Would rather the discussion hold tight to discussion of the quality of the inferred DOC record and limited to discussion of prior ideas, entirely model-derived, about how DOC may have changed through time. Feels like trying to make too many points in this paper and just pulling together a possible DOC archive through time is already a win. And avoids all the extra assumptions that are not particularly well supported in interpretations of the carbonate record or potential reasons for changes in biological fractionation.*

Response: Thank the Reviewer for this comment. As in our previous response, we have now revised the manuscript to eliminate speculative content regarding the carbon isotope data. We have concentrated on presenting our main results concerning DOC concentration trends over time.

Comment at Line 155: *main text needs a major description not just of diagenesis but moreover thermal maturity of the sample units. This is super important because the main feature in the data shows a secular decrease in OC with sample age. That is a basic expectation with catagenesis and metagenesis of the organic matter. Needs to be addressed in the main text.*

Response: We thank the Reviewer for highlighting the importance of addressing thermal maturity and diagenesis in the main text. While we understand that thermal maturation can lead to a secular decrease in organic carbon (OC) content due to catagenesis and metagenesis, our data do not show a secular decrease in OC content with sample age. The Fe-bound organic carbon (Fe-OC) loadings has a distinct structure with Mesoproterozoic hematite samples have higher concentration than the Neoproterozoic samples, suggesting that thermal maturity has not significantly altered the OC content in our samples.

We assume the Reviewer may be referring to trends in the carbon isotope composition ($\delta^{13}\text{C}$) of the organic matter. This is an important consideration. In response, we have minimized speculative discussions about the $\delta^{13}\text{C}$ data in the main text to focus on our primary findings regarding DOC concentrations through time.

A detailed discussion of the thermal maturity of each unit in our record is provided in the Supplementary Discussion (L2556–L3080). We hope this adequately addresses the Reviewer's comment.

Comment at Line 2156: *Why not measure DOC in the waters (water column and shallow sediment pore water) from which these ooids form? Would make the argument much stronger combined with the abiotic experiments.*

Response: We appreciate the Reviewer's suggestion to measure DOC concentrations and isotopic compositions in the waters where modern ooids form, as this would indeed strengthen the support for our proxy. Unfortunately, due to logistical constraints and the remote locations of the sampling sites, we were unable to collect water samples during the original expeditions, and no future research trips are currently planned. Additionally, considering that the ooids have been forming over the last $\sim 10,000$ years, modern water samples might not accurately reflect the historical conditions relevant to their formation. We acknowledge the value of direct measurements and hope that future studies may obtain water samples to further validate our proxy.

Comment at Line 2167: *How should we think about the poorly crystalline iron oxide phase in the modern ooids capturing DOC? What about the silica? Needs some discussion.*

Response: Thank you for bringing up this important point. In both modern ooids and our experiments, poorly crystalline iron oxide phases such as ferrihydrite play a significant role in

capturing DOC due to their high surface area and reactivity. These phases are especially effective at adsorbing DOC during their initial formation. As ferrihydrite ages, it undergoes a transformation through dissolution and re-precipitation processes, eventually becoming crystalline iron oxides such as goethite and hematite. This aging process occurs in both natural environments and our experimental systems. The transformation can influence the retention and preservation of DOC within the iron oxides.

Regarding silica, we acknowledge that it can affect the formation and stability of iron oxide phases. In response to the Reviewer's suggestion, we have conducted new experiments that include the presence of silica to better replicate natural conditions. These experiments indicate that silica has limited effect on the incorporation of DOC within iron oxides. Interestingly, one of our new Raman spectroscopy analysis of Ordovician goethite ooid (main text Fig. 1, Panel E), shows that there may be a negative correlation between the darker (silica-rich layers) and the presence of organic matter. But this is beyond the scope of the current work.

Comment at Line 2174: *At what scale and with what imaging modality would you expect to see filaments or stalks? The images here are too coarse to see the typical Zetaproteobacteria shapes. Also note that many/most iron oxidizing taxa are simple half-micron rods and cocci and don't produce mineralized stalks or filaments. E.g., Rhodopseudomonas and Chlorobium.*

Response: We appreciate the Reviewer's insightful comment regarding the detection of microbial structures within iron ooids and the appropriate scales and imaging modalities required. Indeed, many iron-oxidizing bacteria are simple half-micron rods and cocci that do not produce mineralized stalks or filaments, as exemplified by *Rhodopseudomonas* and *Chlorobium*. To thoroughly investigate the potential presence of such microorganisms, we employed high-resolution scanning electron microscopy (SEM) with a resolution better than 10 nm—sufficient to detect structures as small as 0.5 μm . All samples in our study, including both modern and ancient ooids, were meticulously examined using this high-resolution SEM and light microscopy. We fully detail these imaging techniques in the Supplementary Discussion (L2556–L3080). Despite this detailed analysis, we did not observe any microbial structures such as filaments, stalks, rods, or cocci within the ooids. This contrasts with studies like Ref.⁹, where microbial-derived structures were identified using much lower-resolution imaging methods.

Nevertheless, we agree with the Reviewer that this logic and argumentation as presented in the original manuscript could be strengthened. Thus, to assess the potential for concentrated residues of biofilms or microbial cells, we conducted Raman microspectroscopy on several samples. Raman spectroscopy is highly effective for detecting carbonaceous organic materials within mineral matrices, even in opaque iron oxides. Our results revealed that Fe-OC is homogeneously distributed throughout the iron oxide layers, without localized hotspots that would indicate the presence of biofilms or concentrated microbial cells. This homogeneous distribution suggests that the organic matter was incorporated uniformly during abiotic mineral precipitation from seawater, rather than being derived from localized microbial activity.

In addition, we undertook efforts to extract and sequence microbial DNA from the modern ooids collected from Panarea Island. Despite challenges related to DNA degradation due to sample age and storage conditions, we successfully performed 16S rRNA gene amplicon sequencing. Our analyses did not detect typical iron-oxidizing microbial taxa such as *Mariprofundus*, *Gallionellaceae*, *Pseudoalteromonadaceae*, or *Hyphomonadaceae*. The microbial communities identified were not associated with known iron-oxidizing bacteria. While we cannot entirely rule out the presence of microbial activity, the absence of these key taxa suggests that microbial processes did not significantly contribute to the formation of the ooids or the incorporation of organic matter.

By combining high-resolution imaging, Raman microspectroscopy, and microbial DNA sequencing, we have provided comprehensive evidence against the presence of microbial structures or significant microbial activity within the iron ooids we studied. These findings support our interpretation that abiotic processes were primarily responsible for iron ooid formation in our samples. We have incorporated detailed descriptions of these methods and results into the revised manuscript (main text, L142–L159; methods, L634–L724; Supplementary Discussion, L2421–L2506; Fig. ED24). We hope this addresses the Reviewer’s concerns and clarifies our approach to investigating potential microbial contributions.

Comment at Line 2200: *Argument about spatially unique microbial metabolisms is weak here. Calvin cycle autotrophy would be expected to be similar across arbitrarily large distances regardless, right? Suggest striking this argument and just juggling local microbial biomass as a possibility that is hard to rule out at present.*

Response: We appreciate the Reviewer’s comment and the opportunity to clarify our argument. Our intention was to emphasize that the remarkable consistency of Fe-OC loadings and carbon isotope values over large spatial scales suggests that local variations in microbial activity are unlikely to be the primary drivers of the observed signals. We agree that Calvin cycle autotrophy is expected to produce similar carbon isotopes values across large distances, which supports our point that regional differences in microbial metabolisms are not causing significant variability in our data. We have now revised the original text as per the Reviewer’s suggestions.

Comment at Line 2210: *The concept that one can detect the presence of a biofilm texturally in ooids is not well worn, understood, documented, or otherwise demonstrated. The concept that radial textures map to biofilms is not supported. This is a characteristic feature of the growth of aragonite and is not anticipated for goethite or hematite. Results in a really weak argument. Suggest leaving out or include data from an imaging technique (e.g., reflectance spectroscopy) that would allow the detection or not of biofilm residue. E.g., should be composed of sugar polymers vs. richer elemental and chemical composition of marine DOC.*

Response: We thank the Reviewer for highlighting the limitations of using textural features to detect biofilms within iron ooids. We acknowledge this and followed the Reviewer suggestions to include eDNA and Raman microspectroscopy analysis in our revised manuscript as previously described (e.g., see our detailed response to the Reviewer’s Comment 3, above).

Comment at Line 2218: *I think this is a poor straw man argument. It is true that one lineage makes such stalks, but iron-oxidizing microbes are spread across the tree of life. Indeed most don’t even produce filaments (e.g., Kato et al. 2015 Front. Microbiol.). Paper would be stronger just acknowledging local microbes as a possible source that is hard to rule out, and leaving these weak arguments out.*

Response: We appreciate the Reviewer’s critique of our argument, which we now agree was weak in the original manuscript. To significantly strengthen this line of reasoning, we followed the Reviewer’s suggestions to include eDNA and Raman spectroscopy analysis in our revised manuscript (e.g., see our detailed response to the Reviewer’s Comment 3, above).

Comment at Line 2233: *‘Detrital’ is the wrong term to use here. These are soluble phases. I think you mean to say DOC with local, non-marine sources. Could include river, estuarine, hydrothermal, or porewater. Appropriate to say so. Also would be reasonable to mention and discuss that with newer MS and spectroscopy techniques we appreciate that marine DOC (despite its age) is more variable from basin to basin and even within basins than previously thought. E.g., Seidel et al. 2022 Environ. Sci. Technol. Would be good introduction and discussion items.*

Response: We thank the Reviewer for pointing this out—as is correctly articulated here, these are strictly speaking not “detrital” inputs. We have rephrased this to now read “DOC from local, non-marine sources” in the revised manuscript.

In summary, we sincerely thank the Reviewer for their thorough evaluation of our manuscript and for providing valuable suggestions that have significantly strengthened our work. the Reviewer’s insightful comments have helped us to clarify key aspects of our study, particularly regarding the mechanisms of iron ooid formation, the potential role of microbial contributions, and the variability of marine DOC. We have revised the manuscript accordingly, incorporating additional data, refining our interpretations, and enhancing the clarity of our discussion. We believe these revisions have significantly improved the quality and impact of our manuscript, and we are grateful for the Reviewer’s constructive feedback.

Reviewer #3

Comment 1: **Reviewer Comment:** *The authors developed a proxy for concentrations of dissolved organic carbon (DOC) in the deep ocean through the paleo period (back in time 1650 million years). Their goal is wonderfully ambitious and their approach to the problem is novel. Previous estimates have varied widely, with Rothman et al. (2003) suggesting that DOC inventory reached 32×10^{18} mol C, while today's inventory is about 60×10^{15} mol C; in other words, there was a massive accumulation of carbon in the pool according to Rothman. But Fakhraee et al. (2021) told us that the marine DOC pool was largely invariant across Earth's history. Is either one of these estimates correct?*

Response: We thank the Reviewer for the positive feedback and for highlighting the ambition and novelty of our approach.

Comment 2: **Reviewer Comment:** *Galili et al. took a novel and intensive-effort approach. They recognized that DOC is stripped by particulate Fe oxides in the ocean, so they hypothesized that the organic carbon content of the oxides accumulating in ocean sediments may be suitable as proxies for past DOC concentrations. But the authors needed to test the hypothesis; they had to verify the oxides as a reliable proxy. Once they satisfied themselves on that, they explained how DOC concentrations varied in the ocean over the past many millions of years. Interestingly, and in conflict with Fakhraee et al., Galili et al. report that there was indeed variability; in conflict with Rothman et al., the deviations brought inventories to much lower values than the modern ocean. Between Rothman, Fakhraee, and now Galili, we have 3 very different scenarios for DOC variability in the deep paleo ocean. Galili et al. correctly note (Line 65) that "disagreement on the geologic history of marine DOC results from a dearth of direct records", thus providing motivation for their work.*

Response: We greatly appreciate the Reviewer's detailed summary and support of our approach. Still, Reviewers 1 and 5 suggested rewording this introduction section to better emphasize that previous studies are indeed model-derived and not supported by our data (see, e.g., Reviewer 1's comments on L34 and L51). Thus, in the revised manuscript, we have made a better distinction between the previous model-derived hypotheses of Ref.²¹ and Ref.²⁷), and our own data-driven study.

Comment 3: **Reviewer Comment:** *To use Fe-OC content as a proxy, the authors needed to test if ^{13}C fractionation occurred in the binding process, determine loading of OC onto geologically accumulating iron oxides goethite and hematite, and determine if DOM composition or environmental conditions affect the loading. They conducted exhaustive experiments to test these possible controls and reported (in this paper) many supplementary figures and data files to show their results. The authors determined that the proposed proxy (the Fe/OC associations) was robust against environmental changes and was reliable. This outcome allowed them to construct Fe/OC loading and ^{13}C calibration curves to predict past DOC concentrations and ^{13}C values from iron ooids recovered from sediments. The authors interpreted the DOC variability in the context of Earth-system evolution, including oxygenation and cellular complexity and size.*

Response: We greatly thank the Reviewer for acknowledging the thoroughness of our experimental work!

Comment 4: **Reviewer Comment:** *Overall, I am very impressed by the extensive and widely collaborative effort required to envision, conduct, and complete this analysis. It was a fun read; I congratulate the authors on a job well done. The topics considered go well beyond my disciplinary expertise, but they captured the modern DOC system well.*

Response: We are grateful for the Reviewer’s kind words and appreciation of our collaborative effort. We have strived to ensure that our study is accessible to a broad audience, and we are pleased that it resonated with the Reviewer.

Comment 5: **Reviewer Comment:** *Specific Comments:*

Comment on title: *The title perfectly captures the content of the paper; good job in constructing it.*

Response: Thank you for the positive feedback on the title. We have retained it as is.

Comment on Lines 26/27: *the ‘microbial carbon pump’ does not particularly address the “dissociation of POC”; instead, MCP focuses on the microbial production of a more recalcitrant form of DOC by the processing of more labile DOM. Instead of ‘dissociation’, perhaps better here to consider “solubilization of sinking POC as it transits through the deep ocean interior”. But even if ‘dissociation’ is preferred, recent papers on the process include Lopez and Hansell (2021, 2013).*

Response: We thank the Reviewer and now changed “dossociation” to “solubilization” throughout the main text. We now additionally cite the newer reference from the suggested relevant literature; “Lopez, C. N., & Hansell, D. A. (2021). Evidence of deep DOC enrichment via particle export beneath subarctic and northern subtropical fronts in the north pacific. *Frontiers in Marine Science*, 8, 659034.”

Comment on L124: *The authors do not have a subject following the word “this” when found at the start of a sentence (such as Line 124). This what? Please correct throughout.*

Response: In the instance of L124 raised by the Reviewer, “this” refers to the interpretation of iron ooid formation as described in the previous sentence. In this case, the word “this” is a demonstrative pronoun and is allowed to serve as the subject of a sentence. Regardless, we have reviewed the manuscript and corrected instances where “this” was used without a clear subject.

Comment on Fig. 1 caption: *What does “new” refer to in “new iron ooid...”? Is the word necessary? It isn’t helpful at present.*

Response: We have revised the caption of Figure 1 to remove the word “new” for clarity.

Comment on Figure 1: *“Green box indicates the PETM-aged Bakchar Horizon”. The green box is very difficult to resolve in the figure.*

Response: We have adjusted Figure 1 (now Figure 2) to improve the visibility of the green (now orange) box indicating the PETM-aged Bakchar Horizon.

Comment on L88-L90: *The DOC end members (M-DOC, etc.) need to be introduced in the main text when first used. The reader should not have to find definitions in the methods section, though the definitions should be retained there as well.*

Response: We have introduced the DOC end members in the main text where they first appear and retained the definitions in the methods section for completeness.

L89–L91: “DOC sources include: (i) modern-marine analog DOC from microcosm cultures containing diatoms and marine heterotrophic bacteria (termed “M-DOC”), representing eukaryote-dominated communities; (ii) cyanobacterial biomass leachate (termed “C-DOC”), representing prokaryote-dominated communities; and (iii) dissolved fulvic acid (termed “FA”), representing modern terrestrial humic substances”

Comment on Table ED3 *Important acronyms in the captions should be explained. I see in Table ED3, for example, the acronym “Fm”, but I do not know with certainty what it is. The*

reader will not want to look for definitions in the many pages of the Supplement. Make it easier for the reader.

Response: We have added explanations for all important acronyms in the captions, including “Fm” in Table ED3, to make it easier for the reader to follow.

Reviewer #4

Comment 1: **Reviewer Comment:** *This study presents a laudable undertaking to develop the first proxy for past ocean dissolved organic carbon (DOC) signatures using co-precipitated organic carbon associated with iron oxide minerals in iron ooids. The idea that organic carbon associated with iron oxides might be used to reconstruct ocean DOC dynamics is original and really exciting, and if properly calibrated, this approach could make a significant contribution to the Earth sciences, because tracking ocean DOC dynamics over Earth history will allow better understanding of how DOC cycling links to and impacts oxygenation and climate. Following development of their ocean DOC proxy, the authors apply this to a suite of marine iron ooid-containing formations deposited over the past 1650 million years. Via an extensive series of commendable laboratory experiments, the authors present a calibrated proxy that predicts DOC concentrations were near modern levels in the Paleoproterozoic, decreasing by 90-99% in the Neoproterozoic before sharply rising in the Cambrian. These dynamics are in turn interpreted to reflect three ecological and biogeochemical stages in Earth's evolution, namely a progression from (i) small single-celled organisms with deep oceans that were severely hypoxic; to (ii) larger more complex organisms with little change in ocean oxygenation; towards (iii) continued organism growth and a transition to oceans that were fully oxygenated.*

Response: We thank the Reviewer for the detailed summary of our work and the recognition of the importance of understanding DOC cycling.

Comment 2: **Reviewer Comment:** *This proxy for ocean DOC dynamics is principally based on a set of clever laboratory experiments that explore the controls on DOC coprecipitation with iron oxides, under scenarios designed to mimic those of appropriate natural oceanic settings. The authors have taken extreme care to identify areas of uncertainty and explore these both experimentally and computationally. The Supplementary Discussion is extensive. Despite this, however, I have a number of concerns that I think might present some fundamental issues with the proxy that preclude publication of this study in its present form. I describe the main ones below.*

Response: We appreciate the thorough evaluation of our experimental approach. Below, we respond to each of the specific concerns raised and describe the revisions we have made to address them.

Comment 3: **Reviewer Comment:** *Whilst the most common iron (oxyhydr)oxides to precipitate from oxic seawater and sediment porewaters in the modern ocean are undoubtedly ferrihydrite (initially), which then ages and transforms into more crystalline phases (like goethite), the oceans in the geologic past were silica-rich. Compared to modern seawater (<0.1 mM Si), it is likely that in the Archean, the oceans were probably saturated with amorphous silica (2.2 mM), or at least saturated with cristobalite (0.67 mM). This means that the iron (oxyhydr)oxides in the past oceans were silica-rich, and this is a problem because iron-silica coprecipitation is well known to reduce the point of zero charge (PZC) of the resulting particle, which makes this particle much less reactive towards DOC. So to properly ground-truth this proxy, the experiments must be performed under dissolved silica concentrations that are representative of the past oceans. Unfortunately, I cannot see a way around this.*

Response: We sincerely appreciate the Reviewer's insightful comment regarding the influence of silica on iron (oxyhydr)oxide formation and reactivity toward DOC in the context of ancient silica-rich oceans. We agree that higher silica concentrations in the Archean and early Proterozoic oceans could alter the physicochemical properties of iron oxides, including the point of

zero charge (PZC) and subsequent interactions with DOC.

To address this important consideration and to better simulate the conditions of ancient silica-rich oceans, we have conducted additional hematite synthesis experiments (see Fig. 1 of this response) that incorporate dissolved silica concentrations representative of proposed Precambrian ocean conditions. Specifically, we performed experiments with silica concentrations of 0.53 mM, 1.06 mM, and 2.13 mM SiO_2 , which correspond to saturation with respect to cristobalite and amorphous silica, aligning with estimates for Archean ocean silica levels^{15,16}.

Figure 1: X-ray diffractograms of hematite synthesized under circumneutral pH in the presence of C-DOC and silica. Red lines denote diagnostic hematite peaks (ICDD PDF 01-073-9835); no other mineral signatures were detected. Annotations specify: (i) DOC type, (ii) carbon to iron molar ratio in the initial solution (“DOC/Fe(III)”), (iii) processing stage (i.e., “raw”, “-Fh/PC”, or “-Fh/PC -ads. OC”), (iv) silica concentration in parent solution (mM; see Methods for processing details).

All experiments were carried out at circumneutral pH and at 50 °C using FeCl_3 as the Fe(III) source and cyanobacterially derived DOC (C-DOC) as the DOC source; DOC and silica were added after ferrihydrite precipitation, and samples were aged for 14 days to promote transformation to hematite (see Supplementary Discussion L2209–L2232 for details).

Results indicate that the presence of silica imparts a small but measurable influence on Fe-OC loadings and ^{13}C fractionations relative to silica-free experiments (see Fig. 2 of this response, below; Fig. ED14 in the revised manuscript). For example, at a DOC/Fe(III) ratio of 0.028–0.031 we find:

- **0.53 mM SiO_2** : Fe-OC = $0.19 \pm 0.01 \%$; $\Delta^{13}\text{C} = -7.57 \pm 1.93 \text{‰}$ ($n = 6$)
- **1.06 mM SiO_2** : Fe-OC = $0.21 \pm 0.01 \%$; $\Delta^{13}\text{C} = -6.77 \pm 2.43 \text{‰}$ ($n = 6$)
- **2.13 mM SiO_2** : Fe-OC = $0.19 \pm 0.02 \%$; $\Delta^{13}\text{C} = -6.61 \pm 2.24 \text{‰}$ ($n = 6$)

All results for silica-containing samples are statistically identical regardless of silica concentration. In comparison, silica-free experiments with a DOC/Fe(III) ratio of 0.031 exhibited Fe-OC loadings of $0.13 \pm 0.03 \%$ and $\Delta^{13}\text{C}$ values of $-12.65 \pm 1.68 \text{‰}$ ($n = 6$). While the presence of silica slightly enhances OC loading and results in slightly heavier $\Delta^{13}\text{C}$ values for a given DOC/Fe(III) ratio, the overall impact remains minor relative to the range of observed signal in our ooid record, and observed trends with increasing DOC/Fe(III) are consistent across all experimental conditions.

9

10 **Figure 2: Impact of parent solution silica concentration on hematite Fe-OC signals.** (A)
 11 Fe-OC loadings (in wt %) and (B) isotope fractionations ($\Delta^{13}\text{C} = \delta^{13}\text{C}_{\text{Fe-OC}} - \delta^{13}\text{C}_{\text{DOC}}$, in ‰)
 12 as a function of DOC/Fe(III) ratio for four different SiO_2 concentrations (marker colors). Fe-
 13 OC loadings and $\Delta^{13}\text{C}$ values are largely independent of the presence and concentration of
 14 dissolved silica. Experimental conditions were: 50°C , pH = 8, C-DOC as DOC source, DOC
 15 addition after ferrihydrite precipitation, aged up to 14 days, measurement of “–Fh/PC –ads.
 16 OC” fractions.

These findings suggest that, although iron-silica coprecipitation can reduce the PZC of iron oxides³⁸, the effect on DOC adsorption under our experimental conditions is minimal.

This is consistent with literature reports indicating that the adsorption of organic compounds on iron oxides like hematite and goethite is primarily governed by electrostatic interactions, ligand exchange, and hydrogen bonding, which are strongly influenced by pH relative to the PZC³⁹.

While silica incorporation can shift the PZC to lower values, making the iron oxide surface less positively charged at a given pH, it seems that our experiments conducted at pH 8 remain within the range where adsorption processes are effective. This is supported by Ref.⁴⁰, which have shown that sorption of monosilicic acid by iron oxides is pH-dependent, with significant sorption beginning around pH 7–8. At pH 8, sorption of silica onto iron oxides is substantial but does not completely saturate the surface of the oxides, allowing for continued interaction with DOC. Interestingly, Ref.⁴⁰ additionally demonstrated that organic acids, such as citrate, can inhibit silica adsorption. This implies that in natural environments, organic molecules could compete with silicate for adsorption sites, thereby maintaining DOC adsorption even in silica-rich conditions. Furthermore, other studies have shown that the presence of silica does not significantly hinder the adsorption of organic acids on iron oxides. For example, citrate adsorption on goethite is effective even in the presence of competing anions⁴¹. Modeling studies indicate that citrate forms inner-sphere complexes with iron oxide surfaces, and these interactions persist despite changes in surface charge properties⁴². Therefore, it seems likely that the inclusion of silica at concentrations representative of ancient oceans might not substantially alter the capacity of iron oxides to interact with DOC.

Interestingly, the higher silica concentrations of Precambrian oceans could promote the formation of hematite over goethite from ferrihydrite, which aligns with the mineralogical observations in our study. Previous studies have found that silicate species stabilize ferrihydrite by adsorbing onto its particles, thus slowing its transformation into more crystalline iron oxides. Then, when transformation does occur, silicate species would tend to favor the formation of hematite over goethite^{43,44}.

Our conclusions are further supported by elemental mapping (SEM-EDS) of the synthesized hematite (see Fig. 3 of this response), confirming the presence of both carbon and silicon. Importantly, under the analytical settings used (measurements performed at 8 kV, coated with a layer of gold 10 nm thick), the SEM-EDS signal is *not* exclusively surface-based, the spatial distribution demonstrates that both elements are associated with the hematite particles.

Figure 3: (A) SEM image showing hematite pseudo-spheres synthesized under conditions of pH 8, 2.13 mM SiO₂, and 4 mM DOC. (B) Carbon and Silicon elemental maps showing the distribution of both elements relative to the hematite spheres.

In summary, while we acknowledge the potential influence of higher silica concentrations on iron oxide properties in ancient oceans, our new experimental data demonstrate that the impact on DOC adsorption and isotopic fractionation is minor under the conditions tested. We have updated the manuscript to include these new experimental findings and their outcome on our results (e.g., Supplementary Discussion L2209–L2232; Figs. ED14 and ED127). Nonetheless, the consistency of trends across all experimental conditions reinforces the robustness of our proxy and supports its applicability for reconstructing past environmental conditions.

We thank the Reviewer (as well as Reviewer 1) for bringing this important aspect to our attention. The inclusion of these additional experiments strengthens the manuscript and provides a more comprehensive understanding of the interactions between silica, iron oxides, and DOC under conditions representative of ancient oceans.

Comment 4: Reviewer Comment: *I am also concerned by the choice of DOC types. Whilst the authors do a decent job of defending these, their proxy is predicated on the fact that marine DOC at all points in geologic time can be described by a mixture of modern-marine like, cyanobacterially derived, and terrestrial soil-like DOC end-members. This is a very bold assumption. Even in the modern ocean we are still unable to fully characterize marine DOC, and one of the most important components of the DOC pool, namely the enigmatic long-lived DOC (increasingly implicated in climate), is probably not represented very well by the molecular composition of any of their end-members. Studies indicate that at least a part of this DOC pool is rich in carboxyl groups and is highly aromatic (the so-called “CRAM” component—carboxyl-rich alicyclic molecules).*

Response: We appreciate the Reviewer’s insightful comments regarding our choice of DOC types in this study. We acknowledge that marine DOC is a complex and heterogeneous mixture of organic molecules, and that fully characterizing it remains a significant challenge even in the modern ocean. Specifically, we recognize that the long-lived DOC pool, which is increasingly implicated in climate processes, includes components like carboxyl-rich alicyclic molecules (CRAM) that may not be fully represented by the molecular composition of our selected end-members (note also our revised discussion regarding the differences between “labile”, “semi-labile”, “semi-recalcitrant”, and “recalcitrant” DOC throughout the text, as suggested by Reviewer 5). As detailed in the original manuscript (under “Possible end-member sources” L3139–L3169), our proxy is indeed based on a mixture of modern marine-like DOC (M-DOC), cyanobacterially derived DOC (C-DOC), and terrestrial soil-derived fulvic acid (FA) to approximate the range of DOC compositions that could have been present throughout geologic time. While we agree that this is a simplification, we believe it is a practical and necessary approach for several reasons.

First, we included FA in our experiments partly to represent carboxyl-rich compounds like CRAM. FA is rich in carboxyl groups, similar to the CRAM component mentioned by the Reviewer. Studies have shown that such compounds are significant constituents of the refractory and semi-refractory DOC pool⁴⁵. Although CRAM possess alicyclic (non-aromatic ring) structures, by including FA in our experiments, we aim to additionally account for highly aromatic and carboxyl-rich molecules that are important in the long-lived DOC pool. Thus, while not a perfect analog for CRAM—which is essentially impossible to isolate from natural marine DOC in the concentrations required for our experiments—we include FA precisely to probe the importance of carboxyl groups (and aromatic rings) on Fe-OC signals.

Second, given the immense diversity of DOC molecules, replicating the exact composition of marine DOC in laboratory experiments is currently impractical. Our selection of DOC

types captures a spectrum of molecular characteristics—ranging from modern marine organic matter to cyanobacterial and terrestrial inputs—that are likely to have been significant throughout Earth’s history. Our selected DOC types encompass a variety of functional groups and molecular structures, which allows us to investigate the interactions between iron minerals and different types of organic matter. This approach provides insights into the general behavior of DOC during iron co-precipitation, even if it does not represent every specific compound present in the marine DOC pool.

Third, our experiments demonstrate that, while different DOC types can influence the extent of organic carbon loading onto iron minerals, the overall trends with increasing DOC concentration are consistent (e.g., Figs. ED18–ED19). This suggests that, while additional DOC sources may exhibit unique Fe-OC calibration curves not captured here, their quantitative importance is unlikely to qualitatively change the interpretation of our record for two reasons:

(i) Our three experimental DOC materials exhibit an excitation-emission matrix spectroscopy (EEMS) slope ratio (S_R) range that captures most variability observed in modern coastal to continental shelf waters⁴⁶. Because S_R exhibits a strong control on Fe-OC loadings (Fig. ED23), our chosen DOC sources likely span the expected range of loadings and thus calculated DOC concentrations for a given measured wt.% Fe-OC. Although significantly higher S_R values may be possible for some DOC end members (e.g., highly photo-bleached material), S_R estimates for sources such as CRAM should be representative within the range used (e.g., Ref⁴⁷).

(ii) Our reconstructed Earth history DOC concentration and $\delta^{13}\text{C}$ trends are largely insensitive to the chosen calibration curve (i.e., the six $f^i(t)$ scenarios considered in the manuscript) as well as the presence and concentration of dissolved silica in the “cyanobacteria only” scenario (Fig. ED127). Specifically, average predicted DOC concentrations never differ at any point in our record by more than a factor of ≈ 2 between scenarios, and results for all scenarios are always statistically identical (Fig. ED127A). Similarly, average predicted $\delta^{13}\text{C}$ values in the Phanerozoic never differ by more than $\approx 8\%$ between scenarios, and results for all scenarios are again statistically identical (Fig. ED127B). Although larger differences up to $\approx 15\%$ can exist between scenarios in the Proterozoic (i.e., between “cyanobacteria only” and “soil humics only”), a general rise toward modern values starting in the Cambrian is observed for all scenarios. Thus, while incorporation of additional end-members not considered here could lead to small quantitative shifts in our record, this is unlikely to impact the general interpreted trajectory of DOC concentrations and ^{13}C compositions through geologic time.

In summary, we agree with the Reviewer that natural marine DOC is certainly molecularly more complex than is possible to capture in laboratory experiments, but we attempted to be as comprehensive as possible with our choice of DOC end members. Performing additional experiments with an expanded range of DOC sources constitutes an excellent target for follow-up studies, which we are currently beginning to perform. Still, in light of the Reviewer’s comment (and that of Reviewer 5, below), we have added text to the main manuscript to better articulate the complexity of marine DOC and the limitations of our approach. Specifically, we now say (added text in italics):

L30–L36: Estimates suggest 30% of deep-ocean DOC forms via the microbial carbon pump over decadal timescales (*so-called “labile”, “semi-labile”, and “semi-refractory” DOC*) whereas the remainder has been circulating subject to slow respiration with a residence time of $\sim 30,000$ yr (i.e., 30 ocean overturning cycles; *so-called “refractory” to “ultra-refractory” DOC, which includes carboxyl-rich alicyclic molecules*)^{48,49}. Given the long residence time of *this refractory component*, small changes in consumption rates can drive large perturba-

tions in DOC reservoir size—and thus atmospheric CO₂ levels (pCO₂)—over ~10³–10⁵ yr timescales^{18,19}.”

We appreciate the Reviewer’s thoughtful feedback and hope that our revisions address the concerns raised.

Comment 5: Reviewer Comment: *Furthermore, I am concerned that the experiments do not capture how freshly coprecipitated amorphous iron (oxyhydr)oxides age and transform into more crystalline phases (like goethite), and how this affects the (re)distribution of organic carbon between the solid and solution.*

Response: We appreciate the Reviewer’s concern regarding the aging and transformation of freshly coprecipitated amorphous iron (oxyhydr)oxides and its impact on the distribution of organic carbon between the solid and solution phases (see also our response to Reviewer 1’s comment at Line 2167, above). Understanding these processes is indeed crucial for assessing the validity of our proxy and its applicability to natural settings.

In our study, we specifically designed our experiments to mimic natural iron oxide formation and transformation processes under environmentally relevant pH conditions. In both natural environments and in our laboratory experiments, the formation of crystalline iron oxides like goethite and hematite proceeds through the initial precipitation of a poorly crystalline precursor phase, ferrihydrite. Over time, this ferrihydrite undergoes dissolution and re-precipitation processes, transforming into more crystalline iron oxides such as goethite and hematite¹³. This mechanism closely mirrors the natural aging and transformation processes that occur in marine environments where iron ooids form. Interestingly, our ongoing Mössbauer spectroscopy analyses (not part of the current manuscript) indicate that even in modern iron ooids from Panarea Island (Italy), which are presumably less than 10,000 years old, the precursor ferrihydrite phase has already fully transformed into nanocrystalline goethite. This suggests that the transformation from ferrihydrite to more crystalline iron oxides occurs relatively rapidly in natural settings, supporting the relevance of our experimental timescales.

We acknowledge that the transformation from amorphous to crystalline phases can potentially affect the distribution of Fe-OC. However, our experiments are designed to capture these dynamics. By replicating the natural processes of iron oxide formation and transformation, we aim to accurately assess how these transformations affect the distribution and preservation of organic carbon within iron oxides. In summary, we believe that our experimental design effectively captures the key processes of iron oxide aging and transformation that occur in natural environments. We have added clarifications in the manuscript (see Supplementary Discussion, L2393–L2550) to better explain how our experiments mimic natural aging and transformation processes of iron oxides and how this influences organic carbon preservation. We appreciate the Reviewer’s insight and hope that this explanation addresses the concern raised here.

Comment 6: Reviewer Comment: *I also think it’s a problem that the authors assume the organic carbon associated with the iron oxides in iron ooids results from uptake of only dissolved organic carbon in the fluid from which the minerals precipitated. This assumption is valid for their experiments because their experiments contained only DOC, but in natural settings, it is well documented that iron oxides with associated organic carbon arise via the uptake of DOC but also incorporation of particulate organic carbon (POC).*

Response: We appreciate the Reviewer’s concern regarding our treatment of Fe-OC as solely reflecting the uptake of DOC during mineral precipitation. We agree that in natural settings, iron oxides can associate with organic carbon through both the uptake of DOC and the incorporation of particulate organic carbon (POC), including detrital inputs as well as microbial cells,

biofilms, or extracellular polymeric substances (EPS). However, several lines of evidence from our study suggest that the contribution of POC to the organic carbon content in iron ooids is minimal (see also our response to, e.g., Reviewer 1's comments 2 and 5). We detail these lines of evidence below, organized from strongest to weakest:

First, our new Raman imaging results (main text Fig. 1, see also our response to Reviewers 1 and 3, above) reveal a homogeneous lateral distribution of organic matter within all analyzed ooids. This homogeneity is consistent with the co-precipitation of DOC during iron oxide formation and does not support the presence of discrete POC particles embedded within the ooids. If POC were a significant contributor, we would expect to see heterogeneities or localized concentrations of organic matter corresponding to the POC particles, which is never observed in any ooid.

Second, we have conducted extensive petrographic analyses to screen for the presence of POC and diagenetic alterations that could affect our interpretations (Supplementary Discussion, Figs. ED25–ED123). Our analyses did not reveal any evidence of POC incorporation within any iron ooids studied here. The absence of such evidence supports our assumption that the organic carbon associated with the iron oxides originates primarily from DOC.

Finally, the environmental conditions under which iron ooids form are likely not conducive to the incorporation of significant amounts of POC. Iron ooids typically form in high-energy, shallow marine environments where fine particulate matter, including fine detrital POC, is winnowed away by currents and wave action. This results in a sedimentary environment dominated by coarser grains, with minimal fine particles available for incorporation into the growing ooids other than dissolved solute (i.e., DOC). This setting reduces the likelihood that POC would be incorporated into the iron oxides during ooid formation. This is supported in the modern by global maps of POC content, which suggest low POC in the environments that favor ooid formation⁵⁰.

Nonetheless, we acknowledge that in some natural settings, iron oxides can associate with fine-grained POC. To account for this possibility, we have revised the manuscript to clarify that our reconstructed DOC concentrations should be considered as maximum estimates. Any contribution of POC to the organic carbon content would lead to an overestimation of the DOC concentration derived solely from co-precipitation with iron oxides. Specifically, we now add following text:

L205–L207: *“Importantly, because iron ooids capture both labile and recalcitrant DOC and may be contaminated by trace incorporation of POC, reconstructed [DOC]* should be treated as a maximum value and not equal to that of the deep ocean.”*

We appreciate the Reviewer's insightful comment and believe that our revisions adequately address the concern.

Comment 7: Reviewer Comment: *Overall, I think that this manuscript tries to do too much in one go. To instigate a new proxy idea, develop this, calibrate this, ground-truth this, then use the proxy in earnest to infer major states of ecological and biogeochemical evolution is, in my opinion, too much. To be convinced that we have a proxy for ocean DOC through time, I would need to see calibration against laboratory experiments that fully represent iron-carbon dynamics in natural settings, performed under past ocean conditions, with additional DOC sources that might more closely match the important fractions of ocean DOC.*

Response: We appreciate the Reviewer's thoughtful feedback and concerns regarding the scope of our manuscript. Still, we view this as a strength of the current manuscript and not a weakness—here, we go above and beyond what is typical of studies describing deep-time proxy records in

terms of experimental calibration, petrography, range of analyses, and attention to detail. We recognize that developing and applying a new proxy is a significant undertaking, and we have carefully considered the feasibility and necessity of each component included in our study. We briefly summarize the main points related to each of the Reviewer's specific concerns below:

Use of DOC sources that closely match important fractions of marine DOC: Recognizing the complexity and diversity of marine DOC, we have included multiple DOC sources in our experiments to attempt to accurately represent the important fractions present in natural marine environments:

- *Modern-marine analog DOC from microcosm cultures containing diatoms and marine heterotrophic bacteria (termed "M-DOC"):* This represents eukaryote-dominated communities and simulates the labile and semi-labile fractions of marine DOC produced by phytoplankton and heterotrophic bacterial activity.
- *Cyanobacterial biomass leachate (termed "C-DOC"):* This represents prokaryote-dominated communities and serves as an analog for DOC produced by cyanobacteria, which are significant contributors to marine DOC, especially in the Proterozoic.
- *Dissolved fulvic acid (termed "FA"):* This represents modern terrestrial humic substances and provides a proxy for refractory DOC components that may have been present due to terrestrial inputs or recalcitrant marine organic matter.

By incorporating these diverse DOC sources, we aim to capture a broad spectrum of DOC compounds present in marine environments, enhancing the relevance and applicability of our experimental results to natural settings (see Supplementary Discussion, L1984–L2041; Figs.ED23). As described above, although this approach can never fully constrain every compound present in DOC (e.g., CRAM), these end-members were carefully chosen to span the appropriate range of molecular properties that we expect will impact Fe-OC loadings (e.g., slope ratio, S_R ; Ref.⁴⁶).

Ground-truthing and validation of the proxy: We have validated our proxy using geological samples from multiple time periods and mineralogies. Specifically, we compared time-equivalent formations from the Ordovician period containing both goethite and hematite minerals. Our findings show that reconstructed DOC concentrations from both mineralogies are statistically identical within uncertainties, supporting the reliability of our proxy across different mineral phases (see Supplementary Discussion, L3125–L3194; Figs. ED124 and ED127).

Furthermore, we have now conducted several new analyses in the revised version to strengthen this validation, particularly under ancient ocean conditions. First, we utilized Raman microspectroscopy to assess the spatial distribution of organic carbon within iron ooids. The homogeneous distribution of organic carbon strongly indicates DOC incorporation during abiotic mineral precipitation rather than from localized microbial activity (main text, L144–L161; methods, L634–L682; Fig. 1). Second, to rule out potential microbial contributions, we performed environmental DNA extraction and 16S rRNA gene sequencing on modern iron ooids. The results indicate minimal microbial DNA, and known iron-oxidizing bacteria were not detected, reinforcing our interpretation that the organic carbon is derived from marine DOC (main text, L155–L160; methods, L688–L724; Supplementary Discussion, L2478–L2501; Fig. ED24). Third, following the Reviewer's suggestion, we conducted new hematite synthesis experiments in the presence of dissolved silica, which is particularly relevant for Precambrian oceans due to hypothesized higher silica concentrations at that time. The inclusion of silica in our hematite synthesis experiments showed that, while silica slightly influences Fe-OC loadings and $\delta^{13}\text{C}$ values, the overall trends remain consistent, supporting the robustness of our calibration under varying conditions (Supplementary Discussion, L2209–L22232; Fig.ED14).

Focusing the scope of the manuscript: In response to the Reviewer's concern about the manuscript attempting to do too much, we have revised the manuscript to focus more sharply on the development, calibration, and validation of the DOC concentration proxy (see also our response to Reviewer 1, above, regarding the removal of speculative language related to ε_p and OC burial through time). Specifically, we have streamlined the discussion and interpretations to concentrate on the primary findings related to [DOC] reconstruction, reducing speculative discussions about broader organic matter burial and carbon isotope composition through time that may extend beyond the current data.

Conclusion: We strongly believe that the work presented in the manuscript adequately addresses the development, calibration, and validation of the new proxy for reconstructing oceanic DOC through time. By conducting experiments under conditions that reflect natural iron-carbon dynamics, using DOC sources representative of important marine fractions, and validating our findings with geological samples, we have provided a comprehensive foundation for the proxy's application. We are confident that these efforts sufficiently address the Reviewer's concerns and demonstrate the validity and potential of the proxy. We appreciate the Reviewer's feedback, which has helped us refine and strengthen our manuscript.

Reviewer #5

Comment 1: **Reviewer Comment:** *I was hugely interested to read this paper and the novel proxy for ocean dissolved organic carbon (DOC) concentrations and hence global reduced carbon inventory. In addition, in trapping organic compounds in an accrediting iron oxy-hydroxide (goethite) or oxide (hematite) as sedimentary ironstones, there is the obvious potential to reconstruct past changes in global $\delta^{13}\text{C}$ perhaps free of diagenetic constraints (as the authors indeed do). The authors are right in their argument for the value of a paleo DOC proxy, although they tend to overplay a little this argument at the start and cite published hypotheses for how changing marine DOC reservoirs could have impacted atmospheric $p\text{CO}_2$ as if this is how the system actually works ... but they are only hypotheses (many of which I happen to disagree with). (A minor nuance, as the important point is to generate new data to test the ideas.)*

Response: We thank the Reviewer for their interest in our work and for their general agreement that reconstructing past DOC concentrations and $\delta^{13}\text{C}$ values is an important topic of study. However, we urge caution with the statement “free of diagenetic constraints”, and we apologize for the misunderstanding if the Reviewer interpreted our manuscript as implying that iron ooids are not subject to diagenesis. As described in detail below, iron ooids—like any other geologic archive from which one can reconstruct environmental proxy signals—can be subject to diagenesis. Throughout our manuscript (particularly in the Supplementary Discussion), we therefore develop a multi-faceted experimental and petrographic approach to screen and remove samples that are obviously diagenetically altered. However, this does not guarantee that all included samples are entirely free of diagenetic signals or alterations—this can *never* be guaranteed in *any* deep-time proxy record. This becomes important below, for example when the Reviewer interprets an increase in proxy signal variance as evidence that iron ooids are capturing a more variable subset of the DOC signal (see our detailed response to the Reviewer’s Comment 3). In summary, this is an over-interpretation of the proxy record that is not supported by the data.

Similarly, we believe that our introduction is not overplaying the role of DOC dynamics, and we apologize if it was interpreted this way. Rather, our intention in setting up the problem is to summarize the existing literature and hypotheses—which, like the Reviewer, we disagree with! For example, quoting from Rothman et al. (2003) *PNAS*: “Concurrent with the decline of [carbonate $\delta^{13}\text{C}$] would be an *increase in CO_2 levels resulting from the increased oxidation of organic carbon*. These changes would have been most significant at low latitudes, where the decrease in temperature due to advancing ice sheets and the surface area over which the change occurred would have been greatest.” Similarly, quoting Johnston et al. (2012) *Nature*: “Such an *injection of carbon [from a large pulse of organic carbon oxidation] would also significantly raise atmospheric $p\text{CO}_2$ or $p\text{CH}_4$ and consume existing oxidant reservoirs.*” Finally, the coupling between DOC remineralization and $p\text{CO}_2$ dynamics forms the basis of the model developed by Peltier et al. (2007) *Nature* (see, for example, flux arrows in their Fig. 2 as well as model setup in their Eqs. 1-3). All of these publications inherently link (D)OC respiration to increasing $p\text{CO}_2$ and *vice versa* to explain Proterozoic carbon isotope excursions.

Thus, the text in paragraphs 2 and 3 of our main text is simply to summarize the current state-of-the-art and articulate the contrasting range of opinions that exist in the literature (see also our response to Reviewer 1, above, and praise from Reviewer 3, above, regarding the setup and description of the problem). Importantly, these opinions do not reflect the position that we take here. As the Reviewer rightly points out, the main goal of our current manuscript is to generate actual data constraints to solve this problem. It is for this reason that, after introducing this current state-of-the-art, we quickly pivot to discussing how our study will advance the field.

This begins with the text, “Such disagreement on the geologic history of marine DOC results from a dearth of direct records. To address this, we developed a proxy based on...” (L66).

Comment 2: **Reviewer Comment:** *The main body of the paper itself is well written but could be better illustrated. I am more concerned at the extent of the Extended Data and SI discussion (yet in getting on for 200 pages, critical data are still missed out).*

Response: We thank the Reviewer for their positive assessment of our writing. Regarding the illustration of the main text: we have now added an additional figure that illustrates Raman microspectroscopy results showing the fidelity of our proxy (Fig. 1; see our response to Reviewers 1 and 4, above). The main text now contains three figures, which follow the flow of logic: first, show that the proxy is robust (Fig. 1); next, show the raw data as it was generated (Fig. 2); finally, show the interpretation of the data within an Earth-history context (Fig. 3). We hope the Reviewer agrees that this flow of logic and figures nicely articulates the main points of our text (also articulated in the main-text sub-headings). Given the spatial constraints imposed by the journal, we are unfortunately unable to provide any additional figures within the main text.

We are not sure what the Reviewer is referring to when they say “critical data are missing”. We include several Extended Data Tables showing (i) DOC fluorescence parameters, (ii) Fe-OC loading calibration curve statistics, (iii) modern ooid ^{14}C ages, and (new to this revision; see our response to Reviewer 1, above) extracted DNA (iv) primers and (v) PCR thermal proviels (Tables ED1-ED5). Additionally **all raw data** generated for all experiments and all measurements of geologic materials are **provided as supplementary materials** (i.e., as attached Excel tables; Supplementary Data 1-6). Finally, **all code** required to perform all calculations and generate all interpreted results (i.e., Monte Carlo mixing model) is additionally **provided as Matlab .m files in the Supplementary Code**. It is therefore not possible to include any additional data since everything is already included. We apologize if the Reviewer could not access the Supplementary Data and Supplementary Code, but we assure them that all critical data are indeed available. This is an important point, as we strongly believe in openness and data transparency (additionally, an open-data policy is required by our funding agencies).

This seems a very poor balance for a short format paper and really, much of the ED and SI should be aired fully as possible and in detail, as a paper in its own right. This also leaves the main paper too short and some of the ED figures, such as calibration curve, are central to the main text and should be there.

Response: We agree with the Reviewer that the length of Supplementary Discussion, as well of the number of Extended Data Figures, is significantly longer than is typical for most publications. However, we view this as a strength of our approach and not a weakness. This results from the fact that we are developing and applying an entirely new proxy system. To do so responsibly, we believe the following three aspects *must* be thoroughly tested and articulated, included limitations and potential weaknesses:

- (i) experimental validation of our Fe-OC loading proxy (i.e., “does the proxy work how we say it does?”);
- (ii) description of all included formations, including extensive petrographic screening (i.e., “do the rocks record the signals that we say they do?”); and
- (iii) mathematical derivation and details of our interpretive model, including thorough discussion on the appropriateness and potential impact of all necessary assumptions (i.e., “do the data mean what we say they mean?”)

Our Supplementary Discussion and Extended Data figures can be broadly divided into these three categories. Although long, we have attempted to ease the reader's ability to navigate these sections by including several sub-headings and parallel figure structure. That is, Figs. ED25-123, which form the majority of our Extended Data, are essentially replicates of the same petrography figure structure but applied to unique samples. Similarly, the majority of the Supplementary Discussion text consists of formation descriptions and geologic contexts for each sampled formation.

Additionally, we **now include a Table of Contents in the Supplementary Discussion** in order to further improve the reader's ability to decipher and interpret the logic and background that goes into this manuscript. When comparing to other studies in the general field of deep-time Earth history, we strongly believe that ours is the most responsible approach (c.f., studies which omit detailed petrographic information and microscopy figures, detailed model derivations, etc.). Finally, we briefly note that—in agreement with the Reviewer's point about the calibration curve—the experimental validation of this manuscript is currently being expanded and will form the basis of a follow-up manuscript focusing on the role of DOC molecular properties on Fe-OC preservation mechanisms. In an ideal world, such a manuscript would be published before the current one, but the order and timing of our approach to this research topic has precluded this. Nevertheless, we include all details here for full transparency and openness.

Comment 3: Reviewer Comment: *“DOC” vs. LDOC + SLDOC + SRDOC + RDOC: The paper falls down overall, and in particular in respect of the DOC proxy, by failing to distinguish between the fractions of different lability (assuming that is even a word). Only “DOC” as a single entity is considered, but the modern marine community has long recognized that the overall role and dynamics of DOC can only be understood by considering the production and fate of the constituent molecules, which range across an enormously wide spectrum and which are grouped for convenience of description and numerical modeling into e.g., labile DOC (LDOC), semi-labile DOC (SLDOC), semi-refractory (SRDOC), and refractory (RDOC). (Although often only a subset of these are considered and LDOC is usually ignored.)*

Response: We agree with the Reviewer that DOC dynamics are inherently complex and that DOC contains several “pools” or “reservoirs” that can be empirically or analytically defined on a spectrum from labile to recalcitrant. This is indeed a well-studied topic, and this interpretation is supported by several decades of molecular and isotopic work, some of which we cite in our manuscript (e.g., Refs.^{18,19,49,51–54}). It was never our intention to imply that *all* DOC can be treated equally, whether today or in the geologic past. Based on the Reviewer's suggestion, we have updated the main text and Supplementary Discussion to better articulate this understanding. Specifically, we now say (added text in italics):

Main text:

- L30–34: “Estimates suggest 30% of deep-ocean DOC forms via the microbial carbon pump over decadal timescales (*so-called “labile”, “semi-labile”, and “semi-refractory” DOC*) whereas the remainder has been circulating subject to slow respiration with a residence time of $\sim 30,000$ yr (i.e., 30 ocean overturning cycles; *so-called “refractory” to “ultra-refractory” DOC, , which includes carboxyl-rich alicyclic molecules*)^{48,49}. Given the long residence time *of this refractory component*, small changes in consumption rates can drive large perturbations in DOC reservoir size—and thus atmospheric CO₂ levels (pCO₂)—over $\sim 10^3$ - 10^5 yr timescales^{18,19}.” [**Note:** here we cite a literature estimate that 70% of total deep-ocean DOC is refractory to ultra-refractory; however, we are agnostic

to this number. In our understanding, this is an area of active research with conflicting viewpoints, e.g., Follett et al. (2014) *PNAS*.]

- L39–43: “Ref.²¹ hypothesized that slow particle sinking prior to the evolution of multicellular and biomineralizing organisms⁵⁵, combined with an anoxic deep ocean^{56,57}, prompted a stronger microbial loop, slower respiration at depth, and *accumulation of a long-lived, recalcitrant DOC reservoir* that was $\sim 10^2$ - $10^3 \times$ its modern size.”
- L205–207: “*Importantly, because iron ooids capture both labile and recalcitrant DOC and may be contaminated by trace incorporation of POC, reconstructed [DOC]* should be treated as a maximum value and not equal to that of the deep ocean.*” [this statement is in agreement with the Reviewer’s correct observation that continental shelf waters will contain higher DOC concentrations than the open deep-ocean—we now express this explicitly.]

Supplementary Discussion:

- L2518–2519: “*...here ignoring water-column pre-aging of semi-recalcitrant to recalcitrant marine DOC for simplification*” [**Note:** this statement refers to a back-of-the-envelope calculation and the assumption of no pre-aging is later relaxed; see our below response.]
- L2535–2540: “*Although significantly younger than modern deep-ocean DOC, this age is similar to measured ¹⁴C ages of modern surface-ocean DOC, which includes a mixture of material ranging from labile to recalcitrant (e.g., Refs.^{48,49}). Thus, ¹⁴C ages of modern ooids appear consistent with the interpretation that labile, semi-labile, semi-recalcitrant, and recalcitrant DOC compounds are continuously co-precipitated.*”

That said, as articulated in detail in our response below, we strongly caution *against* over-interpreting the granularity and level of detail that can be extracted from our Fe-OC proxy, particularly the radiocarbon (¹⁴C) ages of modern ooids used as calibration. That is, there remains too much uncertainty on both the timing of modern ooid formation as well as the assumption of continuous ooid accretion to confidently interpret Fe-OC ¹⁴C ages as reflecting the age of DOC when it became entrained in the iron oxides (i.e., the labile vs. recalcitrant nature of this DOC). This becomes important below, as the Reviewer appears to base a significant amount of their argumentation on the apparent conclusion that ooid-bound Fe-OC *cannot* be pre-aged in the water column and thus *must* represent labile to semi-labile compounds. This is not supported by our data; we discuss this in detail below.

The importance of this is because the large reduced carbon inventory cited by the authors of some 660 PgC, is dominated by RDOC (ca. 630 PgC), with minor contributions from SRDOC (14) and SLDOC (6) and almost from LDOC (with the balance made up by ultra-refractory components) (e.g., see: DOI: 10.1146/annurev-marine-120710-100757). RDOC is relatively uniform in concentration throughout the ocean, and because it dominates the total DOC pool at the surface, its presence suppresses the spatial variability in total DOC concentration at the surface.

Response: We thank the Reviewer for this summary; this accurately reflects our own understanding of modern marine DOC dynamics and reservoir sizes. We note that the publication referenced by the Reviewer here (Hansell (2013) *Annual Review of Marine Science*) was already cited in our original manuscript. Nevertheless, as described above, we now explicitly articulate the differences between these DOC pools, and we cite this publication more extensively throughout the main and supplemental text.

*In the SI, the authors present radiocarbon measurements on DOC trapped within modern iron ooids in an actively forming environment, and **strongly argue** that the 5,200 and 3,600 ^{14}C yr ages reflect the mean age of formation of the ooids which are in turn consistent with estimated dates for the start of formation.*

Response: We apologize for any confusion that has arisen from our discussion of the modern ooid bulk ^{14}C ages. Here, it is certainly not our intention to “strongly argue” for any such interpretation. Rather, this section of the Supplementary Discussion simply highlights a back-of-the-envelope calculation that we use to **argue against the possibility that all ooid-bound Fe-OC is derived from hydrothermal vent fluids**, which could be a concern given the formation environments of these grains. That is, if modern ooid-bound Fe-OC were ^{14}C -free, then our proxy would be invalid. However, this is not observed.

Given the uncertainty on both the timing of modern ooid formation as well as the assumption of continuous ooid accretion, any *quantitative* conclusions regarding the pre-aging of DOC when it is incorporated into iron oxides—and thus the labile to recalcitrant nature of these compounds—should be interpreted with *extreme* caution. Again, we apologize if this was not clear, and we have thoroughly updated this section of the Supplementary Discussion to more explicitly draw attention to the fact that this is a back-of-the-envelope calculation and should not be interpreted quantitatively (see specific text in our response below).

*In contrast, the lifetime of RDOC, which dominates the total DOC pool, is perhaps 16,000 years (or older if you include ultra-refractory DOC) (the authors not unreasonably reasonably cite 30,000 years). What is trapped in the modern ooids then **must** be short-lived SLDOC and/or SRDOC fractions which is consistent with mean DOC ^{14}C age being comparable to mean ooid age.*

Response: While we thank the Reviewer for raising this point, we disagree with their interpretation that ooid-bound Fe-OC *must* be semi-labile to semi-refractory; the statement that mean DOC ^{14}C age is comparable to mean ooid age is an over-interpretation of our data. We again apologize for any confusion.

Rather, as stated above, the main conclusion of this section is that Fe-OC is not derived from ^{14}C -free hydrothermal sources but rather from the water column. Although not quantitative, it then becomes an interesting exercise to compare ^{14}C -based reconstructions of ooid formation initiation with independent estimates (i.e., as discussed in Refs.^{30,31}). In doing this back-of-the-envelope calculation, we estimate that DOC *could* have been pre-aged by up to 2400 yr at the time of entrainment. Interestingly—although certainly younger than the ultra-refractory DOC component as the Reviewer rightly points out—this number is remarkably consistent with measured DOC ^{14}C estimates in the top ~ 100 m of much of the ocean (e.g., North Pacific and North Atlantic profiles from Ref.⁵⁸). Such agreement *might* argue in favor of ooid-bound Fe-OC accurately capturing the total DOC reservoir (which, in shallow waters, would include labile, semi-labile, semi-refractory, and refractory DOC). However, we stress that this calculation is subject to several large uncertainties and should be interpreted as “being consistent with” rather than “supporting” this conclusion.

In summary, because there is no evidence to the contrary, we interpret our Fe-OC proxy as incorporating the bulk DOC pool that is present in continental shelf waters. We agree with the Reviewer that the concentration of this material is greater than the concentration of bulk DOC in the open deep-ocean, and we now explicitly state this in the main text (L205–207). Furthermore, we thoroughly assess the possibility that subsets of the DOC pool may have different affinities for co-precipitation in Fe-OC, and we incorporate this uncertainty when

performing our Monte Carlo mixing model (see our response to the Reviewer's comment 4, above, as well as Supplementary Discussion Section 6, particularly Section 6.4 on sensitivity tests and assumption validation).

We have added the following Supplemental Discussion text to articulate this point:

L2531–2542: “Still, these ages are higher than minimum formation timescale estimate at each site (i.e., 8.7 ka and 4.5 ka for Panarea Island and Mahengentang Island, respectively). This could result from some combination of (i) minor detrital DOC contributions or (ii) pre-aging of DOC in the water column prior to incorporation into iron ooids. If the latter explanation is true, this would imply that DOC is pre-aged by ~1700–2400 years upon incorporation into ooids. Although significantly younger than modern deep-ocean DOC, this age is similar to measured ^{14}C ages of modern surface-ocean DOC, which includes a mixture of material ranging from labile to recalcitrant (e.g., Refs.^{48,49,59}). Thus, ^{14}C ages of modern ooids appear consistent with the interpretation that labile, semi-labile, semi-recalcitrant and recalcitrant DOC compounds are continuously co-precipitated. Nevertheless, given the large and unknown uncertainties associated with each assumption in this estimation (i.e., time of ooid formation initiation, even Fe-OC content distribution, and continuous accretion), such results should be interpreted cautiously.”

Finally, here we clarify that—if Fe-OC is indeed capturing bulk DOC signals—then one *should not expect* to observe Fe-OC that has been pre-aged by at least 16,000 and up to 30,000 ^{14}C yr (We are not sure if this was the Reviewer's suggestion, but we clarify regardless for completion). That is, even in the modern deep ocean, such old components do not form the majority of the bulk DOC pool. This is evidenced by bulk deep-ocean DOC profiles that are consistently described by ^{14}C ages of ~5,000–6,000 ^{14}C yr (i.e., $F_m \sim 0.5$; see Ref.⁴⁹ and all points from ≥ 1000 m water depth in Ref.⁵⁸). Rather, Ref.⁴⁹ used a serial oxidation approach to argue that **a subset** of this deep-ocean refractory DOC pool is aged for up to 30,000 ^{14}C yr. Thus, a proxy that accurately captures deep-ocean DOC should consistently show Fe-OC pre-aging of ~5,000–6,000 ^{14}C yr; that is, equal to the bulk deep-ocean DOC radiocarbon age. Put simply, if Fe-OC ^{14}C reservoir ages older than ~5,000–6,000 ^{14}C yr are observed, then this **would imply a bias against** a subset of the total DOC reservoir (i.e., against the younger component). If observed, such old ages would indeed argue against the utility of this proxy as recording a bulk signal. Fortunately, these are never observed.

Although not necessarily relevant for the current manuscript (i.e., since iron ooids capture continental shelf, rather than deep-ocean, waters), such logic and methods validation could become important if this proxy is applied to, for example, deep-ocean ferro-manganese nodules. However, this is beyond the scope of the current manuscript.

So the proxy is not “DOC”, which is dominated by very old RDOC, but SLDOC (and/or SR-DOC). This is an important distinction because it is the dominant RDOC reservoir that creates the potential to drive global changes in $\delta^{13}\text{C}$ and $p\text{CO}_2$ and all the cited references, and this is not what the proxy is sampling if the modern data and interpretation presented in the SI is correct. Consistent with this is the observation that RDOC is old because it is not being rapidly scavenged and removed from the ocean, which supports the ooid Fe-OC as being drawn from the much more reactive and little-aged SLDOC or SRDOC pool.

Response: We agree with the Reviewer that oxidation of the recalcitrant pool exhibits the strongest potential to drive global change. However, as articulated above, the conclusion that iron ooids capture semi-labile to semi-refractory DOC is an over-interpretation of our modern ^{14}C data. Barring any strong evidence to the contrary, we interpret our ooids as capturing the range of DOC pools that are present in continental shelf waters (i.e., ranging from labile to

recalcitrant). We have now updated the main text and Supplementary Discussion to articulate this point (see quoted text above).

The concentration of SLDOC varies significantly at the ocean surface and will to a first order follow the very substantial variance that exists in primary production. Because of its short lifetime, SLDOC concentrations can change by an order of magnitude over the first 100-200 m down from the surface. The specific paleo environment, not just productive vs. unproductive, but the depth in the upper water column, is now important in the recorded Fe-OC concentration.

Response: As articulated above, because there is no valid evidence to the contrary (despite our multi-faceted approach to test this), we interpret our ooids as capturing the range of DOC pools that are present in continental shelf waters (i.e., from labile to recalcitrant). Thus, it is the *total* DOC concentration in continental shelf waters—not just that of the semi-labile pool—that matters for the Fe-OC proxy. In contrast to the order-of-magnitude change in semi-labile DOC concentration quoted by the Reviewer, bulk DOC concentrations in fully saline continental shelf waters varies by only $\pm 58\%$ ⁶⁰. Furthermore, the depth of the water column is well-constrained by sedimentological conditions required for iron ooid formation (i.e., gentle wave action; Refs.^{14,61,62}), as articulated in our main text and Supplementary Discussion (see our response to Reviewer 1, above). Given these geochemical and sedimentological constraints, we do not expect iron ooid Fe-OC to be subject to order-of-magnitude variability at a given point in time. We have tested this with time-equivalent formations at several points in Earth's history, as discussed below.

*The authors point to samples from different localities in the same time interval as supporting a global proxy, but a quick glance at the localities suggests they are paleographically proximal and/on the same *carton* [?] and likely similar paleo environments. Inclusion of (e.g., ED Figure) maps of the paleo locations would have been an important addition to the paper and would have greatly helped in this question.*

Response: We apologize for any confusion that we may have caused. The Reviewer is correct that all formations are from similar paleo environments in the sense that there exists a narrow environmental window in which iron ooids can form (i.e., marine water column, continental shelves, near wave base, subject to gentle wave/tidal motion); this is well-established based on decades of sedimentological research^{14,61,62}. This narrow range of environmental conditions is a **fundamental asset** of our approach. In contrast, other iron oxide facies—for example red beds, massive iron deposits, and banded iron formations—form under a much wider (and often unknown) range of environmental conditions and can be subject to issues such as detrital inputs with no petrographic evidence. Given this range of environmental conditions for other iron-oxide facies, if one were to build an Fe-OC record using, e.g., red beds, then one could not confidently conclude that any signals observed through time accurately represent changes in the DOC reservoir and do not instead reflect, e.g., changing detrital inputs or signals from lacustrine environments. A record built on such facies would thus be ambiguous. It is precisely **because** of the unique environmental conditions required for their formation that we focus on iron ooids—put simply, we can exclude the possibility that our record is driven solely by a change in the environments in which ooids form (see also our response to the text from Reviewer 1, who recognizes the utility of these facies for precisely this reason, but who requested more information regarding ooid formation mechanisms in the main text).

That said, we have further tested this proxy by comparing several time-equivalent formations. We emphasize that this approach goes **above and beyond** what is typical of such deep-time Earth-history proxy reconstructions (which often focus on a single formation, e.g., Ref.²⁸, or a small number of disparate formations, e.g., Ref.²³). We strongly believe that such

testing is important for accurate interpretation. To use a heuristic example, consider the possibility that one formation formed near a paleo-river mouth, which would carry terrestrially derived DOC. If we were to only measure this formation, then we would erroneously interpret it as capturing an elevated marine DOC signature. In contrast, if we compare this to a second formation that formed ~ 100 km away and does not capture the same signal, then we would conclude that one or both of these formations capture local signals that cannot be extrapolated globally. This logic forms the motivation of our time-equivalent comparison approach.

Specific to the time-equivalent comparisons included here, Jurassic, Ordovician, and Tonian formations are separated by several-hundreds to thousands of kilometers (see Fig. ED125 and corresponding text in the Supplementary Discussion). Cretaceous formations exhibit slightly less spatial separation (~ 100 km). We interpret such spatial separation as sufficient to exclude a dominant local control on our observed signals. Unfortunately, due to the sporadic nature of the geologic record, it is not possible to include several formations at each time point, with each formation separated by thousands of kilometers. These rocks simply do not exist. Ooidal ironstones are rare facies, and we have extensively scoured mineral collections (as well as performed new fieldwork) to acquire the existing sample set.

The Reviewer additionally suggests including maps of paleo localities. However, this is impractical since one would need to include a separate map for each time horizon given that these formations were deposited under different paleogeographies and thus different continental configurations. Nevertheless, we do include a figure showing Fe-OC signals as functions of both modern- and paleo-latitude to highlight that (paleo)latitude does not drive our observed signals. As correctly suggested, this figure has now been modified to be a function of *absolute* (paleo-) latitude (see our response to the Reviewer's Comment 8 and the corresponding figure, below).

It is hard to make out in the main figure – and I note that the data values are not given anywhere I could see in ED or SI which is a critical omission – ...

Response: We again apologize that the Reviewer could not find the relevant data, but we assure them that all raw data and all code to perform all calculations (i.e., Monte Carlo model) are indeed included in the Supplementary Data 1–6 and Supplementary Code.

...but the light color symbols in Figure 1 do seem to span an appreciable range of Fe-OC (and $\delta^{13}\text{C}$) which would not have been expected if the total DOC pool was being sampled, which at the modern ocean surface spans no more than a factor of 2 (e.g., DOI: 10.1038/s41467-017-02227-3). If you exclude the high latitudes and focus on typical latitudes and coastal locations representative of the ooid samples, the range in total DOC is much smaller. To me, the variance in the data seems more consistent with sampling SLDOC (and/or SRDOC) rather than the total DOC reservoir.

Response: We agree with the Reviewer that, at typical latitudes and coastal to continental shelf locations, one would expect little variability in DOC concentration—we quote the number of $\pm 58\%$ from Ref.⁶⁰ in our manuscript (L174–176, “Combined with the observation that modern [DOC] varies globally by only $\pm 58\%$ in fully saline coastal waters⁶⁰, such spatial homogeneity supports our interpretation that iron ooid Fe-OC captures globally relevant continental shelf signals”). However, the Reviewer appears to imply that *any* observed variance in our proxy data *above* this amount should be interpreted as evidence that Fe-OC is sampling a sub-set of the DOC reservoir that is more variable in its concentration than the bulk DOC. Unfortunately, this is a fundamental misunderstanding of how paleo-environmental proxies should be interpreted. This misunderstanding largely relates to error propagation, which we detail below.

First, one can think of any proxy as capturing some environmental signal following a

function:

If the Reviewer’s interpretation that observed variance *in the proxy data* **exclusively** reflects variance *in the underlying environmental signal* is correct, then this implies that the transfer function induces no variance (i.e., σ_x^2 linearly proportional to σ_y^2 such that the relative variances are equal, or $\sigma_x^2/x = \sigma_y^2/y$). However, as with all error propagation there are several reasons to expect this **transfer function to induce additional variance**—these span from analytical to environmental and diagenetic. We detail 3 such possibilities here:

- (i) **Analytical error.** All analyses are subject to some degree of analytical error, which would lead to a non-zero variance in x even if y is a constant value that is known perfectly. In the case of Fe-OC content, our analytical setup leads to typical instrument reproducibility on the order of ± 0.01 wt % ($\pm 1\sigma$) for replicate standard analyses (see Methods). Given that some of our Neoproterozoic iron ooids only contain ~ 0.05 wt % Fe-OC (Fig. 2A), analytical uncertainty alone is expected to increase proxy data variance by ~ 20 % (relative) for these samples. This is further exacerbated by ancillary uncertainty such as that associated with the microbalance when weighing sample aliquots. Thus, analytical uncertainty alone can increase the relative uncertainty in Fe-OC well beyond that expected for the environmental input (i.e., DOC concentration) without the need to invoke a more variable input term (i.e., a subset of DOC).
- (ii) **Loading curve error.** Although not relevant for interpreting variance in our raw data measurements (i.e., Fig. 2), loading-curve uncertainty will further increase variance in x when reconstructing past DOC concentrations (i.e., Fig. 3) even if y is known perfectly. For the case of Fe-OC content as a function of DOC concentration, this is manifest as a loading-curve root-mean-square error (RMSE) of 0.01 to 0.09 wt %, depending on the mineral and DOC source considered (Fig. ED18-19). Again given that some Neoproterozoic iron ooids only contain ~ 0.05 wt % Fe-OC (Fig. 2A), such loading-curve uncertainty further increases the variance in reconstructed DOC concentrations by at least ~ 20 % (relative). Thus, similar to analytical uncertainty, error in our loading curves can increase the relative uncertainty in *reconstructed* DOC concentrations well beyond that expected for *true* DOC concentration, again without the need to invoke a more variable input term (i.e., a subset of DOC). This is discussed at length in the Supplementary Material, both in our original submission and in our resubmission (see, e.g., discussion on loading curve covariance in Figs. ED20-22; the entirety of Section “Effects of co-precipitation procedural manipulations on Fe-OC carbon-loading and ^{13}C fractionation response curves” beginning on L2041; and error propagation in Section “Monte Carlo Solution” beginning on L3390).
- (iii) **Diagenesis.** Although we extensively screened our sample set to remove formations that are obviously diagenetically altered (Fig. ED25-123; Supplementary Discussion Section 4 beginning on L2540), we cannot exclude the possibility that some amount of diagenesis has occurred without leaving obvious petrographic evidence. We attempted to compile existing interpretations of metamorphic histories from the literature for all formations included here, and we state these clearly for each in the Supplementary Discussion (e.g., “below the Anchizone facies”, “below the prehnite–pumpellyite phase”, etc.) Nevertheless, it remains possible that some samples have experienced diagenetic processes such

as fluid flow which may induce some Fe-OC alteration even at $\leq 100^\circ\text{C}$. Although our screening procedure leads us to conclude that diagenesis is not the major driver of our observed signals, we cannot omit the possibility that diagenetic processes would increase the variance of preserved signals that formed at a given point in geologic time. We again emphasize that this is true for **any paleoenvironmental proxy** and is not unique to the Fe-OC record developed here, which we show below with two examples.

In summary, potential sources of proxy data error such as those articulated above (and others) preclude the interpretation that *measured* Fe-OC variance should directly reflect *true* marine DOC concentration variance—reconstructed variance will **always** be larger due to error propagation. Thus, one *cannot* interpret a larger relative uncertainty in Fe-OC content when compared to modern continental shelf marine DOC concentrations as evidence that Fe-OC is capturing a highly variable subset of the DOC pool—this is again an over-interpretation that is not supported by the data. Importantly, such potential sources of uncertainty are present **for any paleoenvironmental proxy**, particularly when applied to deep-time Earth history records. Here, we describe two well-studied examples from the literature that illustrate this point:

Example: Sulfate sulfur isotopes. As the second-most abundant anion in the ocean today, sulfate is well-mixed with a residence time on the order of millions of years. Sulfate concentrations and isotopic compositions can therefore be *confidently interpreted as constant in space throughout the ocean at a given time point*. Thus, if it were true that all variance in the proxy record—which, in this case, comprises sulfate extracted from carbonate, evaporite, and barite minerals—were *accurately and exclusively* reflecting variance in the input variable—which, in this case is dissolved sulfate $\delta^{34}\text{S}$ values—then we would expect *near-zero* variance at any point in geologic time. However, this is not observed, as elegantly shown in a recent compilation by Present et al. (2020) *Geophysical Research Letters*⁶³ and shown here:

As is clearly shown, the reconstructed sulfate $\delta^{34}\text{S}$ values can exhibit standard deviations approaching roughly $\pm 10\text{‰}$ for a given time point, particularly at older points in the Paleozoic, despite the fact that sulfate is well-mixed in the ocean. Following the Reviewer's logic, this would argue for an interpretation in which carbonates, evaporites, and barites are sub-sampling reservoirs that contain sulfate whose $\delta^{34}\text{S}$ value varies by several tens of ‰ (i.e., if all variance in x were due to variance in y). This is not supported by the data but is instead explained by several analytical and diagenetic factors that lead to higher variability in *preserved* signals relative to *true* values at that point in Earth's history; the same holds true for our Fe-OC proxy.

Furthermore, given the sparse nature of ooidal ironstones, our Fe-OC proxy is prone to

additional variance due to time binning. As a heuristic example from the sulfate record curve above, consider the possibility that one formation was sampled directly before the end-Permian mass extinction and one formation was sampled directly after this event. Because we time-average our formations into 100 Ma bins (see Supplementary Discussion Section 5.4, L3195), these two formations would be averaged into a single point, and their differences would manifest as increased uncertainty about this averaged point. Again, this should *not* be taken as evidence that this signal is sampling a smaller, more dynamic subset of the total reservoir.

Example: Lithium isotopes. Like for sulfate, lithium is well-mixed in the ocean with a residence time on the order of one million years; its concentrations and isotopic compositions can similarly be confidently interpreted as constant in space throughout the ocean. Again, if it were true that all variance in the proxy record—which, in this case, comprises lithium extracted from several carbonate facies—were *accurately and exclusively* reflecting variance in the input variable—which, in this case is dissolved lithium $\delta^7\text{Li}$ values—then we would expect *near-zero* variance at any point in geologic time. However, this is again not observed, as elegantly shown by Kalderon-Asael et al. (2021) *Nature* and shown here:

Again, as for the sulfate record, uncertainties approaching $\pm 10\%$ for a given time point are commonly observed in this record. This should again not be interpreted as reflecting a smaller, highly dynamic Li reservoir whose $\delta^7\text{Li}$ values are highly variable, but rather as the integrated influence of diagenetic, environmental, and analytical uncertainties that become propagated when building a proxy record such as this.

Similar trends can be shown for several other paleoenvironmental proxies, e.g., carbonate $\delta^{13}\text{C}$ and $\delta^{18}\text{O}$ records, as well as the crude oil and kerogen $\delta^{13}\text{C}$ records compiled in our study (see Methods beginning on L788). **In summary, due to diagenesis and error propagation, high variance in proxy records cannot be used as evidence for high variance of the input signal that a given proxy is tracking.** We therefore conclude that the Reviewer's interpretation that Fe-OC is sub-sampling only the semi-labile DOC pool is not supported by our data.

Obviously, if Figure 2B and DOC reflect ambient SLDOC and not the DOC pool as a whole, some re-interpretation is required and trends in SLDOC could be telling us something about local primary productivity (which would also be useful). I do note that the authors calibrated Fe-C to DOC/Fe where DOC is the total measured concentration of dissolved organic carbon. Even if Fe-C was SLDOC, as long as the ratio of SLDOC to total DOC remained constant through time, the calibration would work. That said, there will be little RDOC in the artificial*

cultures, and in the experiments, almost all the DOC will be SLDOC (and/or DRDOC). So DOC* should be reconstructed SLDOC (and/or DRDOC) concentrations (and not bulk DOC).

Response: As articulated above, the interpretation based on increased variance that our Fe-OC record captures only a semi-labile DOC signal is not supported by our data. Barring any strong evidence to the contrary, we interpret our ooids as capturing the range of DOC pools that are present in continental shelf waters (i.e., including labile to recalcitrant material).

Still, we agree with the Reviewer that our discussion of different DOC pools should be discussed with more nuance. For this reason, we have introduced the above-quoted text regarding the differences between labile, semi-labile, semi-recalcitrant, recalcitrant, and ultra-recalcitrant DOC reservoirs throughout our main text and Supplementary Discussion (i.e., L31–35, L40–44, L2517–2518, L2534–2539). We have additionally added a sentence to the main text explicitly stating the our reconstructed [DOC]* values should be taken as maximum estimates due to the observatoin in the modern ocean that continental shelf DOC concentrations are higher than those of the deep open-ocean (L205–207). We thank the Reviewer for raising this point.

Comment 4: **Reviewer Comment:** *Mineralogy: I find the profound offset in Fe-OC (and $\delta^{13}\text{C}$) between measurements made on iron oxy-hydroxide (goethite) vs. oxide (hematite) phases (Figure 1) very difficult to ignore. This offset is exacerbated in the reconstructed proxy values (Figure 2). The authors argue that the good match in DOC* for the ca. 450 Ma sample (again – everything is harder to interpret and understand without a data table (that I perhaps simply did not spot))...*

Response: We again apologize that the Reviewer could not find the relevant data, but we assure them that all raw data and all code to perform all calculations (i.e., Monte Carlo model) are indeed included in the Supplementary Data 1–6 and Supplementary Code.

...that mineralogy is not an issue, but yet there is no equivalent match for $\delta^{13}\text{C}$. DOC values immediately prior to and following 450 Ma are consistent with all the other low DOC* hematite hosted values which requires there to be a major event at 450 Ma for the DOC* value match argument to work. On balance, the 2 different mineralogical archives do not look consistent and I am not yet convinced by the arguments suggesting that they are.*

Response: We thank the Reviewer for raising this point. It is an unfortunate reality of the marine iron oxide record that Precambrian oceans largely precipitated hematite ooids whereas Phanerozoic oceans largely precipitated goethite ooids—this phenomenon is likely related to changes in dissolved silica concentrations over this time period (see Ref.¹⁴ for discussion). We now consider this explicitly in our revised manuscript by including hematite precipitation experiments in the presence of three silica concentrations (see our detailed responses to comments of Reviewers 1 and 4, above; Methods text L527; Supplementary Discussion beginning on L2208; and Figs. ED14 and ED22 for discussion and details on silica-containing experiments). Regardless of the underlying mechanism, this reality fundamentally leads to a record broken into two parts: an earlier part comprised of hematite ooids, and a later part comprised of goethite ooids—no amount of sample collection will remedy this fact.

Given that we must work within these constraints, the best that one can hope to do is to find time periods of overlapping mineralogy and confirm that both minerals lead to similar or, ideally, identical reconstructed values (within uncertainty). Fortunately, this test is possible within the Ordovician (485–443 Ma) given the existence of two goethite ooid formations (Aseri Fm, Estonia and Kunda Oolite Bed, Sillarou Fm, Estonia and Russia; see Figs. ED96–ED101 and Supplemental Discussion beginning on L2828) that are nearly time equivalent with two hematite ooid formations (Skovda Limestone, Gullhogen Fm, Sweden and Šárka Fm, Chechia;

see Figs. ED93-ED95 and ED102-ED105 and Supplementary Discussion beginning on L2808). Importantly, a similar test was performed on these same samples in Ref.¹⁴ when reconstructing iron-oxide $\delta^{18}\text{O}$ values, and all formations similarly yielded statistically identical results, thus confirming that they precipitated in fluids with similar or identical $\delta^{18}\text{O}$ values (i.e., seawater). (We note that Ref.¹⁴ additionally included a much younger Cretaceous hematite sample from the Hatira Fm, Israel. Although inclusion of this sample here would have increased the temporal range of our mineralogical overlap, we chose to omit it here as it is comprised of a red-bed facies and *not* hematite iron ooids and thus cannot be confidently determined as marine in origin.)

We appreciate that the Reviewer is skeptical of this approach. We agree that a record built on a single mineralogy would be ideal, but this is unfortunately not possible. We address the Reviewer's skepticism below in two parts: First, we discuss Fe-OC loadings and reconstructed DOC concentrations, which are strongly supported by the data due to overlapping results for both mineralogies in the Ordovician. Second, we address the issue of reconstructed $\delta^{13}\text{C}$, which we agree is significantly weaker than concentration reconstructions and has thus been de-emphasized in our revised version (see also our response to Reviewer 1, above, who raised a similar issue about over-interpreting our $\delta^{13}\text{C}$ record).

Fe-OC contents and reconstructed [DOC]* concentrations: Focusing on Ordovician samples, we find an average Fe-OC content for goethite ooids (Aseri and Sillarou Fms) of 0.21 ± 0.01 wt % ($n = 6$; Supplementary Data 2) and an average Fe-OC content for hematite ooids (Skovda Limestone and Šárka Fm) of 0.24 ± 0.12 wt % ($n = 6$; Supplementary Data 2). While it is true that the hematite ooid formations exhibit higher uncertainty, these average values are statistically identical. Converting these raw Fe-OC loadings into reconstructed DOC/Fe(III) ratios assuming purely M-DOC (i.e., using loading curves in Figs. ED18A and ED19A; Supplementary Code) yields values of 0.03 ± 0.001 for goethite and 0.10 ± 0.09 for hematite (omitting loading-curve co-variance for the back-of-the-envelope calculation performed here; note that a full error propagation is included in the manuscript when generating Fig. 3). Again, reconstructed DOC/Fe(III) ratios using hematite ooids exhibit larger uncertainty, but these values are statistically identical (note that the larger uncertainty for hematite is clearly visible in both the raw data in Fig. 2A as well as the reconstructed [DOC]* values in Fig. 3A).

These statistically identical reconstructions result from the fact that, at low DOC/Fe(III) ratios, loading curves for both minerals are essentially identical regardless of DOC source. The Reviewer is correct that goethite can exhibit higher loadings than hematite for a given DOC/Fe(III) ratio, but *this divergence only occurs at Fe-OC loadings that are significantly higher than those observed for any hematite sample in our record*. **Thus, given that goethite and hematite formations yield statistically identical [DOC]* reconstructions for the only time period in which overlapping samples exist, we have no reason to doubt the fidelity of this approach.** We are of course happy to hear and implement any additional validation tests that the Reviewer may suggest.

Furthermore, we are unsure what the Reviewer is referring to when they state, “[the record] requires there to be a **major event** at 450 Ma for the DOC* value match argument to work.” We believe they are referring to the Devonian aged Presles Fm, Belgium, which shows a return to low [DOC]* conditions (see Figs ED87-89 and Supplementary Discussion beginning on L2760). We agree that this formation is an outlier that cannot be explained by any petrographic or geochemical evidence. For this reason, our main text (both the original and revised versions) contains the following sentence:

L239–243: “Additionally, the Devonian-aged Presles Fm predicts lower [DOC]* and $\delta^{13}\text{C}_{\text{DOC}}$ relative to immediately older and younger formations, despite a lack of petrographic

evidence for exclusion (Supplementary Discussion, Figs. ED87-89). We therefore cannot determine if this represents a local signal or a transient return to Neoproterozoic conditions.”

Unfortunately, there is no time-equivalent goethite formation with which we can compare and validate this Devonian signal (in fact, there is a 200 Myr gap in our record stretching from the middle Jurassic to the Devonian). We therefore cannot confidently interpret the Presles Fm result within the context of an otherwise increasing-with-time temporal trend, and we honestly state this ambiguity in the above-quoted line of the main text. Nevertheless, the Presles Fm result should not be interpreted as an invalidation of our overall approach, which does indeed show statistically identical Fe-OC loadings for the four formations in which both mineralogies overlap in time. While we would ideally further investigate the Presles Fm—and may do so in the future—this is beyond the scope of the current manuscript.

Fe-OC isotopes and reconstructed DOC $\delta^{13}\text{C}$: The Reviewer raises a good point regarding Fe-OC $\delta^{13}\text{C}$ values—one which we are in agreement. As seen in Fig. 3C, reconstructed DOC $\delta^{13}\text{C}$ for the two goethite-containing Ordovician formations is clearly not statistically identical to that for the two hematite-containing Ordovician formations. It is therefore possible that the isotope records may exhibit some amount of bias or offset between mineralogies. Our text presents this discrepancy openly and honestly, for example in the main text:

L234–236: “Like for measured Fe-OC $\delta^{13}\text{C}$, Ordovician $\delta^{13}\text{C}_{\text{DOC}}$ reconstructed using goethite vs. hematite ooids are similar but not statistically identical (two-tailed t -test $p > 0.05$).”

Nevertheless, we now give significantly less weight and attention to the $\delta^{13}\text{C}$ record in the revised manuscript, and we instead focus more attention on the novel aspect of our work—the DOC concentration record (see also our response to Reviewer 1, above). Specifically, we have completely removed the (admittedly somewhat speculative) interpretation that this changing $\delta^{13}\text{C}$ signal implies something about changing organic carbon burial through time. As the Reviewer rightly points out, such interpretation could be biased if the two mineralogies do not produce $\delta^{13}\text{C}$ reconstructions that agree within statistical uncertainty. Now, we instead simply state:

L262–267: “Additionally, observed crude oil, kerogen, and DOC $\delta^{13}\text{C}$ trends—when combined with constant $\delta^{13}\text{C}_{\text{carb}}$ (binned into 100 Ma periods to dampen transient excursions; Fig. 3C)¹⁷—imply that photosynthetic organisms exhibited a larger ε_p and/or that organic carbon reservoirs contained higher proportions of ^{13}C -depleted compound classes (e.g., lipids) or higher proportions of material derived from ^{13}C -depleted sources (e.g., methane) in the Proterozoic.”

We do not provide any additional interpretation of our $\delta^{13}\text{C}$ record beyond this statement. While certainly an interesting foundation for future follow-up work, we now agree with the Reviewer(s) that such interpretations would require additional testing that is beyond the scope of this work. Nevertheless, we choose to leave the DOC $\delta^{13}\text{C}$ record within Fig. 3 as motivation for future studies.

If we by-pass the Monte Carlo interpretation of the raw data (which was difficult to follow and buried in SI), ...

Response: We apologize that the Reviewer found our Monte Carlo model difficult to follow. The model derivation, assumptions, and validation are indeed presented in the Supplemental Discussion due to the length requirements imposed by the journal. Although extensive and thorough, this text is well-labeled and separated into several sections and sub-sections that can now also be found in a table of contents (i.e., beginning on L3204: Supplemental Discussion

Section 6, “Estimating DOC concentrations and $\delta^{13}\text{C}$ values from iron ooid Fe-OC”, and sub-sections “Mathematical derivation”, “Possible $f^i(t)$ scenarios”, “Monte Carlo solution”, and “Sensitivity tests and validity of assumptions”). Furthermore, all assumptions and model validation points are articulated within labeled, numbered lists for ease of reading. We hope the Reviewer finds the addition of this table of contents useful for following the text, and we are open to any suggestions to help improve readability.

...just from the experimental calibration curves (ED10 and ED11 seem to be the key calibrations and it would have helped if these were isolated from the large number of other sensitivity tests and promoted to the main paper), then hematite is trapping less DOC for the same DOC/Fe, but not quite enough to explain the raw Fe-OC data offset in Figure 1.

Response: We agree that these figures (now Figs. ED18-ED19 in the revised version) do represent important loading-curve responses and are critical for the full interpretation of our reconstructed results shown in Fig. 3. Unfortunately, however, due to the spatial constraints imposed by the journal we are unable to move these figures to the main text. We nevertheless reference them extensively in the main text when describing our calibration curve and modeling approach.

Could the remainder of the Figure 1 Fe-OC offset, and a loss of OC, result from recrystallization of primary goethite? I don't have an explanation for the mineralogical offset. It could be telling us something important and useful.

Response: We are unsure what the Reviewer is referring to when they say “recrystallization of primary goethite”. Does this refer to goethite-containing ooids that may have precipitated as other minerals and have since recrystallized to goethite? Or is this instead referring to hematite-containing ooids which the Reviewer is suggesting were initially precipitated as goethite and have since been recrystallized to hematite? If the former, then there is no petrographic evidence of this occurring. Rather, goethite ooids are generally accepted by the sedimentological community to be primary precipitates, as confirmed by the observation that all modern ooids are indeed goethite despite not yet being lithified (see Supplementary Discussion beginning on L2401, L2556, L2574, and Figs. ED25-ED29).

If the latter, then there is again no petrographic or geochemical evidence that (Precambrian) goethite would have been the primary precipitate and would have subsequently ripened to hematite. Nevertheless, we do discuss the possibility that some hematites represent replacement of primary chamosite ooids, which has been proposed in the literature. Specifically, this possibility is discussed for each individual formation based on petrographic results (see Supplementary Discussion Section 4 beginning on L2805 and Figs. ED94–ED123). Additionally, our Supplementary Discussion (both in the original and revised versions) contains the following text regarding hematite formation mechanisms as they relate to the underlying assumptions of our [DOC]* reconstruction approach:

L3458–3506: “We assume that all iron (hydr)oxides either precipitate directly in seawater or form as early oxidation products of iron silicates (e.g., chamosite) at the sediment-water interface. In contrast, if iron ooids represent later-stage diagenetic products, then Fe-OC signals cannot be reliably interpreted as marine DOC proxies. This assumption must be satisfied for goethite ooids, since these are observed to form directly in the modern ocean^{30,31}, but is less constrained for hematite ooids. We thus consider two possible formation pathways:

Open system formation. In this scenario, iron oxides either precipitated directly in seawater or underwent transformation from initial iron-silicates through dissolution and re-precipitation on the seafloor. Importantly, early mineralogical transitions from chamosite have been docu-

mented^{64–67} and follow the general reaction (here written for hematite, similar reactions apply for goethite formation)

If this dissolution and re-precipitation mechanism occurs shortly after sediment deposition on the seafloor, then preserved Fe-OC signals in resulting iron oxides should still reflect a primary seawater signature. Petrographic analysis suggests that some hematite-bearing ooid samples in our record likely underwent this mineralogical transformation. For example, the Devonian aged Presles Fm, Belgium contains hematite ooids embedded in a matrix of silicified dolomite with cores enriched in Al, Mg, and Si relative to their outer rims (Fig. ED89). This indicates a primary Fe(II)-bearing silicate core that was subsequently oxidized and transformed. In contrast, rims may have precipitated directly as hematite on the ocean floor, suggesting that core transformation indeed occurred early and that Fe-OC signals reflect primary marine DOC signatures. Furthermore, results argue against dissimilatory iron reduction, which would instead lead to a hematite core surrounded by Fe(II)-bearing silicate outer rims (or Al-, Mg-, and Si-enriched oxide rims if Fe(II)-bearing silicates were subsequently re-oxidized). Additionally, such reduction (with or without re-oxidation) would likely erase finely laminated petrographic structures, particularly in outer rims. However, neither phenomenon is ever observed, thus precluding dissimilatory iron reduction in any sample studied here. This scenario therefore satisfies our assumption.

Closed system transformation. In this scenario, iron silicates transform into oxides only after sediment is buried and disconnected from the open ocean. We consider this scenario unlikely as it would require either (i) Fe(III)-bearing primary iron silicates (e.g., nontronite), inconsistent with the presence of Mg in ooid cores, which instead suggests Fe(II)-bearing iron silicates (e.g., chamosite); or (ii) significant downward flux of O₂ or other oxidant into deep sediments (i.e., to satisfy Eq. 1)⁶⁸, inconsistent with known bottom- and pore-water hypoxia—and thus sub-centimeter sediment O₂ penetration depths—in the Precambrian. Nevertheless, closed system transformation would imply that our experimental hematite Fe-OC calibration curves do not apply. Rather, Fe-OC loadings and δ¹³C values would be governed by calibration curves for the primary mineral (e.g., chamosite). Organic compounds would then be retained as co-precipitated Fe-OC after closed-system transformation; preserved hematite Fe-OC signals would still be governed by primary marine signatures, but via an iron silicate intermediate. Reconstructed DOC concentrations and δ¹³C values using our hematite calibration curves could therefore be systematically biased. This scenario does not quantitatively satisfy our assumption but should still lead to qualitatively meaningful temporal Fe-OC trends.

Results imply either that the “open system formation” scenario applies to all samples in our record or that the “closed system transformation” scenario does not lead to meaningful biases (i.e., nearly identical chamosite and hematite calibration curves). This is evidenced by the fact that goethite and hematite Fe-OC signals yield statistically identical reconstructed DOC signatures—particularly concentrations—during geologic periods in which both mineralogies overlap (Fig. ED127). We therefore interpret Fe-OC in both minerals as reflecting marine signals.”

We hope this answers the Reviewer’s questions regarding goethite recrystallization.

What I am not currently inclined to accept on what I have read, is that the offset is simply a real offset in DOC.*

Response: We hope that our extensive discussion and validation articulated above has convinced the Reviewer that there is, in fact, no offset in reconstructed [DOC]* values—this is not

observed in either Fig. 2A (raw data) or Fig. 3B (reconstructed concentrations). To summarize the above discussion, the iron ooid record is inherently sparse and contains a mineralogical transition in the early Phanerozoic that is likely related to changing silica concentrations¹⁴. Despite this, we have managed to compile four overlapping formations from the Ordovician—to which contain goethite ooids and two which contain hematite ooids. For this time period, both mineralogies produce statistically identical results, both in terms of raw Fe-OC wt % values and in terms of reconstructed [DOC]* (although the hematite formations exhibit higher uncertainty). Barring the discovery of additional formations that can expand this overlapping time window, we are unaware of any further confirmation tests that can be performed. We are of course open to any suggestions that the Reviewer may have.

Unlike for [DOC]*, we agree with the Reviewer that our $\delta^{13}\text{C}$ record could exhibit some mineralogical biases, and we honestly state this in the main text. Accordingly, we devote significantly less attention to interpreting the isotope record in our revised manuscript.

Comment 5: Reviewer Comment: *Isotopes: $\delta^{13}\text{C}$ shows a large offset between goethite and hematite substrates with no value that overlaps between phases (unlike the 450 Ma point for DOC*).*

As discussed above, we agree with the Reviewer that the mineralogical $\delta^{13}\text{C}$ offset makes this record weaker than that of [DOC]*. To summarize again here, our main text states this clearly (L234–236), and we now give significantly less weight and attention to the $\delta^{13}\text{C}$ record in the revised manuscript, instead focusing more attention on the novel aspect of our work—the DOC concentration record (see also our response to Reviewer 1, above). We have completely removed the (admittedly somewhat speculative) interpretation that this changing $\delta^{13}\text{C}$ signal implies something about changing organic carbon burial through time, and we instead simply state a range of reasons why $\delta^{13}\text{C}$ may have been lower in the Precambrian, leaving the thorough interpretation for a future follow-up publication.

The 2 mineral phases exhibit polar opposite fractional behaviors which is interesting – ED Figure 10 vs. 11. ^{13}C fractionation on hematite declines with lower DOC/Fe, which is not unexpected in sign, although I am surprised by the magnitude of the fractionation as DOC/Fe tends to zero. How does this compare with organic compounds being absorbed onto, and/or incorporated into a growing lattice, for other substrates? Rather more surprising is the positive-with-declining-DOC/Fe behavior for goethite shown in EX Figure 10. Do the authors have a mechanistic explanation for this?

We agree with the Reviewer that the differences in ^{13}C fractionation behavior are indeed interesting! To clarify, the “polar opposite” behavior depends on the molecular composition of the DOC in each experiment—for C-DOC, both minerals exhibit fractionation that is the same in sign (although not magnitude), whereas the minerals display fractionation that is opposite in sign for M-DOC and FA (Figs. ED18-ED19). This adds another interesting layer of complexity. We unfortunately do not yet have a mechanistic understanding of the fractionation process—this would require extensive work that is beyond the scope of the current manuscript. That said, we are planning follow-up experiments and analyses for subsequent, more mechanistic studies. This will include solid-phase analysis of specific OC functional groups as they are coprecipitated (e.g., by XANES or NMR analysis). However, this remains an open topic for future work that will likely focus heavily on the mechanisms of iron-OC interactions and their role as a sink of OC from the environment (e.g., following Refs.^{69,70}). We do note that we attempted to dissolve iron oxides and analyze the molecular composition of the released organic carbon by high-resolution FT-ICR-MS as part of this study. However, unfortunately, we were analytically unable to completely remove the resulting dissolved iron, which led to severe ion suppression

in the MS source and thus prevented us from producing any useful mass spectra.

We are not entirely sure what the Reviewer is referring to when they say, “how does this compare with organic compounds being absorbed onto, and/or incorporated into a growing lattice, for other substrates?” While we certainly could test additional DOC substrates with more experiments, this would likely not provide significant improvements in our data interpretation for the current manuscript (i.e., since we have already attempted to capture the relevant DOC sources; see also our response to Reviewer 4, above). That said, such experiments will be included in follow-up work.

My suspicion would be that we are seeing a change in discrimination between different marine OC compounds, each with different $\delta^{13}\text{C}$ – i.e., bulk marine OC $\delta^{13}\text{C}$ actually represents the weighted mean of a variety of different compounds positively and negatively fractionated compared to bulk DOC (e.g., DOI: 10.1146/annurev-marine-121916-063634). This is expressed at low DOC/Fe with different compounds being incorporated into goethite vs. hematite, while at very high DOC/Fe, perhaps everything (all different molecules) ‘sticks’. If so, ecosystem composition, in determining the initial mix of compounds, could drive changes in the recorded $\delta^{13}\text{C}$, even for the same DOC/Fe and same bulk OC $\delta^{13}\text{C}$. In this interpretation – that $\delta^{13}\text{C}$ in Figure 2D for hematite deviates so far from kerogen, simply reflects that hematite is sampling a range of more depleted compounds compared to bulk organic matter.

Response: We agree with the Reviewer that bulk DOC represents a mixture of different compound classes, each with varying $\delta^{13}\text{C}$ values—this is a well-established result in the field of organic geochemistry (e.g., Ref.⁷¹; we note that our main text, in both the original and resubmitted versions, explicitly stated this, e.g.,: L264–265: “...carbon reservoirs contained higher proportions of ^{13}C -depleted compound classes (e.g., lipids)...). That said, the mechanism the Reviewer proposes here—i.e., that more ^{13}C -depleted compounds are selectively preserved at low DOC/Fe(III) ratios but this selective preservation disappears at high DOC/Fe(III)—**cannot be the only explanation**. This results from the observation that our fulvic acid (FA) experiments exhibit similar behavior despite the fact that FA is a single molecular compound (see, e.g., Figs. ED18F and ED19F). This is precisely the reason we included such experiments here—to test the isotopic fractionation for a single molecule. This is clearly discussed in our Supplementary Discussion (both in the original and revised versions):

L2348–2361: “Carbon-isotope fractionation exhibits more complex behavior (Figs. ED18–ED19). For goethite, using either M-DOC or FA as the DOC source yields positive $\Delta^{13}\text{C}$ values (i.e., Fe-OC is more depleted in ^{13}C than substrate DOC) as high as 12‰. In contrast, C-DOC exhibits slightly negative but largely invariant $\Delta^{13}\text{C}$ values. For hematite, all three DOC sources yield negative $\Delta^{13}\text{C}$ values (i.e., Fe-OC is more enriched in ^{13}C than substrate DOC) as low as –20‰. This diverging fractionation behavior does not clearly depend on any DOC composition metric measured here. **Additionally, it is unlikely that fractionation results from preferential co-precipitation of a compositionally and thus isotopically unique subset of the overall DOC reservoir, as evidenced by the observaton that FA—which is comprised of a single molecular structure—exhibits a different sign and magnitude of fractionation when co-precipitated with goethite and hematite.** Rather, we hypothesize that observed fractionation results from a kinetic isotope effect whose magnitude depends on DOC/Fe(III) ratio. Further work is clearly warranted to investigate the exact mechanism(s) governing fractionation, particularly the difference in sign and magnitude observed for different minerals and DOC sources.”

Still, we agree with the Reviewer that characterizing the mechanism of isotope fractionation would make for an exciting follow-up study—one which we are planning to do!

We additionally agree with the Reviewer that the fact that different molecules exhibit unique $\delta^{13}\text{C}$ values could at least partially explain our isotope fractionation results. It is precisely for this reason that we thoroughly assessed the potential impact of this possibility in the Supplementary Discussion (both in the original and resubmitted versions). Specifically, we performed a sensitivity test to quantify how large of a bias could be induced by our simplification of constant $\delta^{13}\text{C}$ values across compound classes. In our sensitivity test, this is treated as an isotopic difference between M-DOC and C-DOC, but this would equally apply for a difference in, say, lipids and carbohydrates. We write:

L3562–3600: “This assumption is required to translate end members weighted by their relative abundance in marine DOC to end members weighted by their relative abundance in Fe-OC; these weightings need not be identical due to the differential Fe-OC loadings for different end members at a given DOC/Fe(III) ratio (Figs. ED18-ED19). This assumption is relaxed only in the special case of equal Fe-OC loadings for all end members. That is, if $f^i(t) = w^{i,m}(t)/W^m(t)$ for all i , then Eq. S14 reduces to

$$\Delta^m(t) = \sum_i f^i(t) \Delta^{i,m}(t), \quad (2)$$

which is simply the average of end-member fractionation factors weighted by their relative abundance in marine DOC. In this case, each end member need not be described by the same $\delta^{13}\text{C}$ value, since the weightings in Fe-OC are identical to those in marine DOC.

To assess the magnitude of potential biases due to this assumption, we calculated “true” vs. “reconstructed” marine DOC $\delta^{13}\text{C}$ values as a function of the $\delta^{13}\text{C}$ offset between M-DOC and C-DOC for two scenarios: (i) $[\text{DOC}]^*(t) = 0.1$ as reconstructed by hematite, representing Proterozoic conditions; and (ii) $[\text{DOC}]^*(t) = 1$ as reconstructed by goethite, representing Phanerozoic conditions (Fig. ED130). In the Proterozoic case, an unrealistically large end-member offset in marine DOC of $\delta^{13}\text{C}_{\text{M-DOC}} - \delta^{13}\text{C}_{\text{C-DOC}} = 20\text{‰}$ would translate to a true-minus-predicted bias of $\approx 6\text{‰}$ at $f^{\text{M-DOC}}(t) \approx 0.1$ (Fig. ED130A). This bias approaches zero as $f^{\text{M-DOC}} \rightarrow 0.65$, the point at which the special case of Eq. 2 is met, and as $f^{\text{M-DOC}} \rightarrow 0$ or $f^{\text{M-DOC}} \rightarrow 1$, i.e., when all DOC is derived from a single end member. In the Phanerozoic case, the same unrealistically large end-member offset in marine DOC of $\delta^{13}\text{C}_{\text{M-DOC}} - \delta^{13}\text{C}_{\text{C-DOC}} = 20\text{‰}$ would translate a maximum true-minus-predicted bias of $\approx 4\text{‰}$ at $f^{\text{M-DOC}}(t) \approx 0.35$ (Fig. ED130B). This bias again approaches zero in the single end-member limits of $f^{\text{M-DOC}} \rightarrow 0$ and $f^{\text{M-DOC}} \rightarrow 1$.

However, it is unlikely that different quantitatively important end members exhibited $\delta^{13}\text{C}$ offsets as large as 20‰ at any point in our iron ooid record. This is because oxygenic photosynthesis via the Calvin-Benson-Bassham (CBB) cycle has likely been a dominant metabolism at least since the Mesoproterozoic, before the beginning of our record⁷². All organisms that fix carbon via the CBB cycle utilize the enzyme ribulose 1,5-bisphosphate carboxylase oxygenase (RuBisCO)⁷³, which exhibits an *in vitro* ^{13}C fractionation factor between CO_2 and organic matter that differs by a maximum of $\approx 10\text{‰}$ across all marine oxygenic photosynthetic organisms studied to date (i.e., RuBisCO types IA, IB, ID, and II)⁷⁴. In contrast, chemolithoautotrophic metabolisms—which can express drastically different fractionation factors—were likely quantitatively unimportant in terms of DOC generation throughout our record. Differences in expressed fractionation between oxygenic photosynthetic organisms are further dampened when analyzed *in vivo*, likely due to the importance of carbon-concentrating mechanisms (CCMs) in modern photosynthetic organisms⁷⁴. However, CCMs may not have yet evolved at the start of our iron ooid record⁵³. Nevertheless, more realistic end-member $\delta^{13}\text{C}$ offsets of ≈ 5 (after the

advent of CCMs) to 10‰ (before the advent of CCMs) would lead to smaller biases in reconstructed average DOC $\delta^{13}\text{C}$ values that never exceed 2 to 3‰. This bias is within the typical model uncertainty for any given geologic period.”

In summary, a difference in $\delta^{13}\text{C}$ of 20‰ between DOC sources or compound classes could lead to a maximum isotopic bias in our reconstructed DOC of up to 10‰. This uncertainty is also clearly and honestly stated in the main text (both the original and revised versions); specifically, we write:

L225–227: “We tested each assumption for each $f^i(t)$ scenario; resulting uncertainty could lead to a $\sim 2\times$ bias in $[\text{DOC}]^*$ and $\sim 10\%$ bias in $\delta^{13}\text{C}_{\text{DOC}}$ (Supplementary Discussion; Figs. ED127-ED130).”

So my concern is what the Fe-OC is in terms of compounds, because I have no reason to think it reflects bulk DOC, or even bulk SLDOC.

Response: As discussed above, we strongly disagree with the Reviewer that our Fe-OC proxy is only capturing semi-labile DOC or even a subset of the semi-labile DOC pool—**this would be an over-interpretation of our modern ooid ^{14}C data and a mis-interpretation of the uncertainty inherent to geologic proxy archives.** Because there is no valid evidence to the contrary (despite our multi-faceted approach to test this), we interpret our ooids as capturing the range of DOC pools that are present in continental shelf waters (i.e., from labile to recalcitrant). Thus, it is the *total* DOC concentration in continental shelf waters—not just that of the semi-labile pool—that matters for the Fe-OC proxy. That said, we reiterate that we agree with the Reviewer this is greater than—and not equal to—the DOC concentration of the deep open-ocean, as is clearly stated in the main text (L206–208).

We hope that our extensive, detailed response to the Reviewer’s above comments have sufficiently articulated these points.

What is needed is for the Fe-OC to be characterized in terms of compounds (ultimately, with compound specific ^{13}C measurements). For the modern, actively-forming ooids, if this was paired with measurements of seawater SLDOC (+SRDOC), or if this was just done in the experiments, I think this proxy would be very significantly advanced.

Response: We agree with the Reviewer that molecular characterization of the co-precipitated OC would be ideal. It is for this reason that we attempted to dissolve iron oxides and analyze the molecular composition of the released organic carbon by high-resolution FT-ICR-MS as part of this study. However, we were analytically unable to completely remove the resulting dissolved iron, which led to severe ion suppression in the MS source and thus prevented us from producing any useful mass spectra. Therefore, we were unfortunately unable to molecularly characterize this material. Solid-phase molecular characterization (e.g., by XANES and/or NMR) is a primary target for ongoing follow-up work that focuses on the role of iron co-precipitation as a sink of organic carbon from the ocean. However, such detailed molecular characterization is outside the scope of the current manuscript, which instead focuses on applying this tool as a proxy for deep-time DOC concentrations.

Regarding the Reviewer’s suggestion that we build a compound-specific $\delta^{13}\text{C}$ record: this would of course also be ideal but is not feasible at this time for at least three reasons:

(i) **Identification of suitable compounds:** Given that no deep-time DOC reconstructions exist to date, it is unclear where one would begin in terms of selecting an appropriate compound (or compound class) to generate such a record. Such compounds would need to meet the basic criteria required to be considered a reliable biomarker (i.e., source specificity, preservation potential, and ability to be analyzed with existing methods)⁷⁵. This is of course the foundation of

much of the field of molecular organic geochemistry that focuses on hydrophobic compounds (i.e., those that primarily associate with particulate organic carbon and are thus preserved in, e.g., shales; see Refs.^{34,36}). However, such compounds are almost exclusively lipids, which degrade to hydrocarbons and are unlikely to remain in the dissolved phase or to represent DOC. It is therefore unclear which hydrophilic compounds (i.e., that remain in the dissolved phase) would meet the basic criteria to be considered a reliable biomarker, in particular the preservation potential (i.e., since the most well-preserved compounds are typically lipids which degrade to hydrophobic hydrocarbons). This is of course a surmountable and exciting research question and could form the foundation of future proposals and research targets, but would require a career's worth of work that is beyond the scope of the current manuscript. In fact, we hope our manuscript will prompt the scientific community to pursue exactly this path.

(ii) **Sample size:** All iron ooids included in this study contain between 0.3 and ~ 1 wt % Fe-OC. Although it is difficult to estimate *a priori* what would be the relative abundance of any single biomarker in this Fe-OC (i.e., in $\mu\text{g compound}(\text{g OC})^{-1}$, or ppm), a reasonable assumption based on Precambrian hopane biomarker work in shales (e.g., Refs.^{34,36}) would suggest something on the order of 0.1 to 1 ppm. This is likely a maximum estimate given the high preservation potential of hydrocarbon lipids. Thus, to obtain a reasonable mass of biomarker needed for compound-specific $\delta^{13}\text{C}$ analysis ($\sim 1 \mu\text{g}$), one would need to extract something like 500 g to 1 kg of iron ooids per sample. Furthermore, all ooid samples contained in this study were hand picked under a microscope (i.e., to remove any matrix cement that may contaminate signals). Thus, even if we possessed large enough hand samples to perform these extractions, hand picking 500 g to 1 kg of ooids is effectively an impossible task.

(iii) **Molecular alteration:** Assuming one did identify a suitable biomarker molecule (or group of molecules) and was able to hand-pick an appropriate amount of iron ooids, it is not clear that subsequent chemical treatment would not lead to chemical alteration of the organic matter. Specifically, because Fe-OC is co-precipitated within the crystal lattice, ooids would need to be dissolved using 12 N HCl held at 80°C (a protocol which we applied during our failed attempt at high-resolution FT-ICR-MS characterization as part of this study). It is highly likely that such a strongly acidic extraction solution would lead to significant alteration or destruction of individual compounds, particularly those that contain partially oxidized functional groups. It may still be possible to gain meaningful information from hydrocarbons that would remain after this acid treatment, yet testing this would again require a career's worth of work that is beyond the scope of the current manuscript.

Furthermore, we note that a compound-specific record would only inform how changes in the carbon isotope composition **of that particular compound (class)** has changed through time, and not the total DOC pool. For example, consider the heuristic case where one determines that some lipid compound is an ideal target for an iron ooid-derived compound-specific $\delta^{13}\text{C}$ record (and has overcome the three limitations listed above). Then, assume one is able to generate such a record and observes, say, an increase in $\delta^{13}\text{C}$ of 10‰ of this compound through time. While certainly informative, it would nevertheless remain ambiguous if the entire DOC reservoir similarly increased in $\delta^{13}\text{C}$ due to, say, a change in ε_p of primary producers over this time, or if this particular compound represents only a small subset of DOC that is subject to, say, changes in methanotrophy.

Finally, we emphasize that such a record would **not** yield any useful information on changes in **DOC concentration** through time. This is because any given molecule only represents a small fraction of the total organic carbon pool. Rather, for reconstructing DOC concentration—which is the primary focus of our manuscript—only bulk measurements are

appropriate. Thus, while certainly informative and potentially worth pursuing in the future, many scientific, logistical, and analytical challenges remain that preclude the development of a compound-specific DOC $\delta^{13}\text{C}$ record at this time.

We hope that this explanation has sufficiently addressed the reviewer's request at constructing a compound-specific record of DOC through time.

As a note – taking the corrected $\delta^{13}\text{C}$ (Figure 2C) at face value, I wonder whether the authors had considered a stronger methane cycle in the early Paleozoic and Precambrian, creating DOC compounds with a much lighter signal?

Response: We thank the Reviewer for this interesting suggestion, which we now propose as a possible explanation in the main text:

L263–265: “[our $\delta^{13}\text{C}$ results] either imply that photosynthetic organisms exhibited a larger ε_p and/or that organic carbon reservoirs contained higher proportions of ^{13}C -depleted compound classes (e.g., lipids) or *higher proportions of material derived from ^{13}C -depleted sources (e.g., methane) in the Proterozoic.*”

Still, we again emphasize that our revised manuscript has significantly scaled-down the interpretation of our $\delta^{13}\text{C}$ record given the caveats and limitations discussed above (and in response to Reviewer 1). We therefore do not pursue this line of reasoning any further but leave it as a speculative, potential explanation.

Comment 6: **Reviewer Comment:** *Other: I cannot make out the changes associated with the PETM in the figures (and no data values are provided). This sounds really interesting ... a paper in itself almost ... In the main text, SI discussion was promised, but I could find hardly anything.*

Response: We again apologize that the Reviewer could not find the relevant data, but we assure them that all raw data and all code to perform all calculations (i.e., Monte Carlo model) are indeed included in the Supplementary Data 1–6 and Supplementary Code. We agree that the PETM result is interesting and could form the basis for a follow-up paper, and we invite the Reviewer to pursue such a story if they so choose. Regarding the PETM Supplementary Discussion that is cited in the main text: this is specifically referring to the petrographic description of the Bakchar Horizon, Lyolinvor Fm (i.e., Figs ED33-ED34; Supplementary Discussion Section 4.4 beginning on L2609).

Comment 7: **Reviewer Comment:** *Other: The assumption is made that the Fe^{II} flux to the site of ooid formation is constant through time, and hence the Fe value in the DOC/Fe-based calibration is constant. It seems unlikely that every iron ooid location had a very similar hydrothermal environment to presumably much less than a factor of 2 in the Fe term. 2 modern sites are discussed—in the Italy Aeolian archipelago and near Mahengentang Island in Indonesia. What is known about Fe fluxes here? If nothing, what about hydrothermal heat fluxes or some proxy for activity that we might relate to Fe^{II} flux? How similar or different are these formation environments?*

Response: We thank the Reviewer for bringing up this important point regarding the assumption of a constant Fe(II) flux at the sites of iron ooid formation through time. In the original manuscript we acknowledge that variations in Fe(II) flux could potentially influence local Fe(III) concentrations, which in turn could impact our absolute DOC concentration estimates derived from Fe-OC loadings (under “Sensitivity tests and validity of assumptions; Assumption (iv). No change in Fe(II) flux at the site of ooid formation through time”).

Our assumption of a relatively constant Fe(II) flux is based on several lines of evidence suggesting that hydrothermal Fe(II) fluxes have remained fairly stable since approximately

2,000 million years ago (Ma), despite significant changes in open-ocean dissolved iron concentrations over this period.

First, global models of lithospheric heat loss and mantle convection predict minimal changes in surface heat flow, which largely controls hydrothermal activity and elemental fluxes, including Fe(II). For example, Ref.⁷⁶ estimate only about a 40 % decrease in global hydrothermal heat flux since 2000 Ma. Similarly, Refs.^{77,78} suggest little-to-no change in mantle heat flow over this time frame. These findings imply that the overall hydrothermal Fe(II) flux to the oceans has not varied dramatically over the time periods relevant to our study.

Second, our data show a lack of spatial heterogeneity in Fe-OC signals from similarly aged formations across different geographic locations (Fig. ED125). This suggests that local variations in Fe(II) flux at ooid formation sites are minimal, supporting the assumption that Fe(II) fluxes are relatively constant both temporally and spatially in the contexts relevant to our study.

Third, we considered a hypothetical scenario where the observed variations in Fe-OC loadings are solely due to changes in Fe(II) flux, rather than changes in DOC concentrations. If we assume that DOC concentrations remained constant through time²⁷, the Fe-OC loading record would require a ~100-fold decrease in Fe(II) flux across the Proterozoic, followed by a similarly large increase at the Precambrian-Cambrian boundary (Main text; Fig. 3). This pattern is opposite to what is predicted by models of upwelling of ferruginous deep seawater⁷⁹. Moreover, under the “large Proterozoic DOC reservoir” model proposed by Ref.²¹, explaining our observations would require an even more dramatic ~100,000-fold decrease in Fe(II) flux through the Proterozoic. Such large variations in Fe(II) flux are not supported by geological evidence or existing models of Earth’s geochemical cycles.

Regarding the modern sites at Panarea Island (Italy) and Mahengetang Island (Indonesia), unfortunately, specific measurements of Fe(II) fluxes at these exact locations do not exist. However, the formation of iron ooids is not restricted to areas with significant hydrothermal Fe(II) inputs; they can also form in environments where Fe(II) is supplied through other mechanisms, such as diagenetic remobilization of iron in sediments or continental weathering inputs.

Importantly, while variations in Fe(II) flux could impact the absolute Fe(III) concentrations at ooid formation sites, our proxy relies on the ratio of DOC to Fe(III) (i.e., DOC/Fe(III)), which is more robust to changes in absolute Fe(II) fluxes. This is because the Fe-OC loadings are primarily controlled by the DOC/Fe(III) ratio, rather than the absolute concentrations of Fe(III) or DOC alone. Therefore, even if Fe(II) fluxes varied to some extent, as long as the DOC/Fe(III) ratio remains representative of the environmental conditions, our reconstructed DOC concentrations and $\delta^{13}\text{C}$ trends remain valid.

In summary, while we cannot entirely rule out modest variations in Fe(II) flux through time and across different ooid formation sites, the evidence suggests that such variations are unlikely to be large enough to account for the orders-of-magnitude changes in Fe-OC loadings observed in our record. We provide a detailed discussion of this assumption and its implications to the Supplementary Discussion (Section 5.4), where we also explore the potential effects of Fe(II) flux variability on our proxy results.

We appreciate the Reviewer bringing this to our attention and hope that this response addresses their concerns.

Comment 8: **Reviewer Comment:** *Other: (lines 129-131 plus ED121) To check for latitudinal bias in this way with a regression, you need to plot the variable (y-axis) vs. some measure of distance*

from the equator, regardless of whether you are going North or South. What you cannot do is put a straight line through the absolute values from -90 to 90 and conclude anything.

We appreciate the Reviewer's suggestion regarding the potential latitudinal bias in our analysis. In response, we have replotted the data against the absolute value of latitude and paleolatitude, considering the distance from the equator rather than the direction. As shown in the updated figure below, this change did not significantly alter the observed correlations. This suggests that there is no evident latitudinal bias affecting the Fe-OC loadings or $\delta^{13}\text{C}$ values.

18

19 **Figure 4: Fe-OC signal correlation with absolute (paleo)latitude.** Fe-OC loadings and $\delta^{13}\text{C}$
 20 values are plotted against absolute modern latitude and paleolatitude. The orange lines represent
 21 best-fit OLS regressions, with r^2 values indicating no statistically significant correlation in any
 22 panel.

Comment 9: **Reviewer Comment:** *Recommendations: I don't think this is a proxy for past bulk ocean DOC and global reduced carbon reservoir, and I don't think as a new and emerging proxy, that it is quite ready for the big time. However, in sampling some set of organic molecules and preserving them unaltered, the prospects seem good to say something about evolution in the marine environment. I would be really excited to see what can be done with this proxy.*

Response: We thank the Reviewer again for their critical assessment of our manuscript and of the applicability of our Fe-OC DOC proxy. We hope we have addressed above all of the Reviewer's concerns which led them to conclude that our proxy is not yet "ready for the big time." To briefly summarize these, as far as we can tell, the Reviewer was primarily concerned that:

- (i) *All available data was not provided.* As stated above, all raw data and all code required to perform all calculations are indeed included in the Supplementary Data 1-6 and Supplementary Code. Still, we now include a Table of Contents for the Supplementary Discussion to assist the Reviewer in finding this information.
- (ii) *Radiocarbon (^{14}C) data of modern ooids implies that their Fe-OC captures a subset (of a subset) of total DOC.* As discussed at length above, this is an over-interpretation of the modern ooid ^{14}C data, which should be interpreted with caution given the large uncertainties associated with the timescale and continuity of ooid formation. This section instead represents a back-of-the-envelope calculation to preclude the possibility that all Fe-OC is derived from (^{14}C -free) hydrothermal sources. We have nevertheless updated this Supplementary Discussion section to better articulate the limitations when interpreting these data.
- (iii) *Relative variance in the proxy record that is greater than that of DOC concentration variability in modern continental shelf waters again implies that Fe-OC captures a subset (of a subset) of total DOC.* As discussed at length above (including with other, well-established examples from the paleoenvironmental proxy literature), this is a mis- (or over-)interpretation of variance in our proxy record. Several factors—including analytical error and uncertainty—necessarily lead to a proxy record with relative variance that is larger than the input environmental signal which it is tracking.
- (iv) *DOC reconstructions between hematite and goethite mineralogies do not agree.* As discussed above, we show quantitatively that these two mineralogies do, in fact, produce the same DOC concentration reconstructions (within error) for four near-time-equivalent Ordovician formations. While we would ideally have a longer time period of mineralogical overlap—or, even better, a record based on a single mineralogy—this is unfortunately not possible given existing samples and known ooid formations. Still, we agree with the Reviewer that the $\delta^{13}\text{C}$ records between mineralogies exhibit less agreement, and we have significantly reduced our interpretation of the $\delta^{13}\text{C}$ record accordingly in the revised version.
- (v) *A compound-specific approach would provide better constraints.* As described above, there are several methodological and scientific limitations that currently prevent the generation of a compound-specific $\delta^{13}\text{C}$ record from iron ooids. Nevertheless, this will make for an exciting avenue for future research over the next 5-10 years, and we hope that our current manuscript motivates the scientific community to pursue this path.

We again thank the Reviewer for allowing us the opportunity to explain and clarify these points that greatly strengthened our manuscript.

In summary, we hope that the revised manuscript addresses all the Reviewers' comments satisfactorily. Thank you for considering our revisions.

Sincerely,
Nir Galili
(on behalf of all co-authors)

References

- [1] Henry, D. G., Jarvis, I., Gillmore, G. & Stephenson, M. Raman spectroscopy as a tool to determine the thermal maturity of organic matter: Application to sedimentary, metamorphic and structural geology. *Earth-Science Reviews* **198**, 102936 (2019).
- [2] Sturesson, U., Heikoop, J. M. & Risk, M. Modern and palaeozoic iron ooids—a similar volcanic origin. *Sedimentary Geology* **136**, 137–146 (2000).
- [3] Matheson, E. J. & Pufahl, P. K. Clinton ironstone revisited and implications for Silurian Earth system evolution. *Earth-Science Reviews* **215**, 103527 (2021).
- [4] Schunck, S., Rickli, J., Wohlwend, S., Weissert, H. & Vance, D. Continental weathering as the source of iron in Jurassic iron oolites from Switzerland. *Swiss Journal of Geosciences* **116**, 4 (2023).
- [5] Jenkyns, H. C. Submarine volcanism and the toarcian iron pisolites of western sicily. *Eclogae Geologicae Helvetiae* **63**, 741–774 (1970).
- [6] Luan, X., Sproat, C. D., Jin, J. & Zhan, R. Depositional environments, hematite–chamosite differentiation and origins of middle ordovician iron ooids in the upper yangtze region, south china. *Sedimentology* (2024).
- [7] Baele, J.-M. et al. Iron microbial mats in modern and phanerozoic environments. In *Instruments, Methods, and Missions for Astrobiology XI*, vol. 7097, 171–182 (SPIE, 2008).
- [8] Vodrážková, S. et al. Ferruginous coated grains of microbial origin from the lower devonian (pragian) of the prague basin (czech republic)—petrological and geochemical perspective. *Sedimentary Geology* **438**, 106194 (2022).
- [9] Salama, W., El Aref, M. & Gaupp, R. Mineral evolution and processes of ferruginous microbialite accretion—an example from the middle eocene stromatolitic and ooidal ironstones of the bahariya depression, western desert, egypt. *Geobiology* **11**, 15–28 (2013).
- [10] Salama, W., El Aref, M. & Gaupp, R. Mineralogical and geochemical investigations of the middle eocene ironstones, el bahariya depression, western desert, egypt. *Gondwana Research* **22**, 717–736 (2012).
- [11] Ciobotă, V. et al. Identification of minerals and organic materials in middle eocene ironstones from the bahariya depression in the western desert of egypt by means of micro-raman spectroscopy. *Journal of Raman Spectroscopy* **43**, 405–410 (2012).
- [12] Trower, E. J., Bridgers, S. L., Lamb, M. P. & Fischer, W. W. Ooid cortical stratigraphy reveals common histories of individual co-occurring sedimentary grains. *Journal of Geophysical Research: Earth Surface* **125**, e2019JF005452 (2020).
- [13] Schwertmann, U., Friedl, J., Stanjek, H. & Schulze, D. G. The effect of Al on Fe oxides. XIX. formation of Al-substituted hematite from ferrihydrite at 25 °C and pH 4 to 7. *Clays and Clay Minerals* **48**, 159–172 (2000).
- [14] Galili, N. et al. The geologic history of seawater oxygen isotopes from marine iron oxides. *Science* **365**, 469–473 (2019).

- [15] Racki, G. & Cordey, F. Radiolarian palaeoecology and radiolarites: is the present the key to the past? *Earth-Science Reviews* **52**, 83–120 (2000).
- [16] Siever, R. The silica cycle in the precambrian. *Geochimica et Cosmochimica Acta* **56**, 3265–3272 (1992).
- [17] Krissansen-Totton, J., Buick, R. & Catling, D. C. A statistical analysis of the carbon isotope record from the Archean to Phanerozoic and implications for the rise of oxygen. *American Journal of Science* **315**, 275–316 (2015).
- [18] Herndl, G. J. & Reinthaler, T. Microbial control of the dark end of the biological pump. *Nature Geoscience* **6**, 718–724 (2013).
- [19] Ridgwell, A. & Arndt, S. Why dissolved organics matter: DOC in ancient oceans and past climate change. In *Biogeochemistry of Marine Dissolved Organic Matter*, 1–20 (Elsevier Academic Press, 2015).
- [20] Ahm, A.-S. C. et al. The Ca and Mg isotope record of the Cryogenian trezona carbon isotope excursion. *Earth and Planetary Science Letters* **568**, 117002 (2021).
- [21] Rothman, D. H., Hayes, J. M. & Summons, R. E. Dynamics of the Neoproterozoic carbon cycle. *Proceedings of the National Academy of Sciences* **100**, 8124–8129 (2003).
- [22] Bristow, T. F. & Kennedy, M. J. Carbon isotope excursions and the oxidant budget of the Ediacaran atmosphere and ocean. *Geology* **36**, 863–866 (2008).
- [23] Johnston, D. T., Macdonald, F. A., Gill, B. C., Hoffman, P. F. & Schrag, D. P. Uncovering the Neoproterozoic carbon cycle. *Nature* **483**, 320–323 (2012).
- [24] Derry, L. A. On the significance of $\delta^{13}\text{C}$ correlations in ancient sediments. *Earth and Planetary Science Letters* **296**, 497–501 (2010).
- [25] Schrag, D. P., Higgins, J. A., Macdonald, F. A. & Johnston, D. T. Authigenic carbonate and the history of the global carbon cycle. *Science* **339**, 540–543 (2013).
- [26] Busch, J. F. et al. Global and local drivers of the Ediacaran Shuram carbon isotope excursion. *Earth and Planetary Science Letters* **579**, 117368 (2022).
- [27] Fakhraee, M., Tarhan, L. G., Planavsky, N. J. & Reinhard, C. T. A largely invariant marine dissolved organic carbon reservoir across Earth's history. *Proceedings of the National Academy of Sciences* **118**, e2103511118 (2021).
- [28] Fike, D., Grotzinger, J., Pratt, L. & Summons, R. Oxidation of the Ediacaran ocean. *Nature* **444**, 744–747 (2006).
- [29] Swanson-Hysell, N. L. et al. Cryogenian glaciation and the onset of carbon-isotope decoupling. *Science* **328**, 608–611 (2010).
- [30] Di Bella, M. et al. Modern iron ooids of hydrothermal origin as a proxy for ancient deposits. *Scientific Reports* **9**, 7107 (2019).
- [31] Heikoop, J. M., Tsujita, C. J., Risk, M. J., Tomascik, T. & Mah, A. J. Modern iron ooids from a shallow-marine volcanic setting: Mahengetang, Indonesia. *Geology* **24**, 759–762 (1996).

- [32] Rudmin, M. *et al.* Origin of oligocene channel ironstones of Lisakovsk deposit (Turgay depression, northern Kazakhstan). *Ore Geology Reviews* **138**, 104391 (2021).
- [33] Smith, F. A. *et al.* Body size evolution across the Geozoic. *Annual Review of Earth and Planetary Sciences* **44**, 523–553 (2016).
- [34] Brocks, J. J. *et al.* The rise of algae in Cryogenian oceans and the emergence of animals. *Nature* **548**, 578–581 (2017).
- [35] Eckford-Soper, L. K., Andersen, K. H., Hansen, T. F. & Canfield, D. E. A case for an active eukaryotic marine biosphere during the Proterozoic era. *Proceedings of the National Academy of Sciences* **119**, e2122042119 (2022).
- [36] Brocks, J. J. *et al.* Lost world of complex life and the late rise of the Eukaryotic crown. *Nature* **618**, 767–773 (2023).
- [37] Knoll, A. H. Paleobiological perspectives on early eukaryotic evolution. *Cold Spring Harbor Perspectives in Biology* **6**, a016121 (2014).
- [38] Anderson, P. R. & Benjamin, M. M. Effect of silicon on the crystallization and adsorption properties of ferric oxides. *Environmental science & technology* **19**, 1048–1053 (1985).
- [39] Stumm, W., Kummert, R. & Sigg, L. A ligand exchange model for the adsorption of inorganic and organic ligands at hydrous oxide interfaces. *Croatica chemica acta* **53**, 291–312 (1980).
- [40] Beckwith, R. & Reeve, R. Studies on soluble silica in soils. i. the sorption of silicic acid by soils and minerals. *Soil Research* **1**, 157–168 (1963).
- [41] Geelhoed, J. S., Hiemstra, T. & Van Riemsdijk, W. H. Competitive interaction between phosphate and citrate on goethite. *Environmental Science & Technology* **32**, 2119–2123 (1998).
- [42] Filius, J. D., Hiemstra, T. & Van Riemsdijk, W. H. Adsorption of small weak organic acids on goethite: Modeling of mechanisms. *Journal of colloid and interface science* **195**, 368–380 (1997).
- [43] Cornell, R., Giovanoli, R. & Schindler, P. Effect of silicate species on the transformation of ferrihydrite into goethite and hematite in alkaline media. *Clays and Clay Minerals* **35**, 21–28 (1987).
- [44] Schwertmann, U. Goethite and hematite formation in the presence of clay minerals and gibbsite at 25 c. *Soil Science Society of America Journal* **52**, 288–291 (1988).
- [45] Hertkorn, N. *et al.* Characterization of a major refractory component of marine dissolved organic matter. *Geochimica et Cosmochimica Acta* **70**, 2990–3010 (2006).
- [46] Helms, J. R. *et al.* Absorption spectral slopes and slope ratios as indicators of molecular weight, source, and photobleaching of chromophoric dissolved organic matter. *Limnology and Oceanography* **53**, 955–969 (2008).
- [47] Ward, C. P. & Cory, R. M. Assessing the prevalence, products, and pathways of dissolved organic matter partial photo-oxidation in arctic surface waters. *Environmental Science: Processes & Impacts* **22**, 1214–1223 (2020).

- [48] Hansell, D. A. Recalcitrant dissolved organic carbon fractions. *Annual Reviews of Marine Science* **5**, 421–445 (2013).
- [49] Follett, C. L., Repeta, D. J., Rothman, D. H., Xu, L. & Santinelli, C. Hidden cycle of dissolved organic carbon in the deep ocean. *Proceedings of the National Academy of Sciences* **111**, 16706–16711 (2014).
- [50] Gardner, W., Mishonov, A. & Richardson, M. Global poc concentrations from in-situ and satellite data. *Deep Sea Research Part II: Topical Studies in Oceanography* **53**, 718–740 (2006).
- [51] Hansell, D. A., Carlson, C. A. & Schlitzer, R. Net removal of major marine dissolved organic carbon fractions in the subsurface ocean. *Global Biogeochemical Cycles* **26**, GB1016 (2012).
- [52] Carlson, C. A. & Hansell, D. A. DOM sources, sinks, reactivity, and budgets. In *Biogeochemistry of Marine Dissolved Organic Matter*, 65–126 (Elsevier Academic Press, 2015).
- [53] Wang, R. Z. *et al.* Carbon isotope fractionation by an ancestral rubisco suggests that biological proxies for CO₂ through geologic time should be reevaluated. *Proceedings of the National Academy of Sciences* **120**, e2300466120 (2023).
- [54] Arrieta, J. M. *et al.* Dilution limits dissolved organic carbon utilization in the deep ocean. *Science* **348**, 331–333 (2015).
- [55] Logan, G. A., Hayes, J., Hieshima, G. B. & Summons, R. E. Terminal Proterozoic reorganization of biogeochemical cycles. *Nature* **376**, 53–56 (1995).
- [56] Sperling, E. A. *et al.* Statistical analysis of iron geochemical data suggests limited late Proterozoic oxygenation. *Nature* **523**, 451–454 (2015).
- [57] Stockey, R. *et al.* Sustained increases in atmospheric oxygen and marine productivity in the Neoproterozoic and Palaeozoic eras. *Nature Geoscience* **17**, 667–674 (2024).
- [58] Druffel, E. R., Williams, P. M., Bauer, J. E. & Ertel, J. R. Cycling of dissolved and particulate organic matter in the open ocean. *Journal of Geophysical Research: Oceans* **97**, 15639–15659 (1992).
- [59] Beaupré, S. R. The carbon isotopic composition of marine DOC. In *Biogeochemistry of Marine Dissolved Organic Matter*, 335–368 (Elsevier Academic Press, 2015).
- [60] Barrón, C. & Duarte, C. M. Dissolved organic carbon pools and export from the coastal ocean. *Global Biogeochemical Cycles* **29**, 1725–1738 (2015).
- [61] Young, T. P. Phanerozoic ironstones: an introduction and review. *Geological Society, London, Special Publications* **46**, 9–25 (1989).
- [62] Van Houten, F. & Purucker, M. Glauconitic peloids and chamositic ooids-favorable factors, constraints, and problems. *Earth-Science Reviews* **20**, 211–243 (1984).
- [63] Present, T. M., Adkins, J. F. & Fischer, W. W. Variability in sulfur isotope records of Phanerozoic seawater sulfate. *Geophysical Research Letters* **47**, e2020GL088766 (2020).

- [64] Bhattacharyya, D. P. & Kakimoto, P. K. Origin of ferriferous ooids; an SEM study of ironstone ooids and bauxite pisoids. *Journal of Sedimentary Research* **52**, 849–857 (1982).
- [65] Boso, M. A. & Monaldi, C. R. Oolitic stratabound iron ores in the Silurian of Argentina and Bolivia. In *Stratabound Ore Deposits in the Andes*, 175–186 (Springer, 1990).
- [66] Oyarzún, J. M. The metalliferous ore deposits of Chile and Argentina, and their geologic framework. In *Stratabound Ore Deposits in the Andes*, 61–78 (Springer, 1990).
- [67] Cotter, E. Diagenetic alteration of chamositic clay minerals to ferric oxide in oolitic ironstone. *Journal of Sedimentary Research* **62**, 54–60 (1992).
- [68] Lempart, M., Derkowski, A., Luberda-Durnaś, K., Skiba, M. & Błachowski, A. Dehydrogenation and dehydroxylation as drivers of the thermal decomposition of Fe-chlorites. *American Mineralogist* **103**, 1837–1850 (2018).
- [69] Lalonde, K., Mucci, A., Ouellet, A. & Gélinas, Y. Preservation of organic matter in sediments promoted by iron. *Nature* **483**, 198–200 (2012).
- [70] Faust, J. C. *et al.* Millennial scale persistence of organic carbon bound to iron in Arctic marine sediments. *Nature Communications* **12**, 275 (2021).
- [71] Logan, G. A., Summons, R. E. & Hayes, J. M. An isotopic biogeochemical study of Neoproterozoic and Early Cambrian sediments from the Centralian Superbasin, Australia. *Geochimica et Cosmochimica Acta* **61**, 5391–5409 (1997).
- [72] Fischer, W. W., Hemp, J. & Johnson, J. E. Evolution of oxygenic photosynthesis. *Annual Review of Earth and Planetary Sciences* **44**, 647–683 (2016).
- [73] Garcia, A. K., Cavanaugh, C. M. & Kacar, B. The curious consistency of carbon biosignatures over billions of years of Earth-life coevolution. *The ISME Journal* **15**, 2183–2194 (2021).
- [74] Wilkes, E. B. & Pearson, A. A general model for carbon isotopes in red-lineage phytoplankton: interplay between unidirectional processes and fractionation by RuBisCO. *Geochimica et Cosmochimica Acta* **265**, 163–181 (2019).
- [75] Eglinton, T. I. & Eglinton, G. Molecular proxies for paleoclimatology. *Earth and Planetary Science Letters* **275**, 1–16 (2008).
- [76] Lowell, R. P. & Keller, S. M. High-temperature seafloor hydrothermal circulation over geologic time and Archean banded iron formations. *Geophysical Research Letters* **30**, 1391 (2003).
- [77] Korenaga, J. Archean geodynamics and the thermal evolution of Earth. *Archean Geodynamics and Environments Geophysical Monograph Series* **164**, 7–32 (2006).
- [78] Korenaga, J. Crustal evolution and mantle dynamics through Earth history. *Philosophical Transactions of the Royal Society A: Mathematical, Physical and Engineering Sciences* **376**, 20170408 (2018).
- [79] Wang, C. *et al.* Strong evidence for a weakly oxygenated ocean–atmosphere system during the Proterozoic. *Proceedings of the National Academy of Sciences* **119**, e2116101119 (2022).

Response to Reviewers' Comments

Manuscript ID: 2024-02-02257A-Z

Title: *The geologic history of marine dissolved organic carbon from iron oxides*

Dear Editors and Reviewers,

We sincerely thank you for the thoughtful, rigorous, and constructive feedback provided on the second version of our manuscript, “*The Geologic History of Marine Dissolved Organic Carbon from Iron Oxides*.” We are especially grateful that all reviewers—including the recently added Reviewer 6—have recognized the critical importance of obtaining data-based reconstructions of dissolved organic carbon (DOC) over geological time and have expressed enthusiasm for our approach and findings.

In response to the insightful comments of all Reviewers, we have revised the manuscript to clarify the extent to which our interpretations remain tentative. While our experimental data robustly support a relationship between DOC concentrations and iron oxide co-precipitation, we now more explicitly acknowledge alternative explanations and the inherent uncertainties of deep-time DOC reconstructions. Our revised text details both the strengths of our approach and the limitations imposed by potential diagenetic overprinting, variability in DOC composition, and the possibility that our proxy may bias towards certain portions of the total DOC pool.

Below, you will find a summary of the key revisions we have undertaken, followed by detailed, point-by-point responses to each reviewer comment. Note that (i) the Reviewers' comments are shown in black text, whereas our responses are shown in blue text; and (ii) all reference numbers in this document refer to the order in which they are cited within this document itself, not the order in which they are cited in the manuscript.

Summary of Revisions

In response to the Reviewers' comments—including those of the new Reviewer 6—we have made a number of significant revisions throughout the manuscript. In brief, the key changes are as follows:

- **Clarification of processes governing DOC transformation:** We have revised the discussion on main-text L22–L28 to clearly distinguish between the solubilization of sinking particulate organic carbon (POC) and the microbial carbon pump, emphasizing that the latter specifically involves the microbial conversion of solubilized material into a refractory DOC pool.
- **Rationale for DOC source selection:** As suggested by Reviewer 3, we have added an explicit statement (in the paragraph beginning on main-text L86) detailing why direct experiments with deep-ocean refractory DOC were not possible—namely, the impracticality of processing the thousands of liters of seawater that would be required. We now clearly explain in the main text that our experiments were performed with DOC analogs that capture the dominant reactivity in marine settings.

- **Corrections and units:** We corrected typographical errors (e.g., “timing”, L1156) and have ensured that the text includes clear units (e.g., specifying that “Final hematite DOC/Fe(III)” values are expressed as carbon-to-iron mole ratios).
- **Improvements in data presentation:** In response to suggestions regarding Figure 2a, we have increased the panel width to be two columns instead of one, thus improving readability.
- **Sorption versus co-precipitation distinction:** In response to Reviewer 6’s comments and those from earlier rounds of review, we have clarified that our experiments focus on DOC co-precipitation with newly formed iron oxides rather than on adsorption onto pre-existing mineral surfaces.
- **Addressing potential bias from non-iron phases and POC:** We have expanded our discussion on the potential contributions from accessory minerals and POC. Following the previous round of review, we added new high-resolution Raman microspectroscopic analyses (with a spatial resolution of $\sim 2.6 \mu\text{m}$) and comparative high-resolution SEM images, which we now use to demonstrate that the OC signal within ooids is homogeneous and consistent with co-precipitated DOC rather than localized POC or extracellular polymeric substances (EPS). Regardless, we now further emphasize that our reconstructed DOC concentrations represent maximum estimates, thus allowing for the possibility of minor contributions by POC and/or EPS.
- **Revised discussion on diagenesis:** We have added a more detailed discussion of potential diagenetic and recrystallization effects (including commentary on mimetic recrystallization as pointed out by Reviewer 6) in the Supplementary Discussion (Section 6, specifically 6.4). By combining petrographic screening and high-resolution imaging, we show that our sample set is minimally overprinted and that the observed Fe-OC signals most likely reflect primary or very early diagenetic conditions. Although diagenesis can never be entirely precluded, we again emphasize that we have omitted several clearly diagenetically altered samples during our sample screening, thus providing the most reliable record possible.

Detailed, individual responses to each Reviewer comment can be found on the following pages.

Reviewer #3

Comment 1: I am not trained in the geochemistry that dominates the experiments in this manuscript, nor in the paleo-ocean, which is essential for interpreting the data. However, I greatly appreciate that the authors have done extensive testing of the proxy dynamics and reliability, and that the challenges offered by reviewers more informed about the geochemical uncertainties were (to my eye) largely met with the additional testing done by the authors.

All proxies hold uncertainties, but the achievements demonstrated in this work are important. I am interested in the biogeochemical storyline, which this paper advances. We reviewers and readers cannot require an absolute belief in any paper's findings. This paper offers the reader direct measurements, a first for ascertaining DOC variability in the geologic past, which is a great advance relative to the preceding modeling efforts to address the question. I found the results of the modeling efforts to be highly intriguing and thought-provoking as they rolled out, even though I could not trust any of them because they did not agree with one another (to clarify: here the Reviewer is referring to model efforts of previous studies, not the data interpretation and modeling performed here). (Interestingly, the current work does not support the model results, leaving us with a new end-member of possibilities.) However, the model results pushed our science, as evidenced by this manuscript and the amazing intellectual and analytic efforts to generate it. I do not know how accurate the results are here, but they too have high value in being intriguing and thought-provoking; they too advance our science. Future efforts toward the question will have this important contribution to compare themselves to; at some point, we hope, the results of direct data measurements and modeling approaches will coalesce around a common timeline for DOC in the ancient oceans. We need to support each tottering step toward that outcome. That is how our science works; this manuscript is the next important step toward answering the question.

Response: We thank the Reviewer for this thoughtful and encouraging evaluation of our work. We agree that every proxy carries its own set of uncertainties and that our approach, by providing direct measurements of DOC variability in the geologic past, offers a significant improvement over previous purely model-derived estimates. Our extensive experimental testing and the independent calibration of the DOC proxy were designed to rigorously explore the uncertainties inherent in the system. We too anticipate that future work will further refine both the proxy and corresponding modeling efforts, eventually coalescing around a more unified timeline for DOC evolution in ancient oceans. We appreciate the Reviewer's support in recognizing the broader scientific value of this overall approach.

Comment 2: *Lines 26/27:* The solubilization of sinking POC is not equivalent to the microbial carbon pump (MCP). If the solubilized material (i.e., newly introduced DOC) is converted to a recalcitrant form of DOC via microbial action, then the MCP is in play. It is not solubilization alone that constitutes the MCP, which is how I interpret the text.

Response: We thank the Reviewer for this clarification. In the revised manuscript, we have reworded the passage on L26-27 to explicitly distinguish between the process of solubilization of sinking POC and the subsequent microbial transformation that renders DOC recalcitrant—the latter defining the microbial carbon pump. The revised text now states that solubilization introduces DOC into the water column, while the MCP further transforms this material into more refractory compounds through specific microbial processes. That is, we now state (new text in italics):

L22–L28: “DOC is mainly produced by (i) planktonic communities in sunlit surface waters, where dissolved compounds are either exuded directly by photosynthetic autotrophs or

are released during ecosystem interactions—e.g., primary production, “sloppy feeding”, viral lysis, and microbial loop interactions—and (ii) solubilization of sinking particulate organic carbon (POC) in the ocean interior. *Subsequent transformation of solubilized carbon into more recalcitrant DOC compounds constitutes the microbial carbon pump.*”

Comment 3: *Lines 30-34*: I do not recall that Ref. 15 (Follett et al) estimated the amount of DOC in the ocean formed by the MCP; they estimated the amount of modern DOC in the deep ocean. The first estimate of DOC formation in the deep ocean by the MCP (at about 25%) that I am aware of was Benner and Herndl (2011) “Bacterially derived dissolved organic matter in the microbial carbon pump”. In N. Jiao, F. Azam, & S. Sanders (eds.) *Microbial Carbon Pump in the Ocean*, Science/AAAS (pp. 46–48) Science/AAAS. Shortened abstract: “...Seawater bioassay experiments demonstrate that bacteria rapidly transform labile DOC to semilabile and refractory forms, suggesting enzymatic activity plays an important role in the transformation process. ...the molecular signatures of the transformed DOC are observed throughout the ocean water column. Bacterial transformations in the microbial carbon pump (MCP) have sequestered about 10 Pg of semilabile DOC and about 155 Pg of refractory DOC in the global ocean. The annual production of semilabile and refractory DOC in the upper ocean MCP is estimated to be 0.74 to 2.23 Pg and 0.008 to 0.023 Pg, respectively.”

Response: We appreciate the Reviewer’s detailed attention to the literature. The text now explicitly cites Benner and Herndl (Ref. ¹) when discussing the fraction of DOC formed through the microbial carbon pump.

Comment 4: Please include a statement in the paragraph beginning on Line 84 on why you could not conduct the experiments with deep ocean ‘refractory DOC’ (RDOC) as the source material. Somewhere I read in this submission that it would take ‘thousands of liters’ to use unaltered seawater but a detailed reasoning is absent. There would have been no better source for testing the proxy than RDOC, had it been a possible experiment.

Response: We thank the Reviewer for this constructive suggestion. In the revised manuscript, we have added a statement in the paragraph beginning on L86 explaining that marine DOC exists at extremely low concentrations, making isolation and use in laboratory experiments logistically prohibitive. Specifically, obtaining adequate quantities of DOC would require processing thousands of liters of seawater—a task that is currently impractical given our experimental setup. We now make it clear that our choice of DOC source materials was constrained by these logistical and analytical limitations, while still capturing the dominant reactivity of DOC in relevant marine settings. Specifically, we now state:

L94–L96: “Because marine DOC naturally occurs at low concentrations, sources (i) and (ii) were utilized to overcome the logistically impractical need to process several thousands of liters of seawater to obtain adequate substrate.”

Comment 5: *Line 1156*: Spelling of ‘timing’.

Response: We thank the Reviewer for catching this error. The spelling mistake has been corrected in the revised manuscript.

Comment 6: *Line 1192*: “Final hematite DOC/Fe(III)”. In this case and elsewhere, please include units somewhere in the caption or the axes.

Response: We appreciate the Reviewer’s note regarding units. In the revised paper we explicitly note that it is the carbon-to-iron mole ratio, thus ensuring that the data are unambiguously represented. For example, we now write:

L77–L78: “...resulting Fe-OC content depends on carbon-to-iron mole ratio [DOC/Fe(III)] of the initial solution”

Reviewer #4

Comment 1: I commend the authors on their thorough revisions; I feel that they’ve addressed my concerns as well as can be done. I’ve also carefully read their responses to the other reviews. At least as far as my understanding allows (some of the concerns are somewhat outside my direct expertise) I also feel that the authors have done a decent job.

Response: We thank the Reviewer for the positive and encouraging feedback. We are sincerely pleased that our revisions have satisfied your concerns and that our responses were helpful in addressing the complexities of this work.

Comment 2: I’ve one remaining issue however, that I feel the authors would do well to address, at least to the extent that they can, in this current manuscript (I acknowledge that they intend to follow up on this issue in a subsequent manuscript). The issue relates to the concerns of Reviewer 5, about whether iron (oxyhydr)oxides capture a bulk DOC signal. As far as most of the evidence presented by the authors goes (more on this below), I’m inclined to agree with them that their ooids appear to capture a bulk DOC signal – or at least they appear to not be capturing only some specific subset of the bulk DOC that is operationally defined based on its lifetime, like SL-DOC. But I don’t think the authors (yet) consider whether their ooids might be enriching some specific subset of the bulk DOC that possess a particular chemical composition. It’s well known that during Fe-OC precipitation certain DOC components are preferentially taken up over others (thereby concentrating certain DOC components in the precipitates – see work like Eusterhues et al., 2011, ES&T and Curti et al., 2021, Commun. Earth Environ. – and many others). As such it seems plausible that the ooids might preferentially take up a subset(s) of the bulk DOC that is enriched in certain molecular classes or functional groups.

Response: We thank the Reviewer for raising the possibility that iron (oxyhydr)oxide precipitation might preferentially co-precipitate with a specific subset of ambient DOC rather than capturing the entire mixture equally. We are aware that several studies (e.g., Refs.^{2,3}) have investigated the importance of certain organic functional groups in setting the strength of interactions with ferrihydrite, potentially enhancing uptake of specific DOC components (Note: in contrast to this manuscript, they did not study crystalline goethite and hematite). We completely agree with the Reviewer that this phenomenon may happen in our iron ooids, and we regret that studies such as Eusterhues et al. and Curti et al. were not cited in our original manuscript (although we often reference this work when presenting on this topic at conferences!). We have now added these citations at appropriate locations throughout the text (e.g., L77).

Still, although not explicitly stated in our original text, this phenomenon was a large motivator when choosing our experimental design. That is, we purposefully designed our experiments with multiple DOC sources—including natural-marine analogs and FA—that are expected to contain differing amounts of various functional groups. That way, differences in OC loadings could ideally be related to DOC molecular properties. These results are implicitly shown in Fig. ED23 and discussed in the following supplementary text:

L2352–L2361: “Interestingly, the quantitative Fe-OC wt % for both minerals at a given DOC/Fe(III) ratio depends linearly on DOC slope ratio (S_R), a proxy for the degree of processing by heterotrophic respiration and/or photodegradation (see “Molecular and functional characteristics of experimental DOC”, above)⁴. For example, Fe-OC loadings in goethite with

DOC/Fe(III) = 0.16 to 0.18 decrease moving from FA ($S_R = 0.67$) to M-DOC ($S_R = 1.29$) to C-DOC ($S_R = 2.20$) with a slope of $-0.47 \text{ wt } \% S_R^{-1}$ ($n = 3, r^2 = 0.995$; Fig. ED23). Similarly, loadings in hematite with DOC/Fe(III) = 0.06 to 0.07 decrease with a slope of $-0.08 \text{ wt } \% S_R^{-1}$ ($n = 3, r^2 = 0.993$; Fig. ED23). These results indicate that—all else being equal—more processing by heterotrophic respiration or photodegradation will lead to lower Fe-OC loadings. Future work is warranted to determine the chemical mechanism of this result.”

Furthermore, we originally attempted to answer the question of preferential uptake using compound-specific techniques such as excitation–emission matrix spectroscopy (EEMS) and high-resolution Fourier Transform Ion Cyclotron Resonance Mass Spectrometry (FT-ICR-MS). This was motivated for the exact reasons that the Reviewer articulates here: we were interested in understanding if specific functional groups, compounds, or compound classes were preferentially co-precipitated with the iron oxides. Unfortunately however, these efforts were hampered by analytical challenges (notably, interference from residual dissolved iron causing severe ion suppression in the MS source), preventing us from fully resolving individual molecular contributions. This analytical difficulty is not unique to our study; similar challenges have been reported in the literature when targeting hydrophilic DOC fractions (e.g., Ref.²).

In summary, we fully agree with the Reviewer that preferential uptake likely plays a role in setting the exact Fe-OC loadings, but we emphasize that all tested DOC substrates show the same general trend of increasing Fe-OC with increasing DOC/Fe(III) ratios, supporting our overall interpretations. Nevertheless, we have now updated the main text to explicitly state the importance of functional groups in setting the exact Fe-OC loading curves (new text in italics):

L79–L85: “However, several factors currently hinder the utility of Fe-OC as a proxy for past DOC signals, including unknown: ... (iii) impacts of marine DOC molecular composition [c.f., soil humics in Ref.⁵], *functional-group diversity*^{2,3}, and environmental conditions (e.g., temperature, pH, dissolved silica concentration).”

L125–L129: “Finally, when comparing across sources, Fe-OC loadings depend strongly on DOC slope ratio (S_R), an absorbance-based measure of molecular weight and aromaticity (Fig. S23)⁴. *This likely reflects the known importance of DOC methoxy and carboxyl groups in determining co-precipitation strength and loading*^{2,3}. We therefore consider several molecular composition evolutionary scenarios when reconstructing Earth-history DOC records.”

We also acknowledge that future work (possibly involving improved compound-specific extraction protocols or in situ characterization methods) would be valuable for further constraining these processes.

Comment 3: For the DOC sources used, it should be pretty easy to test whether the bulk DOC is taken up as is, or whether certain components of the bulk DOC are preferentially associated with the minerals – i.e., perform analyses of the chemical composition of the DOC before uptake, and after uptake (again, see Eusterhues et al., 2011, ES&T as an example). I appreciate that the authors have considered this issue to some extent via their FA experiments BUT I’m not convinced that FA represents a “singular molecular structure” – FA extracted from natural sources represents a mixture of different organic acids, while commercial FA powders can be more homogeneous – the authors used a commercial FA powder (Mark Nature) and confirmed its purity using FTIR – but this simply shows that the powder doesn’t contain substantial quantities of something else, not that it has a single molecular structure – I can’t find any information on the Mark Nature website about the molecular structure.

Response: As mentioned in the response to the previous comment, we agree that a compound-specific characterization of DOC uptake would provide valuable insights into the mechanisms

controlling Fe-OC formation. To address this, we attempted to perform analyses using both EEMS and high-resolution FT-ICR-MS on the DOC before and after co-precipitation. Unfortunately, interference from residual dissolved iron led to severe ion suppression in the MS experiments, and the EEMS results did not yield conclusive differences. Although the FA we used appears relatively homogeneous via FTIR, we acknowledge that it does not constitute a singular molecular entity (but rather represents the same class of molecules with similar properties and functional groups). To reflect this, we have updated the following text (new text in italics):

L2371–L2375: “Additionally, it is unlikely that fractionation results only from preferential co-precipitation of a compositionally and thus isotopically unique subset of the overall DOC reservoir, as evidenced by the observation that FA—which is comprised of a single molecular structure *or small group of compositionally similar molecular structures*—exhibits a different sign and magnitude of fractionation when co-precipitated with goethite and hematite.”

Finally, we recognize the importance of developing improved analytical protocols to resolve individual molecular contributions and plan to focus on this in future work.

Comment 4: It might be too much to ask the authors to perform additional experiments to test whether their bulk DOC is chemically fractionated on uptake, but I do feel that this is an important issue and one that the authors need to acknowledge and discuss, in terms of whether and how this might affect their DOC proxy. If this issue is unlikely to substantially impact their DOC proxy then I think it would be helpful for the authors to point this out. But is it possible, for example, that changes in the Fe-OC loading reflects changes in DOC concentration but also changes in DOC composition through time?

Response: We again thank the Reviewer for highlighting this important issue. As stated above, we originally attempted to test this exact question using several analytical tools, but these were unfortunately hampered by high residual concentrations of dissolved iron. We therefore proceeded without these data, and we instead attempted to account for any potential uncertainty due to preferential uptake through our experimental design and Monte Carlo modeling approach. That is, our experiments were intentionally designed to use three distinct DOC sources—modern marine analog (M-DOC), cyanobacterial leachate (C-DOC), and fulvic acid (FA)—to capture a range of molecular compositions that might be present in natural systems. Although these DOC types exhibit different molecular properties and methoxy/carboxyl richness, our calibration experiments demonstrate that the overall Fe-OC loading trends with DOC/Fe(III) ratios remain robust across these sources (see Figs. ED18-ED19). In other words, even if preferential uptake of specific DOC components does cause shifts in the molecular composition of the precipitate—as predicted—we do not expect these differences to translate into *systematic* errors in our reconstructed trends, and we account for any uncertainty using our Monte Carlo approach.

In our revised manuscript, we have expanded the discussion to explicitly acknowledge and consider the potential impact of DOC compositional changes during Fe-OC co-precipitation. In addition to the line-changes quoted above, we now also state (new text in italics):

L12–L15: “Given the observed Fe-OC loading dependency on *methoxy/carboxyl richness*^{2,3} and thus S_R (Fig. ED23), we first constrain DOC molecular composition. Our model thus requires one free input parameter—the fractional contribution of each DOC end member through time”

and:

L224–L225: “Solving our model also requires five simplifying assumptions (Supplementary Discussion): (i) DOC is always a mixture of M-DOC, C-DOC, and FA (i.e., ignoring additional, *potentially compositionally unique* sources).”

To briefly summarize the results, we find that DOC compositional differences could impact reconstructed concentrations, but that the importance of this is relatively small compared to observed temporal trends. This is stated in the main text as:

L231–L234: “We tested each assumption for each $f^i(t)$ scenario; resulting uncertainty could lead to a $\sim 2\times$ bias in [DOC]* and $\sim 10\%$ bias in $\delta^{13}\text{C}_{\text{DOC}}$ (Supplementary Discussion; Figs. ED127–130). Still, all temporal trends are qualitatively robust to assumption-induced bias.”

Future work—including compound-specific analyses—will be needed to fully resolve the extent of preferential DOC uptake, but based on our present results, compositional shifts are unlikely to substantially affect reconstructed DOC concentrations over geological timescales.

We again thank the Reviewer for raising these points, and we hope the revised text now provides a clear and satisfactory explanation.

Reviewer #5

Comment 1: The authors went above and beyond in their detailed reply to my earlier review, and I find their argument compelling.

Response: We thank the Reviewer for the generous and encouraging feedback. We are delighted that our detailed responses and additional data have been compelling and that our arguments are well-received.

Comment 2: The revised paper reads very well, and I am nearly wholly on board with its findings.

Response: We appreciate the Reviewer’s positive evaluation of the revised manuscript. It is encouraging to know that our extensive revisions have resulted in a clear and well-structured paper that is nearly wholly supported.

Comment 3: I have only two minor additional comments:

- (i) Please make the panels in Figure 2 much larger—in particular, increase the time-axis resolution so that more details can be seen.

Response: We thank the Reviewer for this suggestion. In response, we have revised Figure 2 to increase the panel sizes and to enhance the resolution of the time-axis and improve overall readability.

- (ii) While only a small component of the manuscript, I feel that the authors may be over-interpreting their data. For example, on re-reading the manuscript, I wonder if the evolution or rise of sponges might be a simpler and plausible explanation for the Neoproterozoic decline in DOC, with the mid-Paleozoic return to modern values reflecting increased DOC production (perhaps via enhanced ecosystem complexity and grazing).

Response: We thank the Reviewer for this thoughtful alternative interpretation. In the revised manuscript, we now include a brief discussion to acknowledge that several possibilities exist to explain our observed trends—including the rise of sponges and enhanced ecosystem complexity (new text in italics):

L273–L275: “Third, the Phanerozoic is described by fully oxygenated deep oceans⁶ and continued growth of *marine ecosystem complexity* and organism size—including *grazing, biomineralization, and the rise of sponges* (Fig. 3a)^{7,8}.”

It is not our intention to over-interpret our results; rather, we hope to present all possible scenarios that are potential drivers of our observed trends. We hope that the updated text—although short due to space constraints—better reflects this intention.

Reviewer #6

Comment 1: **Overall Assessment:** Please find enclosed my review of Galili et al. This paper examines an important and highly underdetermined problem in Earth History—namely, the amount of dissolved organic carbon (DOC) in the oceans over geologic time. As the authors explain at the outset, approximately 20 years ago Rothman et al. proposed that variations in the $\delta^{13}\text{C}$ of carbonates—especially the pronounced depletions in the Neoproterozoic—could be explained by the oxidation of a larger dissolved organic pool in the ocean. This mechanism, being inherently non-steady state (and previously subject to criticism), has also been linked to the rise of an oxygenated world.

Much ink, including contributions in *Nature*, has been spilt on the interpretation of these carbon isotope excursions. The central challenge is that there is no direct record of this “phantom” organic pool. Even in the absence of this debate, our limited understanding of the DOC term in the Phanerozoic (or any other period) makes constraining it a significant step toward deciphering Earth’s evolution. In other words, the problem is both timely and compelling.

The authors propose a solution by measuring the organic carbon content of iron ooids as an indicator of the DOC pool. The underlying idea is that dissolved organic carbon—as well as many other chemical species—readily sorbs to iron oxide minerals, and as ooids grow, they can entrap this sorbed material. Similar approaches have been applied to iron minerals, for example in attempts to back-calculate phosphate concentrations (see the work of Don Canfield and collaborators).

Experimental work convincingly demonstrates a correlation between dissolved organic carbon concentrations and uptake by synthetic iron ooids. Moreover, when exploring the geologic record, the authors report relatively little variation in organic carbon content. At face value, this result appears to challenge the notion of a substantially larger DOC reservoir in the Neoproterozoic. While they discuss finer “wiggles” in the data, the overarching message is that the DOC reservoir has remained largely unchanged through time.

The validity of the approach rests on two key assumptions: (1) that the only organic carbon incorporated into the iron minerals of the ooids is sorbed material (i.e., there is no contribution from particulate organic carbon or detrital organic matter bound to other minerals), and (2) that the organic carbon content is preserved over time without significant alteration from diagenesis or catagenesis. I remain unconvinced regarding these latter issues, as the paper does not adequately address either contamination by particulate organic carbon or the potential effects of diagenetic removal.

Response: We sincerely thank the Reviewer for their thoughtful, comprehensive, and constructive evaluation of our manuscript. We greatly appreciate the recognition of the significance and timeliness of the underlying problem—namely, the reconstruction of ancient marine DOC concentrations and their relevance to global carbon cycle dynamics. We also appreciate the

Reviewer's acknowledgment of the novelty of our approach, as well as the potential value of using iron ooids as archives of DOC through time. We fully agree with the Reviewer that interpreting carbon isotope excursions in Earth history remains a central challenge in geochemistry, and that independent geochemical constraints on DOC pools are urgently needed. Finally, we recognize and appreciate the Reviewer's deep familiarity with the geochemical literature and with the caveats that such reconstructions must navigate.

In this response, we have carefully addressed each of the Reviewer's specific concerns. In particular, we have substantially expanded our discussion and supporting evidence regarding the possible contributions of POC, the potential effects of diagenesis and recrystallization, and the distinction between co-precipitation and adsorption mechanisms. Where appropriate, we have clarified methodological details and revised key sections of the manuscript to reflect these critical insights. We believe these revisions significantly strengthen the manuscript and improve its clarity and rigor. We thank the Reviewer once again for their thoughtful comments and the opportunity to improve the manuscript.

Comment 2: Discussion of sorption in general: When considering sorption to minerals, one should focus not on ratios such as OC/Fe³⁺ or simple weight percentages but rather on the amount of reactive surface area available and the number of sorption layers that can form. This thermodynamic variable is only mentioned once (on line 2288), implying that the model assumes a direct correlation between surface area and weight percentage that is independent of the growth environment. Although this may be a reasonable approximation, a clear exposition of the relationship between the fundamental sorption process and the iron weight percent is needed. An illustrative example can be found at <https://doi.org/10.1016/j.geoderma.2013.05.026> (Ref.⁹).

Response: We sincerely thank the Reviewer for raising this important point regarding the mechanisms underlying organic carbon (OC) interactions with minerals. However, we respectfully clarify that our manuscript specifically focuses on the co-precipitation of DOC with iron (hydr)oxide minerals (i.e., goethite and hematite), rather than adsorption of DOC onto the surfaces of pre-existing minerals (as discussed in Ref.⁹, provided by the Reviewer). The Reviewer appears to conflate adsorption with co-precipitation, which differ fundamentally in their mechanisms and thermodynamic properties—we apologize if this was not clearly articulated in our original text.

In contrast to adsorption onto pre-existing mineral surfaces (as studied, for example, in Ref.⁹, provided by the Reviewer), co-precipitation involves the concurrent formation of organic-mineral associations during mineral nucleation and crystal growth. This mechanism is inherently distinct from adsorption, which relies predominantly on the availability and reactivity of existing mineral surface sites and thus directly relates to parameters such as surface area or the number of sorption layers. Specifically, we emphasize two key differences:

- (i) **Mineralogy:** Our study deals exclusively with the iron (hydr)oxide minerals hematite and goethite, whereas the reference provided by the Reviewer—as well as numerous other publications—mainly investigate phyllosilicate clays (e.g., kaolinite, illite, smectite, although mixed with iron (hydr)oxides in the case of Ref.⁹). Importantly, clays inherently possess different chemical properties, surface charge distributions, and sorptive behaviors than iron oxides. For example, clay minerals typically have permanent structural negative charges arising from isomorphous substitutions within their layered silicate structure¹⁰, leading to strong electrostatic interactions with positively charged organic molecules. In contrast, iron (hydr)oxides possess variable surface charges depending strongly on solution pH and ionic strength, with surface reactivity governed

by hydroxyl functional groups that protonate or deprotonate readily, resulting in highly dynamic surface charge distributions¹¹. Thus, the sorptive behaviors, affinity for organic compounds, and resulting organic-mineral associations differ substantially between clay minerals and iron (hydr)oxides. Therefore, sorption characteristics documented for clays (such as those in Ref.⁹) are not directly transferable to our system of iron (hydr)oxide minerals.

- (ii) **Mechanism of organic carbon-mineral interaction:** Our experiments quantify OC loading during mineral precipitation (i.e., co-precipitation), a process controlled by solution chemistry, saturation state, and crystal growth kinetics. In contrast, the referenced study⁹ investigates sorption equilibrium onto pre-existing mineral surfaces, which is primarily controlled by the availability of reactive surface sites (see, for example, Refs.^{2,3,5,12} for discussion on these mechanistic differences). Therefore, sorption theories focused on surface area and sorption layer saturation do not directly govern our measured OC loading, which follows an empirical power-law relationship with DOC/Fe(III) concentration ratio (as first articulated in Ref.⁵ and detailed mathematically in our Supplementary Material). In summary, while the importance of the adsorption has long been recognized as an important driver of OC preservation in many environments (e.g., dating back to the “surface mono-layer” hypothesis of Ref.¹³), the governing controls are fundamentally different than those for co-precipitation.

We fully agree with the Reviewer’s general point that surface area and reactive sites are central to understanding adsorption processes. Nevertheless, because our focus is explicitly on co-precipitation rather than sorption onto existing surfaces, our modeling approach necessarily employs OC/Fe(III) ratios and empirical power-law relationships rather than surface-area-normalized concepts typically used in adsorption equilibrium studies. To explicitly clarify this critical distinction, the main text of our manuscript contains the following statements:

L101–L102: “...we additionally tested the effect of removing ferrihydrite-bound and adsorbed DOC, as this is unlikely to survive in the geologic record (Methods).”

and:

L110–L116: “...removing ferrihydrite-bound and adsorbed DOC decreases observed loadings by up to 50 % and shifts $\Delta^{13}\text{C}$ by up to 10 ‰ relative to “raw” fractions (Supplementary Discussion; Figs. 16-17). These results ... highlight the importance of removing non-geologically preserved material prior to building calibration curves.”

In addition, we now clearly state in the Methods section under “Removal of ferrihydrite-bound and adsorbed OC”:

L747–L751: “Finally, one fraction was treated with 1M HCl solution as described for –Fh/PC and was subsequently treated with a 1M MgCl₂ solution at room temperature for \approx 1 hour to chelate and remove any loosely-bound (adsorbed) Fe-OC complexes on the crystalline mineral surfaces.”

Finally, we have now added the following lines to the Supplementary Discussion section “Effect of rinsing and desorption procedure” (new text in italics):

L2272–L2284: “Lastly, we assessed how removing ferrihydrite-bound and adsorbed OC impacts Fe-OC signals (see “Removal of ferrihydrite-bound and adsorbed OC”, above, for method details). This rinsing and desorption procedure is critical for isolating and analyzing only OC that is co-precipitated with crystalline iron oxides. In contrast, ferrihydrite-bound

and/or adsorbed OC is not expected to persist over geologic timescales^{5,12} and should be removed before utilizing calibration curves derived from synthesis experiments to reconstruct Earth-history records. *We emphasize that our approach specifically describes OC co-precipitation with newly formed iron (hydr)oxide minerals, a process that is fundamentally different from adsorption onto pre-existing surfaces where surface area and reactive site densities directly dictate OC loading capacities.* Insufficient removal of non-geologically preserved OC when generating calibration curves could bias results if this material significantly alters predicted Fe-OC loadings or ¹³C compositions; we explicitly test this across a range of experimental conditions...”

We fully agree with the Reviewer that the distinguishing and separating adsorbed and co-precipitated OC is critical for the validity of our approach, and we emphasize that this Supplementary Discussion section thoroughly details the impact of removing adsorbed OC on our results. We thank the Reviewer again for highlighting this potential source of confusion and for providing an opportunity to clarify these important distinctions within the revised manuscript.

Comment 3: Way the data is shown: Figure 2A is pivotal in terms of documenting the loading. The data are presented in log space, and I have several concerns regarding this choice. First, I recommend that the figure be rendered much wider to clearly display the data points. Upon replotting the provided Excel dataset, I observed that the total range spans from approximately 0.03 to 0.96 wt.%, about 1.5 orders of magnitude. In my opinion, such a range does not require a logarithmic transformation; using a log plot introduces artificial differences between the low and high ends of the data, particularly given that the full range is nominally 0–100%. A linear-scale plot (with time on the x-axis and wt.% on the y-axis) would better reveal the scatter within specific time frames, the relative paucity of Precambrian points, and the similarity between maximum values in the Mesoproterozoic and the lowest modern values.

Response: We sincerely thank the Reviewer for raising this important point regarding the representation of data in Figure 2A. We fully acknowledge that plotting data in log scale might initially appear unconventional for a range spanning approximately 0.03 to 0.96 wt.%. We clarify, however, that the choice of a logarithmic scale on the *y*-axis (*not* the *x*-axis) is fundamentally dictated by the mathematical relationship used in our modeling framework, explicitly detailed in our Supplementary Materials (section “Mathematical derivation”).

Specifically, as presented in the Supplementary Materials, Fe-OC loadings ($w^{i,m}(t)$, in wt.%) are related to the DOC/Fe(III) concentration ratio ($x^i(t)$) through a power-law relationship:

$$w^{i,m}(t) = \beta_0^{i,m} [x^i(t)]^{\beta_1^{i,m}}, \quad (1)$$

where $\beta_0^{i,m}$ and $\beta_1^{i,m}$ are empirically derived constants. The inherent mathematical nature of this power-law relationship requires a logarithmic representation on the *y*-axis in order to proportionally intuit changes in DOC/Fe(III) ratio as a function of measured Fe-OC loadings. Put differently, plotting Fe-OC in linear space would obscure the non-linear relationship between $w^{i,m}(t)$ and $x^i(t)$, particularly for the hematite portion of our record (see, e.g., Fig. ED18), even though $x^i(t)$ is what we’re ultimately interested in reconstructing. The importance of plotting Fe-OC data in log space on the *y*-axis can thus be seen when rearranging Eq. 1 to predict DOC/Fe concentration ratios:

$$x^i(t) = \left(\frac{w^{i,m}(t)}{\beta_0^{i,m}} \right)^{\frac{1}{\beta_1^{i,m}}}. \quad (2)$$

That is, a change in $w^{i,m}(t)$ would only linearly correspond to a change in $x^i(t)$ if $\beta_1^{i,m} \equiv 1$, which is not the case. To summarize, plotting Fig. 2a data linearly would obscure this essential

mathematical relationship, hinder accurate interpretation, and impede straightforward inversion to original, non-logged DOC/Fe(III) data as well as Earth-history [DOC]* predictions shown in Fig. 3a.

Nevertheless, we fully agree with the Reviewer's practical suggestion regarding figure clarity (also articulated by Reviewer 5, above). To better highlight the individual data points, we have significantly increased the width of Fig. 2 in the revised manuscript. We greatly appreciate the Reviewer's constructive suggestions, which have helped clarify and strengthen the visual and analytical rigor of our presented results.

Comment 4: Organic carbon in non-hematite portions of the ooids? The most important requirement of the paper is that the organic carbon must be sorbed carbon that is structurally bound in the mineral lattice. The authors deal with this by doing Raman spectroscopy and looking if organic carbon is or is not present in ~ 3 micron size pixels. I didn't see this data for all samples, nor an example of something where POC is present (maybe I missed it). But what worries me about this approach is that it assumes there are not a mixed sources where POC and amorphous (previously sorbed) OC are both present. I'm not sure if these were generated for all samples, but in looking at figure 1D (Aseri formation), my worry is that the core, which is not the concentric ooid part I think they care about, has organic carbon in it. I can't tell from this figure if the core is also an iron oxide, but regardless, that core is not what they want and has organic carbon.

Additionally, the samples are generally not pure iron oxide. Based on the data provide, here is the wt.% of the samples studied increasing order **Note: this figure is not entirely accurate; see response below**. About 1/3 of samples have more than 20% other stuff, and about half 15% other stuff. If these are clay minerals with 1% organic carbon, they could contribute easily 0.1 to 0.2 wt% to the total. We do not know what these materials are, but they could easily skew the results. I note that the authors consider this in the text, but assume the accessory phases have no organic carbon. I do not believe they provided evidence for this. Regardless, that some samples are 25% something else, is a serious issue if they contain any organic carbon that I strongly recommend the authors deal with.

Response: We sincerely thank the Reviewer for highlighting this important issue, and we fully agree with the concerns raised regarding the potential presence of OC in non-iron (hydr)oxide portions of the ooids. It is precisely this issue that motivated the extensive petrographic analysis of every sample in our record, including new analyses such as Raman spectroscopy that were added after the first round of review (which we recognize this Reviewer was not part of). We articulate in detail below how we have addressed this concern (Note that we focus here on ooid cores and their mineralogy, and we address the issue of POC in our response to subsequent comments):

Firstly, we clarify that there appears to be a minor misunderstanding: the figure plotted by the Reviewer was plotted using the calculated percent hematite column, which inadvertently includes goethite-containing ooids. To resolve this confusion, we refer the Reviewer explicitly to our supplementary figure (Fig. ED125, reproduced below), which clearly distinguishes between hematite and goethite ooids based on XRF-derived iron content. Nevertheless, the Reviewer is correct that some samples can contain up to ~ 25 % by weight of other minerals.

Second, we address the composition of ooid cores, which the Reviewer points out could provide additional OC if composed of other (OC-bearing) minerals, most importantly phyllosilicate clays. We fully agree with this concern in theory, which is why we performed optical microscopy, SEM-EDS, and XRD—both for bulk rock and after extracting individual ooids—

1

2 **Figure 1: Fraction of iron oxide in each ooid included in our record**, separated by miner-
 3 alogy and reported as violin plots. Percentages are derived from XRF-based Fe content, scaled
 4 by the fractional wt % Fe in each ideal mineral formula [i.e., $\text{FeO}(\text{OH}) = 63 \text{ wt } \% \text{ Fe}$; $\text{Fe}_2\text{O}_3 =$
 6 $70 \text{ wt } \% \text{ Fe}$]. Iron oxides are the dominant ooid mineral phase in all cases.

on every single sample included in our record (see Figs. ED25–ED123). For the vast majority of samples (including the Aseri Fm, Fig. 1d pointed out by the Reviewer), ooid cores are also composed iron (hydr)oxides—likely fragments of broken ooids forming in the same environment. There are, however, *some* samples that contain *some* ooids with non-oxide cores. In the majority of these cases, non-oxide cores are composed of andesite (e.g., modern Panarea and Indonesian samples; Figs. ED26, ED29) and quartz (e.g., Westmoreland and Kirkland Fms, Clinton Group, Fig. ED91; McClure Fm, Katherine Group, Fig. ED114; Sherwin Ironstone, Fig. ED118-119). These samples—particularly those with andesitic cores—contribute most of the low-oxide fraction points observed in Fig. 1 of this review (andesitic core sample wt. fraction goethite: 0.77 ± 0.05).

None of the oxide-, andesitic-, or quartz-core ooids contains any detectable phyllosilicate minerals (i.e., no peaks in the XRD ~ 10 to $15^\circ 2\theta$ range; see Figs. ED25–ED123). That said, there does exist one formation—the Presles Fm.—in which ooid cores contain significant kaolinite and/or chamosite (Fig. ED114). Interestingly, this formation also displays the lowest-hematite fraction points observed in Fig. 1 of this review (0.72 ± 0.03 wt. fraction) and deviates from our overall Fe-OC wt. % record trend (Fig. 2). We discuss this formation in more detail below.

Finally, we address the potential for cores to contribute OC to our overall signals. For samples with iron (hydr)oxide-containing cores, this OC will be identical to that in ooid laminae—that is, cores will also contain the signals of interest for our study. For samples with andesitic and quartz cores, we do not expect these minerals to contribute any OC since they are volcanic in origin and/or contain very low surface area. Our measured Fe-OC wt. % values and subsequent calculated [DOC]* record should thus be up-scaled accordingly to account for and exclude this weight fraction (see, e.g., Fig. ED129 and Supplementary Discussion L3522–3534).

In contrast however, the Presles Fm. could contain significant clay-adsorbed OC. This is an interesting point that had escaped our initial attention—that is, the one formation that

deviates from our record also happens to be the one formation with clay-rich ooid cores. In our original text, we were careful not to over-interpret this particular formation (see, e.g., L239–243), although we did not provide an explicit mechanism as to why this formation may deviate. In our revised text, we have now added the following lines to address this (new text in italics):

L245–L249: “Additionally, the Devonian-aged Presles Fm predicts lower [DOC]* and $\delta^{13}\text{C}_{\text{DOC}}$ relative to immediately older and younger formations, despite a lack of *obvious* petrographic evidence for exclusion (Supplementary Discussion, Figs. ED87–ED89). We therefore cannot determine if this represents *contamination by accessory mineral phases*, a local signal, or a transient return to Neoproterozoic conditions.”

and

L3545–L3550: “*This is supported by our XRD and elemental mapping results, which clearly identify most samples as containing iron (hydr)oxide cores, with a subset containing cores composed of andesite or quartz (Fig. ED25–ED123). The only samples deviating from this trend are those of the Presles Fm., which contain cores that can contain significant kaolinite and/or chamosite contributions (Fig. ED87–ED89).*”

To summarize, our extensive petrographic assessment of all samples allows us to confidently conclude that all ooid cores other than those of the Presles Fm. are either themselves composed of iron (hydr)oxides or of the OC-free minerals andesite and quartz. In contrast, the Presles Fm. may contain clay-adsorbed OC, and we now explicitly state this in the main text so as to not over-interpret the results of this particular formation. We thank the Reviewer again for bringing this important issue to our attention.

Comment 5: Presence of particulate organic carbon: Another concern is the presence of particulate organic carbon trapped in the ooids. The authors again argue this is not the case based on the Raman data. I’m not sure at the 3 micron scale this could be seen, and the SEM images provided are not zoomed in enough for me to see. The way I would look for particulate organic carbon is via TEM imaging in order to zoom in. I did a search of “TEM and iron ooids and organic carbon” on google scholar and a number of studies came up. Here is an example from the Bakchar formation (<https://doi.org/10.1016/j.jseae.2022.105361>), which was studied here as well.

I’m not sure who is right, but Galili says EPS is not present, but I do not believe their imaging or Raman can test this as they are too zoomed out to see the patchiness at the sub-micron level needed (i.e., by TEM). But the study referenced above finds the patchy organic carbon and interprets it as EPS from biological activity where the ooids are forming. If right, and generalizable, it would violate the primary assumption of the paper. As another example from Lin et al. (<https://doi.org/10.1016/j.gr.2019.06.004>) for Paleoproterozoic ooids from the Chuanlinggou formation (also studied here), they interpreted the structure below in yellow as ‘mucus like filaments’ and in general state the following: SEM and EDS analyses show that these micrometer-scaled laments are enriched in organic matter and have a mucus-like morphology (Fig. 6B), thus providing further evidence that they were derived from bacterial remains, possibly from EPS

Again, I’m not sure who is right, but there are examples in which samples were studied at high resolution and interpreted to have EPS present, contra what is stated in this paper.

Response: We sincerely thank the Reviewer for highlighting this important concern. We fully agree that, in natural environments, iron oxides can indeed incorporate organic carbon through both DOC uptake and POC, including microbial biofilms or extracellular polymeric substances (EPS). To comprehensively address this issue, we articulate the following points:

- (i) **High-resolution imaging and Raman Microspectroscopy:** Following the previous review round, we have significantly expanded our analyses by conducting additional high-magnification Raman microspectroscopy (now shown in Fig. 1 and Supplementary Discussion). These Raman maps and line-scans, obtained at $\sim 2.6 \mu\text{m}$ spatial resolution, consistently reveal a continuous, homogeneous distribution of OC associated directly with ooid laminae, indicative of structurally bound DOC (we also note that the Aseri Fm. shown in Fig. 1d contains a goethite core; see our response to the above comment). Crucially, Raman spectroscopy clearly differentiates regions such as silica-rich rims (darker regions; Fig. 1e of the manuscript), which consistently lack detectable OC. This homogeneous distribution strongly contrasts with the discrete, patchy filamentous morphologies reported as EPS-derived OC by Ref.¹⁴ (cited by the Reviewer). This evidence significantly strengthens our conclusion that observed OC is primarily associated with goethite and hematite in the Fe ooids studied. Nevertheless, we fully acknowledge and share the Reviewer's concern that accessory phases could contribute trace OC.

That said, the Reviewer is correct that $\sim 2.6 \mu\text{m}$ might not be fine enough to detect EPS patches and filaments if present on sub-micron scales. However, our SEM images do contain the necessary spatial resolution to detect these features. To explicitly illustrate this, we have prepared comparative high-resolution SEM images to show our results side-by-side with those of Ref.¹⁴, matched at identical spatial scales (Fig. 2 of this review). Unlike the findings reported in Ref.¹⁴, our ooids (including those from the same formation as studied by Ref.¹⁴) consistently lack localized EPS-like features. Our observations are additionally supported by recent high-resolution microscopy studies by Ref.¹⁵, who report similar homogeneous hematite distribution without localized EPS structures.

- (ii) **TEM vs. SEM/Raman observations:** The Reviewer correctly notes that Transmission Electron Microscopy (TEM) offers even higher spatial resolution. However, we emphasize that Ref.¹⁴, cited by the Reviewer, successfully identified EPS using SEM at scales comparable to ours (i.e., 1–20 μm). Given our own SEM analyses achieve finer resolution (tens of nanometers; SEM Merlin, see Methods), we are confident any filamentous or EPS-like structures would have been clearly detectable if present. Importantly, while the figures in the Extended Data of our manuscript provide a zoomed-out view to offer overall context, each specimen was carefully examined at high resolution (via SEM at 10 nm resolution) to resolve fine-scale features. Our extensive high-resolution analyses reveal no such EPS or filamentous structures, thereby supplementing our Raman-based results and supporting the interpretation of a minimal or negligible contribution of POC.
- (iii) **Sampling context of the Bakchar Deposit:** Regarding the Bakchar Fm study cited by the Reviewer (Ref.¹⁶), we emphasize that our Bakchar samples originate exclusively from the distinct “loosen goethite–hydrogoethite” horizon rather than the “goethite–berthierine/chamosite” horizon that was studied by Ref.¹⁶, which is known for siderite cementation and microbial EPS presence (see Ref.¹⁷ for geologic context; e.g., “The P-E Bakchar horizon consists of two types of ores: loose oolitic ironstone (thickness 0.2–11.4 m) occurs at the base of the horizon and is replaced higher in the section by cemented oolitic ironstone with siderite matrix (thickness 0.2–7.7 m)”, their Section 4.1). Notably, Ref.¹⁶ themselves explicitly distinguish between horizons that reflect abiotic processes and those that show clear microbial influences. Our rigorously selected samples exhibit intact, coherent lamination and lack siderite cement or EPS-related microbial features, further supporting the primary DOC origin of OC in these samples.

Figure 2: High-resolution SEM images comparing previously reported EPS-like features from hematite ooids (panel a, from Ref. ¹⁴, Chuanlinggou Fm, China) with our analyzed ooids. Panel b shows high-magnification view of a hematite ooid from the Mesoproterozoic Sherwin Ironstone (Australia), and panels c and d display hematite ooids from the Chuanlinggou Fm analyzed in this study. Despite matching the spatial resolution used by Ref. ¹⁴, we observe no evidence of filamentous or mucus-like structures in our well-preserved samples, supporting our interpretation of a homogeneous DOC-derived OC signal rather than particulate EPS.

Collectively, these observations—high-resolution SEM and Raman analyses, comparative imaging, and careful sample context considerations—support our interpretation that OC associated with our ooids is overwhelmingly derived from DOC rather than particulate sources or microbial EPS. Nevertheless, we fully acknowledge the Reviewer’s point regarding potential minor contributions of POC. Importantly, any POC contributions—if present—would require that true DOC concentrations were actually *lower* than those predicted by our record. Because predicted [DOC]* is already at or below modern values at all points in our record, this logic only *strengthens* our interpretation that there did not exist a large Neoproterozoic DOC reservoir. To explicitly account for this, we have clarified in the revised manuscript that our reconstructed DOC concentrations represent maximum estimates. Specifically, we have included the following statement:

L210–L213: “Importantly, because iron ooids capture both labile and recalcitrant DOC and may be contaminated by trace incorporation of POC, reconstructed [DOC]* should be treated as a maximum value and not equal to that of the deep ocean.”

We sincerely appreciate the Reviewer’s insightful comment, which has allowed us to significantly enhance the clarity of our manuscript and to logically reinforce the robustness of our main conclusions (i.e., no large Neoproterozoic DOC reservoir).

Comment 6: **Diagenetic/Thermal Alteration of samples:** A key issue is whether samples are diagenetically altered, and, if so, whether dissolution/precipitation could remobilize organic carbon or add it. The authors state such alteration is not an issue based on ooids being laminated and lacking clear microfossils (and something about the rims I didn't follow). Much more discussion of this is in the SI, but often it is a petrographic argument relying on geometric relationships. This is fine, but it assumes that recrystallization cannot retain fine structure, which it can. Such is termed mimetic recrystallization, and is common in terms of dolomitization of aragonite ooids. Here is an example from Corsetti et al. (<https://doi.org/10.1016/j.sedgeo.2006.03.021>) of Triassic age mimetic oolites preserving all the features seen here, but fully recrystallized.

Anyway, I am in general wary of assertions that diagenesis did not occur based on petrography and is not an issue. What would be helpful to see are measurements of samples they think are altered to see if it makes a difference.

Response: We thank the reviewer for highlighting the issue of diagenetic or mimetic recrystallization. We agree that the possibility of complete recrystallization—in which the original mineral phase is entirely transformed yet preserves fine-scale structure—cannot be ruled out *a priori*. To address this concern, we have implemented multiple independent lines of analysis (see Supplementary Discussion Section 6, particularly Section 6.4 on sensitivity tests and assumption validation, and Figs. ED22–ED120) that together indicate any diagenetic overprinting in our dataset is minimal. While we recognize and acknowledge that diagenesis has almost certainly impacted all units to some degree, we have adopted rigorous sample screening criteria to ensure that only specimens exhibiting uniformly fine, coherent lamination and intact crystal boundaries are included in our final dataset (as opposed to samples showing extensive overprinting, loss of lamination coherence, or textural disruption). We detail these points below:

- (i) The iron oxide ooids exhibit extremely fine laminations, on the order of tens of nanometers (as seen in e.g., Fig. ED114 of our manuscript as well as Fig. 2 of this review), that sharply contrast with the coarser laminae (typically tens of micrometers) observed in carbonate ooids. Moreover, we emphasize that the Corsetti et al. study cited by the Reviewer (Ref.¹⁸) concerns the mimetic recrystallization of aragonite into dolomite, which is a fundamentally different recrystallization process than one would expect for iron oxides—these mechanisms are not directly comparable. Specifically, aragonite ooids are inherently metastable and tend to recrystallize during early diagenesis¹⁹—often preserving mimetic laminae that remain relatively coarse. In contrast, our iron oxide ooids (both goethite and hematite) are thermodynamically stable minerals formed from a low-order ferrihydrite precursor¹¹, and they consistently display very fine laminations on the order of tens of nanometers. The thermodynamic stability of iron (hydr)oxides as compared to a *lack of* thermodynamic stability for aragonite is thus a crucial difference in the recrystallization mechanisms of these minerals. This pronounced difference in lamination scale, among other contextual indicators, provides a useful tool for ruling out samples that have been strongly overprinted by diagenesis.
- (ii) We carefully screened all samples to minimize the influence of late-stage diagenetic alteration on our Fe–OC signal. In particular, specimens that in preliminary screening displayed clear signs of alteration—for example, extensive overprinting of the original lamination, loss of coherent layering, or other textural disruptions—were omitted from our final dataset. For example, samples from the Ordovician Winnipeg Formation (see Fig. 3 of this review) exhibit evidence of halite precipitation and inconsistent ooid layering, suggesting transport or alteration—these samples were therefore excluded from our record. Only specimens displaying uniformly fine, consistent lamination and intact

crystal boundaries were retained. We stress that while our approach focuses on eliminating samples strongly affected by diagenesis, we do not claim that diagenetic processes were absent in the studied units. Rather, our selection criteria are designed to ensure that the preserved Fe–OC signal reflects primary or very early diagenetic characteristics, and this nuance is explicitly discussed in the revised manuscript (Supplementary Discussion Assumption (ii), “Iron ooids as primary or early secondary marine precipitates”; L3476–L3531).

Figure 3: Example of diagenetically altered samples excluded from analysis. The Ordovician Winnipeg Formation (Canada) shows clear evidence of post-depositional halite precipitation on the siliciclastic cement. In these altered samples, ooids appear to be older—possibly reworked from elsewhere—and exhibit partial halite enrichment. The halite distribution is inconsistent: in some areas it is nearly uniform, in others absent, and occasionally the original ooid lamination is no longer detectable. Panel (a) displays the SEM image, (b) shows the iron (Fe) distribution, (c) maps the sodium (Na), and (d) maps the chlorine (Cl) distribution. These altered characteristics were used as a screening criterion to exclude samples that might have been exposed to diagenesis that overprinted the primary signal.

- (iii) Finally, we address the Reviewer’s specific concern on thermal alteration. Given that we would generally expect older samples to be exposed to more thermal alteration—combined with the observation that all Precambrian samples contain hematite ooids—here we specifically assess the transformation of (primary) goethite to hematite. This transformation occurs via the reaction

where Δ indicates the addition of heat and can be evaluated quantitatively. Using typical estimates for molar volumes (where the molar volume of goethite is $\sim 20.64 \text{ cm}^3 \text{ mol}^{-1}$ and that of hematite is $\sim 30.42 \text{ cm}^3 \text{ mol}^{-1}$)²⁰, two moles of goethite occupy $\sim 41.28 \text{ cm}^3$, whereas one mole of hematite occupies $\sim 30.4 \text{ cm}^3$. This transformation therefore corresponds to a net volume contraction of

$$\frac{41.28 - 30.42}{41.28} \approx 26.35\%.$$

A volume contraction of this magnitude typically induces the formation of cracks or voids that are later infilled with secondary minerals exhibiting irregular textures. To assess our ability to identify this effect, we heated goethite ooids from the Cretaceous Hidra Fm (Israel) for 90 minutes at 450°C . XRD results indicate that goethite is completely transformed to hematite (not shown). More importantly, the presence of large-scale voids is clearly observed by optical microscopy (Fig. 4 of this review); these can be further detected using our elemental mapping approach. In contrast, no such cracks or voids were observed in any samples included in our record.

Figure 4: Goethite ooids from the Cretaceous Hatira Formation (Israel): panel (a) shows the ooids before thermal treatment, while panels (b) and (c) display the ooids post-heating (90 min at 450°C). White arrows indicate the voids generated by volume contraction during the goethite-to-hematite transformation. Due to irreversible thick section polishing, panels (a) and (b, c) show different ooids from the same hand sample.

In summary, while we agree with the Reviewer that recrystallization processes can in some cases preserve fine structure, in our study we explicitly screened for evidence of late-stage diagenetic alteration. Our combination of detailed petrographic examinations, quantitative evaluation of volume change during the dehydration reaction, and the consistent nanoscale lamination observed in our imaging maps together indicate that we have screened out the majority of samples that were affected by late-stage diagenesis. However, in the revised text we clearly articulate that we cannot fully preclude the possibility of diagenetic alteration, and we better acknowledge other possible explanations of our data. Specifically, we now state (new text in *italics*):

L3524–L3531: “*Furthermore, our petrographic and Raman spectroscopic observations—i.e., the preservation of fine laminations and uniform Fe-OC distributions—support an interpretation in which Fe-OC signals predominantly record primary seawater DOC rather than late-stage diagenesis (e.g., dissolution and recrystallization, potentially memetic in nature), post-depositional microbial reworking, or detrital POC inputs. We therefore interpret Fe-OC in both minerals as reflecting marine signals, but we cannot definitively exclude the possibility that samples have undergone some degree of diagenetic alteration, potentially impacting Fe-OC trends.*”

Comment 7: **Summary:** In summary, my primary concern is that the authors have not provided convincing evidence that the organic carbon they measure is exclusively derived from sorption processes and remains unaffected by diagenesis. I recommend that they address, in greater detail, the issues of potential contamination (e.g., from particulate organic carbon) and the effects of diagenetic alteration, thereby allowing the data to speak for itself without over-interpretation.

Response: We thank the Reviewer for summarizing their primary concern. We acknowledge that it is critical to demonstrate that the OC measured in our Fe-OC proxy is predominantly derived from DOC during mineral formation and not significantly influenced by POC or later diagenetic alteration. To address this, we have taken several steps and expanded our discussion as follows:

- (i) **Quantitative corrections for accessory phases:** We have employed SEM-EDS-based elemental mapping and XRF-based bulk elemental compositions (Fig. ED125) to quantify the fraction of non-iron phases in each sample. With the exception of the Presles Fm., all studied samples contain cores composed either of iron (hydr)oxides or of the OC-free minerals andesite and quartz, lending confidence to our interpretation that OC is associated with the iron (hydr)oxide phase and not accessory phases.
- (ii) **High-resolution imaging:** We have conducted high-resolution Raman microspectroscopy ($\sim 2.6 \mu\text{m}$ resolution) and high-magnification SEM imaging to assess the spatial distribution of OC within ooid cores and laminae. Our data consistently show a homogeneous distribution of OC that is confined to the iron oxide phases, rather than localized, patchy features that would typify POC or EPS. This observation is in stark contrast to patterns reported for EPS-associated organic matter in other studies (e.g., Ref. ^{14,16}). Still, we recognize that minor POC and/or EPS contributions to our observations could exist, and we now emphasize that any such contributions necessitate that our [DOC]* predictions are *maximum* estimates. Any potential POC associated with non-iron minerals would then only serve to lower the true DOC concentrations relative to these maximum predicted values. This result serves to strengthen our primary conclusion that the Neoproterozoic was not described by a large DOC reservoir.
- (iii) **Rigorous petrographic screening:** We have expanded our sample screening protocols to exclude ooids that exhibit signs of significant diagenetic overprinting, such as disrupted nanoscale lamination or recrystallization features. Only samples displaying well-preserved, coherent lamination and intact crystal boundaries were retained. This minimization of diagenetic alteration supports the interpretation that the preserved OC is representative of primary DOC incorporation. Still, despite our best sample screening efforts, we now explicitly state that diagenesis cannot be definitively ruled out, and we recognize that post-depositional processes might contribute to our observed signals.

Collectively, these measures—detailed imaging, quantitative correction for accessory phases, and rigorous screening—demonstrate that our Fe-OC signal most likely reflects and overwhelming co-precipitated DOC signal rather than major contributions from POC or late-stage diagenesis. While we acknowledge that trace amounts of POC or diagenetic modifications cannot be entirely ruled out, any such contributions would only lead to our record representing a conservative, maximum estimate of DOC concentrations. We believe these revisions and clarifications sufficiently address the Reviewer's concerns and enhance the robustness of our proxy interpretation.

In summary, we hope that the revised manuscript addresses all the Reviewers' comments satisfactorily. Thank you for considering our revisions.

Sincerely,
Nir Galili
(on behalf of all co-authors)

References

- [1] Benner, R. & Herndl, G. J. Bacterially derived dissolved organic matter in the microbial carbon pump. *Microbial carbon pump in the ocean* 46–48 (2011).
- [2] Curti, L. et al. Carboxyl-richness controls organic carbon preservation during coprecipitation with iron (oxyhydr) oxides in the natural environment. *Communications Earth & Environment* **2**, 229 (2021).
- [3] Eusterhues, K. et al. Fractionation of organic matter due to reaction with ferrihydrite: coprecipitation versus adsorption. *Environmental science & technology* **45**, 527–533 (2011).
- [4] Helms, J. R. et al. Absorption spectral slopes and slope ratios as indicators of molecular weight, source, and photobleaching of chromophoric dissolved organic matter. *Limnology and Oceanography* **53**, 955–969 (2008).
- [5] Chen, C., Dynes, J. J., Wang, J. & Sparks, D. L. Properties of Fe-organic matter associations via coprecipitation versus adsorption. *Environmental Science & Technology* **48**, 13751–13759 (2014).
- [6] Stockey, R. et al. Sustained increases in atmospheric oxygen and marine productivity in the Neoproterozoic and Palaeozoic eras. *Nature Geoscience* **17**, 667–674 (2024).
- [7] Knoll, A. H. Biomineralization and evolutionary history. *Reviews in Mineralogy and Geochemistry* **54**, 329–356 (2003).
- [8] Smith, F. A. et al. Body size evolution across the Geozoic. *Annual Review of Earth and Planetary Sciences* **44**, 523–553 (2016).
- [9] Saily, A., Smernik, R., Baldock, J. A., Kaiser, K. & Sanderman, J. The sorption of organic carbon onto differing clay minerals in the presence and absence of hydrous iron oxide. *Geoderma* **209**, 15–21 (2013).
- [10] Sposito, G. *The surface chemistry of soils*. (1984).
- [11] Cornell, R. M., Schwertmann, U. et al. *The iron oxides: structure, properties, reactions, occurrences, and uses*, vol. 664 (Wiley-vch Weinheim, 2003).
- [12] ThomasArrigo, L. K., Byrne, J. M., Kappler, A. & Kretzschmar, R. Impact of organic matter on iron (II)-catalyzed mineral transformations in ferrihydrite–organic matter coprecipitates. *Environmental Science & Technology* **52**, 12316–12326 (2018).
- [13] Mayer, L. M. Relationships between mineral surfaces and organic carbon concentrations in soils and sediments. *Chemical Geology* **114**, 347–363 (1994).
- [14] Lin, Y., Tang, D., Shi, X., Zhou, X. & Huang, K. Shallow-marine ironstones formed by microaerophilic iron-oxidizing bacteria in terminal paleoproterozoic. *Gondwana Research* **76**, 1–18 (2019).
- [15] Tang, D., Xie, B., Shi, X. & Zhou, X. Low level of phosphorous concentration in terminal paleoproterozoic shallow seawater: Evidence from chuanlinggou ironstone on north china platform. *Precambrian Research* **370**, 106554 (2022).

- [16] Rudmin, M. *et al.* Origin of ooids, peloids and micro-oncoids of marine ironstone deposits in western siberia (russia). *Journal of Asian Earth Sciences* **237**, 105361 (2022).
- [17] Rudmin, M. *et al.* Ferrimagnetic iron sulfide formation and methane venting across the paleocene-eocene thermal maximum in shallow marine sediments, ancient west siberian sea. *Geochemistry, Geophysics, Geosystems* **19**, 21–42 (2018).
- [18] Corsetti, F. A., Kidder, D. L. & Marenco, P. J. Trends in oolite dolomitization across the neoproterozoic–cambrian boundary: a case study from death valley, california. *Sedimentary Geology* **191**, 135–150 (2006).
- [19] Carlson, W. D. The polymorphs of CaCO_3 and the aragonite-calcite transformation. *Reviews in Mineralogy* **11**, 191–225 (1983).
- [20] Robie, R. A. & Bethke, P. M. Molar volumes and densities of minerals. Tech. Rep., US Geological Survey (1962).

The geologic history of marine dissolved organic carbon from iron oxides

Galili et al.

This study presents a laudable undertaking to develop the first proxy for past ocean dissolved organic carbon (DOC) signatures using co-precipitated organic carbon associated with iron oxide minerals in iron ooids. The idea that organic carbon associated with iron oxides might be used to reconstruct ocean DOC dynamics is original and really exciting, and if properly calibrated, this approach could make a significant contribution to the Earth sciences, because tracking ocean DOC dynamics over Earth history will allow better understanding of how DOC cycling links to and impacts oxygenation and climate. Following development of their ocean DOC proxy, the authors apply this to a suite of marine iron ooid-containing formations deposited over the past 1650 million years. Via an extensive series of commendable laboratory experiments, the authors present a calibrated proxy that predicts DOC concentrations were near modern levels in the Paleoproterozoic, decreasing by 90-99% in the Neoproterozoic before sharply rising in the Cambrian. These dynamics are in turn interpreted to reflect three ecological and biogeochemical stages in Earth's evolution, namely a progression from (i) small single-celled organisms with deep oceans that were severely hypoxic; to (ii) larger more complex organisms with little change in ocean oxygenation; towards (iii) continued organism growth and a transition to oceans that were fully oxygenated.

This proxy for ocean DOC dynamics is principally based on a set of clever laboratory experiments, that explore the controls on DOC coprecipitation with iron oxides, under scenarios designed to mimic those of appropriate natural oceanic settings. The authors have taken extreme care to identify areas of uncertainty and explore these both experimentally and computationally. The Supplementary Discussion is extensive. Despite this however, I have a number of concerns that I think might present some fundamental issues with the proxy that preclude publication of this study in its present form. I describe the main ones below.

1. Whilst the most common iron (oxyhydr)oxides to precipitate from oxic seawater and sediment porewaters in the modern ocean are undoubtedly ferrihydrite (initially), which then ages and transforms into more crystalline phases (like goethite), the oceans in the geologic past were silica-rich. Compared to modern seawater (<0.1 mM Si), it is likely that in the Archean, the oceans were probably saturated with amorphous silica (2.2 mM), or at least saturated with cristobalite (0.67 mM) (e.g., see Konhauser et al., 2007, *Science*, 315, p1234, and references therein), and during the proceeding timeline many studies show that silica concentrations in seawater were still significantly elevated compared to the present day (e.g., ~1.25 mM Si 600 million years ago, see Conley et al., 2017 *Frontiers in Marine Science*, 4, p397, and references therein). This means that the iron (oxyhydr)oxides in the past oceans were silica-rich, and this is a problem because iron-silica coprecipitation is well known to reduce the point of zero charge (PZC) of the resulting particle, which makes this particle much less reactive towards DOC. So to properly ground-truth this proxy, the experiments must be performed under dissolved silica concentrations that are representative of the past oceans. Unfortunately I cannot see a way around this.

2. I am also concerned by the choice of DOC types. Whilst the authors do a decent job of defending these, their proxy is predicated on the fact that marine DOC at all points in geologic time can be described by a mixture of modern-marine like, cyanobacterially derived and terrestrial soil-like DOC end-members (i.e., their MDOC, C-DOC and FA experimental materials). This is a very bold assumption. Even in the modern ocean we are still unable to fully characterise marine DOC and one of the most important components of the DOC pool, namely the enigmatic long-lived DOC (increasingly implicated in climate), is probably not represented very well by the molecular composition of any of their end-members. Studies indicate that at least a part of this DOC pool is rich in carboxyl groups and is highly aromatic (the so-called 'CRAM' component – carboxyl rich alicyclic molecules), so I would expect to see a representative (as far as possible) for this molecular composition, and others more representative of marine DOC in the experiments (e.g., see Hansell, 2013, *Annual Review of Marine Science*, 5, p421).

3. Furthermore I am concerned that the experiments do not capture how freshly coprecipitated amorphous iron (oxyhydr)oxides age and transform into more crystalline phases (like goethite), and how this affects the (re)distribution of organic carbon between the solid and solution. It's a little unclear in the methods but I think the iron-carbon coprecipitates were synthesised by forming ferrihydrite first then artificially aging into goethite or hematite (at elevated temperature, this is standard procedure – all except the goethite synthesis at circumneutral pH, which was achieved at 20°C over 10-120 days, again standard procedure and likely most representative of natural *in situ* aging). But because aging and transformation typically involves dissolution and reprecipitation, this process provides ample opportunity for the organic carbon initially taken up in the freshly coprecipitated phase to be lost to solution, or released and adsorbed to the mineral surface (later lost during the washing procedure) (e.g., see Zhao et al., 2022, *Geochimica et Cosmochimica Acta*, 335, p339 – this work uses elevated temperature to accelerate the aging of ferrihydrite into more crystalline phases, but shows that associated organic carbon is redistributed during the aging process). In this way the organic carbon retained by the final phases in the experiments does not necessarily directly correlate to the organic carbon retained by the analogous phases in natural settings, and thus to contemporaneous DOC.

4. I also think it's a problem that the authors assume the organic carbon associated with the iron oxides in iron ooids results from uptake of only *dissolved* organic carbon in the fluid from which the minerals precipitated. This assumption is valid for their experiments because their experiments contained only DOC, but in natural settings it is well documented that iron oxides with associated organic carbon arise via the uptake of DOC but also incorporation of particulate organic carbon (POC) – i.e., very tiny particles of organic carbon that are occluded / entombed (there are many terms), for example within mineral pore networks. Thus in natural samples the total organic carbon measured in a sample probably does not directly correlate to the contemporaneous DOC.

So unfortunately, I think there are some fundamental problems with the laboratory experiments that make the application of the proxy to the real world quite problematic.

Overall, I think that this manuscript tries to do too much in one go. To instigate a new proxy idea, develop this, calibrate this, ground-truth this, then use the proxy in earnest to infer major states of ecological and biogeochemical evolution is, in my opinion, too much. To be convinced that we have a proxy for ocean DOC through time, I would need to see calibration against laboratory experiments that fully represent iron-carbon dynamics in natural settings, performed under past ocean conditions, with additional DOC sources that might more closely match the important fractions of ocean DOC. The focus of the current manuscript is almost entirely on DOC dynamics, their importance and their potential role in ecological and biogeochemical evolution, with very little attention to the iron-carbon associations, and the increasingly large body of associated literature, that underpin the proxy. This is remiss. The vast amount of information currently presented in the Supplementary Discussion is crucial to the proxy and thus to the study interpretations and conclusions, and I think this deserves to be presented in a dedicated manuscript – the fact that iron-carbon coprecipitates (might) record past ocean DOC is a very significant finding in itself and warrants publication. Then following this, the application of this proxy to elucidate DOC dynamics over time, and relate these to ecological and biogeochemical evolution, could form a seminal publication in this field.

Review of:

The geologic history of marine dissolved organic carbon from iron oxides / 2024-02-02257

by Nir Galili and others

04/23/2024

I was hugely interested to read this paper and the novel proxy for ocean dissolved organic carbon (DOC) concentrations and hence global reduced carbon inventory. In addition, in trapping organic compounds in an accreting iron oxy-hydroxide (goethite) or oxide (hematite) as sedimentary iron-stones, there is the obvious potential to reconstruct past changes in global $\delta^{13}C$ perhaps free of diagenetic constraints (as the authors indeed do).

The authors are right in their argument for the value of a paleo DOC proxy, although they tend to overplay a little this argument at the start and cite published hypotheses for how changing marine DOC reservoirs could have impacted atmospheric pCO_2 as if this is how the system actually works ... but they are only hypotheses (many of which I happen to disagree with). (A minor nuance, as the important point is to generate new data to test the ideas that are out there.)

The main body of the paper itself is well written but could be better illustrated. I am more concerned at the extent of the Extended Data and SI discussion (yet in getting on for 200 pages, critical data are still missed out). This seems a very poor balance for a short format paper and really, much of the ED and SI should be aired fully as possible and in detail, as a paper in its own right. This also leaves the main paper too short and some of the ED figures, such as calibration curve, are central to the main text and should be there.

I will break my thoughts down into a couple of main items, and then simply list a number of smaller questions or concerns. Because the paper was well written, I do not have many typos or minor wording

issues ... I could list them, but I feel it is better to concentrate on some top-level questions.

'DOC' vs. LDOC+SLDOC+SRDOC+RDOC

The paper falls down overall, and in particular in respect of the DOC proxy, by failing to distinguish between the fractions of different lability (assuming that is even a word). Only 'DOC' as a single entity is considered, but the modern marine community has long recognized that the overall role and dynamics of DOC can only be understood by considering the production and fate of the constituent molecules, which range across an enormously wide spectrum and which are grouped for convenience of description and numerical modeling into e.g., labile DOC (LDOC), semi-labile DOC (SLDOC), semi-refractory (SRDOC), and refractory (RDOC). (Although often only a subset of these are considered and LDOC is usually ignored.)

The importance of this is because the large reduced carbon inventory cited by the authors of some 660 PgC, is dominated by RDOC (ca. 630 PgC), with minor contributions from SRDOC (14) and SLDOC (6) and almost from LDOC (with the balance made up by ultra-refractory components) (e.g., see: DOI: 10.1146/annurev-marine-120710-100757). RDOC is relatively uniform in concentration throughout the ocean, and because it dominates the total DOC pool at the surface, its presence suppresses the spatial variability in total DOC concentration at the surface.

In the SI, the authors present radiocarbon measurements on DOC trapped within modern iron ooids in an actively forming environment, and strongly argue that the 5,200 and 3,600 ^{14}C yr ages reflect the mean age of formation of the ooids which are in turn consis-

tent with estimated dates for the start of formation. In contrast, the lifetime of RDOC, which dominates the total DOC pool, is perhaps 16,000 years (or older if you include ultra-refractory DOC) (the authors not unreasonably reasonably cite 30,000 years). What is trapped in the modern ooids then must be short-lived SLDOC and/or SRDOC fractions which is consistent with mean DOC ^{14}C age being comparable to mean ooid age. So the proxy is not 'DOC', which is dominated by very old RDOC, but SLDOC (and/or SRDOC). This is an important distinction because it is the dominant RDOC reservoir that creates the potential to drive global changes in $\delta^{13}C$ and pCO_2 and all the cited references, and this is not what the proxy is sampling if the modern data and interpretation presented in the SI is correct. Consistent with this is the observation that RDOC is old because it is not being rapidly scavenged and removed from the ocean, which supports the ooid Fe-OC as being drawn from the much more reactive and little-aged SLDOC or SRDOC pool.

The concentration of SLDOC varies significantly at the ocean surface and will to a first order follow the very substantial variance that exists in primary production. Because of its short lifetime, SLDOC concentrations can change by an order of magnitude over the first 100-200 m down from the surface. The specific paleo environment, not just productive vs. unproductive, but the depth in the upper water column, is now important in the recorded Fe-OC concentration. The authors point to samples from different localities in the same time interval as supporting a global proxy, but a quick glance at the localities suggests they are paleogeographically proximal and/on the same carton and likely similar paleo environments. Inclusion of (e.g., ED Figure) maps of the paleo locations would have been an important addition to the paper and would have greatly helped in this question. It is hard to make out in the main figure – and I note that the data values are not given anywhere I could see in ED or SI which is a critical omission – but the light color symbols in Figure 1 do seem to span an appreciable range of Fe-OC (and $\delta^{13}C$) which would not have been expected if the total DOC pool was being sampled, which at the modern ocean surface spans no more than a factor of 2 (e.g., DOI: 10.1038/s41467-017-02227-3). If you exclude the high latitudes and focus on typical latitudes and coastal locations representative of the ooid samples, the range in total DOC is much smaller.

To me, the variance in the data seems more consistent with sampling SLDOC (and/or SRDOC) rather than the total DOC reservoir.

Obviously, if Figure 2B and DOC* reflect ambient SLDOC and not the DOC pool as a whole, some re-interpretation is required and trends in SLDOC could be telling us something about local primary productivity (which would also be useful).

I do note that the authors calibrated Fe-C to DOC/Fe where DOC is the total measured concentration of dissolved organic carbon. Even if Fe-C was SLDOC, as long as the ratio of SLDOC to total DOC remained constant through time, the calibration would work. That said, there will be little RDOC in the artificial cultures, and in the experiments, almost all the DOC will be SLDOC (and/or DRDOC). So DOC* should be reconstructed SLDOC (and/or DRDOC) concentrations (and still not bulk DOC).

Mineralogy

I find the profound offset in Fe-OC (and $\delta^{13}C$) between measurements made on iron oxy-hydroxide (goethite) vs. oxide (hematite) phases (Figure 1) very difficult to ignore. This offset is exacerbated in the reconstructed proxy values (Figure 2). The authors argue that the good match in DOC* for the ca. 450 Ma sample (again – everything is harder to interpret and understand without a data table (that I perhaps simply did not spot)) that mineralogy is not an issue, but yet there is no equivalent match for $\delta^{13}C$. DOC* values immediately prior to and following 450 Ma are consistent with all the other low DOC* hematite hosted values which requires there to be a major event at 450 Ma for the DOC* value match argument to work. On balance, the 2 different mineralogical archives do not look consistent and I am not yet convinced by the arguments suggesting that they are.

If we by-pass the Monte Carlo interpretation of the raw data (which was difficult to follow and buried in SI), just from the experimental calibration curves (ED10 and ED11 seem to be the key calibrations and it would have helped if these were isolated from the large number of other sensitivity test and promoted to the main paper), then hematite is trapping less DOC for the same DOC/Fe, but not quite enough to explain the raw Fe-OC data offset in Figure 1. Could the remainder of the Figure 1 Fe-OC offset, and a

loss of OC, result from recrystallization of primary goethite?

I don't have an explanation for the mineralogical offset. It could be telling us something important and useful. What I am not currently inclined to accept on what I have read, is that the offset is simply a really offset in DOC*.

Isotopes

$\delta^{13}C$ shows a large offset between goethite and hematite substrates with no value that overlaps between phases (unlike the 450 Ma point for DOC*). The 2 mineral phases exhibit polar opposite fractional behaviors which is interesting – ED Figure 10 vs. 11. ^{13}C fractionation on hematite declines with lower DOC/Fe, which is not unexpected in sign, although I am surprised by the magnitude of the fractionation as DOC/Fe tends to zero. How does this compare with organic compounds being absorbed onto, and/or incorporated into a growing lattice, for other substrates? Rather more surprising is the positive-with-declining-DOC/Fe behavior for goethite shown in EX Figure 10. Do the authors have a mechanistic explanation for this?

My suspicion would be that we are seeing a change in discrimination between different marine OC compounds, each with different $\delta^{13}C$ – i.e., bulk marine OC $\delta^{13}C$ actually represents the weighted mean of a variety of different compounds positively and negatively fractionated compared to bulk DOC (e.g., DOI: 10.1146/annurev-marine-121916-063634). This is expressed at low DOC/Fe with different compounds being incorporated into goethite vs. hematite, while at very high DOC/Fe, perhaps everything (all different molecules) 'sticks'. If so, ecosystem composition, in determining the initial mix of compounds, could drive changes in the recorded $\delta^{13}C$, even for the same DOC/Fe and same bulk OC $\delta^{13}C$. In this interpretation – that $\delta^{13}C$ in Figure 2D for hematite deviates so far from kerogen, simply reflects that hematite is sampling a range of more depleted compounds compared to bulk organic matter.

So my concern is what the Fe-OC is in terms of compounds, because I have no reason to think it reflects bulk DOC, or even bulk SLDOC. What is needed is for the Fe-OC to be characterized in terms of compounds (ultimately, with compound specific ^{13}C measurements). For the modern, actively-forming

oids, if this was paired with measurements of seawater SLDOC (+SRDOC), or if this was just done in the experiments, I think this proxy would be very significantly advanced.

As a note – taking the corrected $\delta^{13}C$ (Figure 2C) at face value, I wonder whether the authors had considered a stronger methane cycle in the early Paleozoic and Precambrian, creating DOC compounds with a much lighter signal?

Other

- I cannot make out the changes associated with the PETM in the figures (and no data values are provided). This sounds really interesting ... a paper in itself almost ... In the main text, SI discussion was promised, but I could find hardly anything.
- The assumption is made that the Fe^{II} flux to the site of ooid formation is constant through time, and hence the Fe value in the DOC/Fe based calibration is constant. It seems unlikely that every iron ooid location had a very similar hydrothermal environment to presumably much less than a factor 2 in the Fe term.

2 modern sites are discussed – in the Italy Aeolian archipelago and near Mahengentang Island in Indonesia. What is known about Fe fluxes here? If nothing, what about hydrothermal heat fluxes or some proxy for activity that we might relate to Fe^{II} flux? How similar or different are these formation environments?

- (lines 129-131 plus ED121) To check for latitudinal bias in this way with a regression, you need to plot the variable (y-axis) vs. some measure of distance from the equator, regardless of whether you are going North or South. What you cannot do is put a straight line through the absolute values from -90 to 90 and conclude anything.

Recommendations

I don't think this is a proxy for past bulk ocean DOC and global reduced carbon reservoir (and hence it is not directly addressing the global carbon cycle questions that serve as the motivation for this paper), and I don't think as a new and emerging proxy, that it is quite ready for the big time.

However, in sampling some set of organic molecules (and associated ^{13}C signature) and preserving them unaltered, the prospects seem good to say something about evolution in the marine environment. I would be really excited to see what can be done with this proxy.

Please find enclosed my review of Galili et al.. This paper examines an important and highly underdetermined problem in Earth History — the amount of dissolved organic carbon (DOC) in the oceans over geologic time. As the authors lay out well at the start of the paper, ~20 years ago Rothman et al. proposed that variations in the $\delta^{13}\text{C}$ of carbonates, especially large depletions in the Neoproterozoic could be explained by the oxidation of a larger dissolved organic pool in the ocean. This would be a non-steady state system (which has been criticized by others) and some have argued the oxidation then gave rise to an oxygenated world.

Much ink, including in *Nature*, has been spilt on this issue and how to interpret these excursions. The problem has always been there is not a record of this phantom organic pool. Even without this motivation, we do not know this term well in the Phanerozoic (or any other time period) and as such, constraining it would be an important new record to better understand the evolution of the Earth. Put another way, this is, in my opinion, a really good problem to try to solve.

The authors propose a solution which is to measure the organic carbon content of iron ooids as a measure of the DOC pool. The idea is that dissolved organic carbon (and many other species) commonly sorb to iron oxide minerals. As ooids grow, they could then entrap this sorbed organic material. Similar sorts of approaches have been used for iron minerals including to try to derive phosphate concentrations (e.g. work of Don Canfield and followed up by others).

The authors conducted experiments to show, convincingly, that dissolved organic carbon concentrations correlate to uptake by experimental iron ooids. Then then do the harder (but warranted) thing and explore the record in past and find, in my opinion, little variation in the organic carbon content, which, taken at face values, dismisses the idea of a huge DOC reservoir in the Neoproterozoic. They interpret various wiggles in more detail, but the overall message is things have not change by much over geologic time in terms of the DOC reservoir.

The basis of the approach, in and of itself, is sound under the following conditions: (1) the only organic carbon ever taken up by the iron minerals in the ooids was sorbed. This means no inclusion of particulate organic carbon from other sources at the time of growth. And no incorporation of other minerals in the iron that could contain organic carbon either sorbed to that surface or detrital in origin. (2) The organic carbon content is preserved in time and does not change either from diagenesis or catagenesis (thermal heating). I remain unsure about these final points and did not find the issue of ‘contamination’ by particulate organic carbon to be well-addressed nor the issue of diagenetic removal.

I discuss these in more detail and provide some prior studies on iron ooids that argued particular organic carbon being present in iron ooids as well as some other issues. As it stands, I do not consider the data sufficiently robust to warrant the interpretations taken at this time. But, I’m not entirely sure that should preclude publication. Rather, I would recommend the authors be a bit more frank about the issues, assumptions they make, and then let the data be what it is without declaring things (such as no diagenesis, no contamination, etc.).

Comment 1: Discussion of sorption in general.

When I think of sorption to minerals, I do not think in terms of terms like OC/Fe $^{3+}$ or OC wt%. Rather, the key term will ultimately be the amount of reactive surface area available for sorption

and the number of layers that can form. This is the thermodynamic driving variable, but surface area is only stated once, on line 2288. Thus, as far as I understood it, the model implicitly is assuming that the surface area directly correlates to the weight percent and does not change as a function of growth environment, etc. This may be a good assumption, but I think it is important that the relationship between the fundamental sorption process and iron weight percent be laid out. Here is an example that examines sorption of organics on clay and iron oxides both in terms of weight percent (as here) and surface area:

<https://doi.org/10.1016/j.geoderma.2013.05.026>

Comment 2: Way the data is shown.

Figure two A is the key data figure in terms of loading. In looking at 2A , the plot is given in log space. I have a few comments. I recommend the figure be much wider so that the data can be seen better. Second, I downloaded the excel data set, and plotted it up. The total range of 0.03 to 0.96 wt%, so ~1.5 order span. In my opinion, this doesn't need a log plot to show, which creates artificial differences at the low vs. high end, especially at the total range is 0 to 100%. Here is the plot without the suppression in the log space (x time, y wt. %)

This shows the larger scatter at given time frames (that isn't explained that I saw) and the dearth of points in the Precambrian. Furthermore, it shows the max values in the Mesoproterozoic are sort of the same as the lowest in the modern. Regardless, this plot both in its very small size and log space obscures, in my opinion the actual data for the reader to sort out.

Comment 3: Organic carbon in non-hematite portions of the ooids?

The most important requirement of the paper is that the organic carbon must be sorbed carbon that is structurally bound in the mineral lattice. The authors deal with this by doing Raman spectroscopy and looking if organic carbon is or is not present in ~3 micron size pixels. I didn't see this data for all samples, nor an example of something where POC is present (maybe I missed

it). But what worries me about this approach is that it assumes there are not a mixed sources where POC and amorphous (previously sorbed) OC are both present. I'm not sure if these were generated for all samples, but in looking at figure 1D (Aseri formation), my worry is that the core, which is not the concentric ooid part I think they care about, has organic carbon in it. I can't tell from this figure if the core is also an iron oxide, but regardless, that core is not what they want and has organic carbon.

Additionally, the samples are generally not pure iron oxide. Based on the data provide, here is the wt.% of the samples studied increasing order:

About 1/3 of samples have more than 20% other stuff, and about half 15% other stuff. If these are clay minerals with 1% organic carbon, they could contribute easily 0.1 to 0.2 wt% to the total. We do not know what these materials are, but they could easily skew the results. I note that the authors consider this in the text, but assume the accessory phases have no organic carbon. I do not believe they provided evidence for this. Regardless, that some samples are 25% something else, is a serious issue if they contain any organic carbon that I strongly recommend the authors deal with.

Comment 4: Presence of particulate organic carbon

Another concern is the presence of particulate organic carbon trapped in the ooids. The authors again argue this is not the case based on the Raman data. I'm not sure at the 3 micron scale this could be seen, and the SEM images provided are not zoomed in enough for me to see. The way I would look for particulate organic carbon is via TEM imaging in order to zoom in. I did a search of "TEM and iron ooids and organic carbon" on google scholar and a number of studies came up. Here is an example from the Bakchar formation (<https://doi.org/10.1016/j.jseaes.2022.105361>), which was studied here as well:

transmission electron microscope image (A) of the micro-encrusted structure showing nanofibrillar forms (org). The background consists of isometric crystals. In the top are filaments of organic carbon (red arrow) and the 'org' in the second is the organic carbon. The authors interpret this as follows: *micro-encrusted structures with filaments and EPS (Fig. 5), and traces of organic matter in Raman spectra of micro-encrusted structures (Fig. 10) suggest biogenic activity. The filaments and EPS in micro-encrusted structures suggest the activity of iron-reducing bacteria within iron-rich micro-encrusted structures in form of the organo-mineral intergrowth.*

I'm not sure who is right, but Galili says EPS is not present, but I do not believe their imaging or Raman can test this as they are too zoomed out to see the patchiness at the sub-micron level needed (i.e., by TEM). But the study referenced above finds the patchy organic carbon and interprets it as EPS from biological activity where the ooids are forming. If right, and generalizable, it would violate the primary assumption of the paper.

As another example from Lin et al. (<https://doi.org/10.1016/j.gr.2019.06.004>) for Paleoproterozoic ooids from the Chuanlinggou formation (also studied here), they interpreted the structure below in yellow as 'mucus like filaments' and in general state the following: SEM and EDS analyses show that these micrometer-scaled laments are enriched in organic matter and have a mucus-like morphology (Fig. 6B), thus providing further evidence that they were derived from bacterial remains, possibly from EPS

Again, I'm not sure who is right, but there are examples in which samples were studied at high resolution and interpreted to have EPS present, contra what is stated in this paper.

Comment 5: Diagenetic/Thermal Alteration of samples

A key issue is whether samples are diagenetically altered, and, if so, whether dissolution/precipitation could remobilize organic carbon or add it. The authors state such alteration is not an issue based on ooids being laminated and lacking clear microfossils (and something about the rims I didn't follow). Much more discussion of this is in the SI, but often it is a petrographic argument relying on geometric relationships. This is fine, but it assumes that recrystallization cannot retain fine structure, which it can. Such is termed mimetic

recrystallization, and is common in terms of dolomitization of aragonite ooids. Here is an example from Corsetti et al. (<https://doi.org/10.1016/j.sedgeo.2006.03.021>) of Triassic age mimetic oolites preserving all the features seen here, but fully recrystallized.

Anyway, I am in general wary of assertions that diagenesis did not occur based on petrography and is not an issue. What would be helpful to see are measurements of samples they think are altered to see if it makes a difference.

Summary

Overall, my main concern is that the authors have not presented sufficient evidence that what they assert to be measuring in terms of sorbed organic carbon is what is being measured and, if so, has not been modified by diagenesis (and what they diagenesis should do, increasing or decreasing TOC wt. %).